# Dipterocarpoidae genomics reveal their demography and adaptations to Asian rainforests

Rong Wang [1,17] ✉, Chao-Nan Liu[1,17], Simon T. Segar[2,17], Yu-Ting Jiang[1,17], Kai-Jian Zhang[3,17], Kai Jiang [1,4], Gang Wang [5], Jing Cai [6], Lu-Fan Chen[1], Shan Chen[1], Jing Cheng[1], Stephen G. Compton [7], Jun-Yin Deng[1], Yuan-Yuan Ding[1], Fang K. Du[8], Xiao-Di Hu[3], Xing-Hua Hu[9], Ling Kang[3], Dong-Hai Li[10], Ling Lu[1], Yuan-Yuan Li[1], Liang Tang[11], Xin Tong[1,4], Zheng-Shi Wang[1], Wei-Wei Xu[3], Yang Yang[1], Run-Guo Zang[12], Zhuo-Xin Zu[3], Yuan-Ye Zhang [13] ✉ & Xiao-Yong Chen [1,14,15,16] ✉

Dipterocarpoideae species form the emergent layer of Asian rainforests. They are the indicator species for Asian rainforest distribution, but they are severely threatened. Here, to understand their adaptation and population decline, we assemble high-quality genomes of seven Dipterocarpoideae species including two autotetraploid species. We estimate the divergence time between Dipterocarpoideae and Malvaceae and within Dipterocarpoideae to be 108.2 (97.8–118.2) and 88.4 (77.7–102.9) million years ago, and we identify a whole genome duplication event preceding dipterocarp lineage diversification. We find several genes that showed a signature of selection, likely associated with the adaptation to Asian rainforests. By resequencing of two endangered species, we detect an expansion of effective population size after the last glacial period and a recent sharp decline coinciding with the history of local human activities. Our findings contribute to understanding the diversification and adaptation of dipterocarps and highlight anthropogenic disturbances as a major factor in their endangered status.

The tremendous loss of global biodiversity poses a major threat to ecosystem functioning and human welfare[1,2]. The persistence of species depends on adaptations to paleo and contemporary environmental changes, including Quaternary glaciations[3–5] and anthropogenic disturbances[6]. Genome-wide studies hold substantial potential to explore phylogenetic and demographic history alongside mechanisms of adaptation[7–9]. Uncovering the molecular footprints behind adaptations and factors shaping demographic history is thus of utmost importance for biodiversity conservation.

Asian rainforests are a critical biodiversity hotspot[10,11], and the cover of such rainforests is probed by plants from the subfamily Dipterocarpoideae (Dipterocarpaceae), termed dipterocarps[12]. This subfamily is distributed across Southeast Asia, India, and the Seychelles, comprising at least 470 species in 13 genera[13]. Recent genomic studies[13–15] found a whole genome duplication (WGD) event preceding the divergence of Dipterocarpoideae species. This WGD event increased the chromosome number of Dipterocarpoideae ancestors to 12, and the following two steps of karyotype evolution gave rise to 11-chromosome species (Dipterocarpeae tribe) and 9-chromosome species (Shoreeae tribe)[13]. Earlier studies proposed that Dipterocarpaceae originated in Western Gondwanaland in Early Cretaceous (c.120 million years ago (MYA))[16]. A recent phylogenomic study[13] supports this Western Gondwanaland origin, showing a very ancient divergence between

Dipterocarpaceae and its closest family Malvaceae (147.3 MYA[13]). New evidence from pollen fossils suggested the diversification of Dipterocarpaceae since *c*.102.9 MYA[17], and indicated that the dispersal of dipterocarps from Africa to India occurred during the late Cretaceous through Kohistan-Ladakh Island Arc and subsequently to Southeast Asia during the middle to late Eocene after India-Asia collision. In contrast to these findings, genomic studies involving different sets of dipterocarp species revealed a more recent divergence between Dipterocarpaceae and Malvaceae (*c*. 84 MYA[14] and 86–98 MYA[15]). Therefore, a controversy still surrounds the

evolution of these rainforest indicator species; resolving this will provide valuable insights into the origins of megathermal angiosperms.

Dipterocarps are renowned for their giant adults, which can reach up to more than 100 m in height, comprising the emergent layer and over half the canopy space of Asian rainforests[16,17] (Fig. 1a). This unique ecological niche has facilitated the diversification of Dipterocarpoideae[16,17]. As a consequence of above-canopy life in Asian rainforests, these species are challenged with various abiotic stresses such as irregular droughts[15] and intensive ultraviolet (UV) radiation[18].

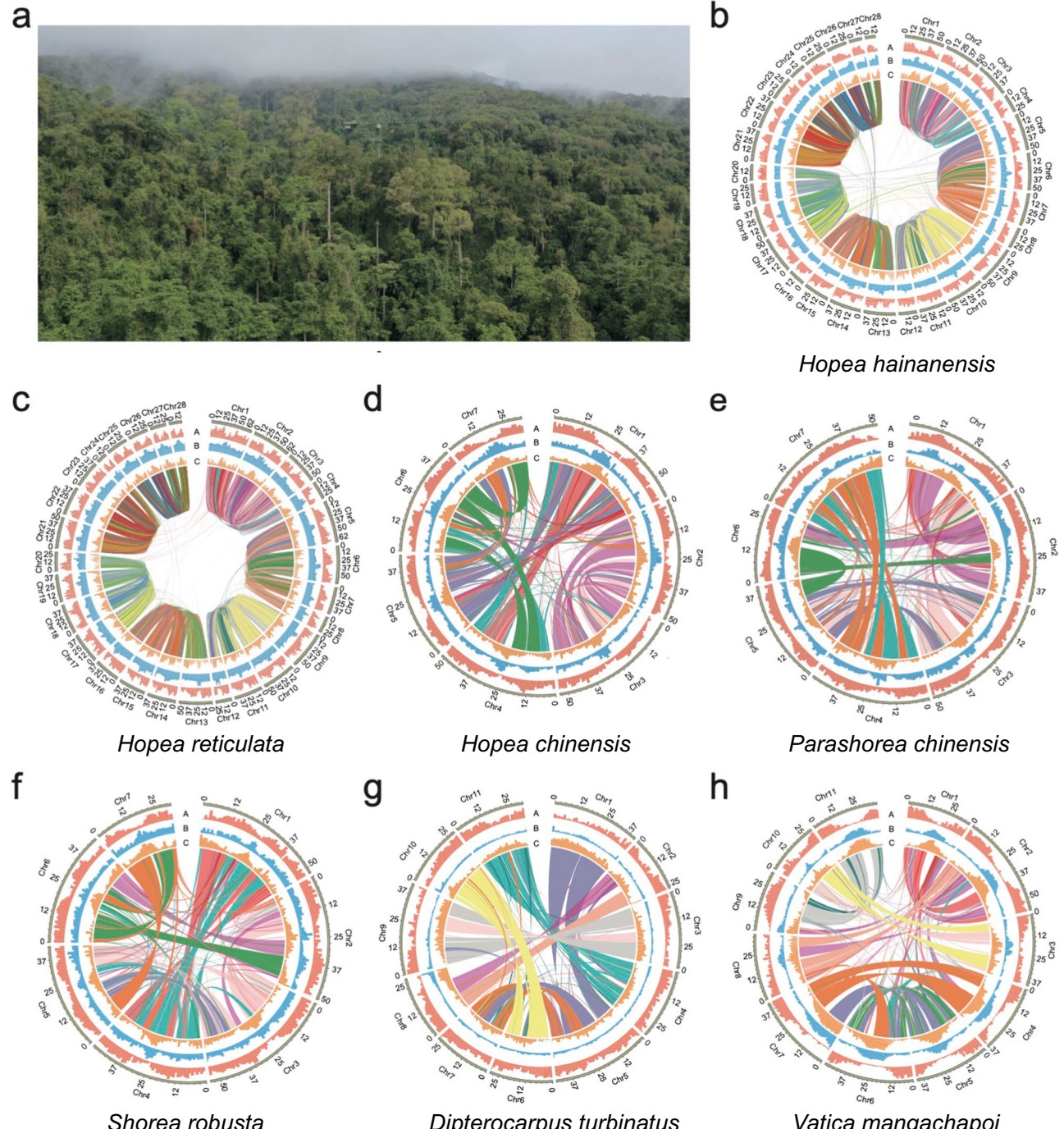

**Fig. 1 | Genomic statistics for our focal Dipterocarpoideae species.** The Asian rainforest where Dipterocarpoideae species dominate the emergent and canopy layers (**a**), and statistics for the assembled genomes of seven focal species (**b**–**h**). **a** Most Dipterocarpoideae trees at/above the canopy layer are taller than 50 m, with the height of the tower being 80 m (photo was taken by Hui Chen from Xishuangbanna, China). **b**–**h** Synteny analysis showed that most conserved syntenic blocks only exist within each set of four homologous chromosomes. A: Gene density. B: GC content. C: Repeat density.

However, the molecular basis underlying the adaptation and diversification of dipterocarps remains unexplored. Comparing dipterocarps to temperate forest trees can provide a robust answer to this question.

Alarmingly, 75% of all dipterocarps are listed as threatened species (https://www.iucnredlist.org/). Due to their high-quality hardwood and moisture resistance, dipterocarps are a major source of timber, which has led to their overexploitation[13,14]. While frequent logging and land use change have resulted in 0.6% of Asian rainforests disappearing annually[11,19], population contraction of many species may have started since the last glacial period (LGP) as Southeast Asia became cooler and drier[20,21]. This range contraction is in accordance with other tropical rainforests[22]. However, this view has been challenged because species distribution models suggest that Dipterocarpoideae underwent range expansion during the LGP, due to an increase in habitat area resulting from sea level deline[12,23]. Inferring historical population sizes using genomic data will provide critical evidence to resolve the debate around threats to species persistence and factors driving population decline.

Here, we performed de novo genome assembly of seven Dipterocarpoideae species, including five species from the Shoreeae tribe and two species from the Dipterocarpeae tribe[24] (Table 1). We then conducted resequencing and population genomic studies on two endangered species (*Hopea hainanensis* and *Hopea reticulata*) endemic to Hainan Island, China. We acknowledge that previous genomic studies have sequenced several Dipterocarpoideae species. Taking advantage of these genomic studies, we aim to (1) conduct a more comprehensive phylogenomic analysis of Dipterocarpoideae to resolve the controversy around their origin; (2) identify positively selected genes related with the adaptation to the emergent layer and tropical forest environments; and (3) reveal the demographic history and infer factors contributing to their endangered status. These efforts have revealed that the divergence between Dipterocarpoideae and Malvaceae and within Dipterocarpoideae occurred at 108.2 (95% HPD: 97.8–118.2) MYA and 88.4 (95% HPD: 77.7–102.9) MYA, supporting the tropical-African origin of Dipterocarpoideae. Our study identified a few genes associated with the adaptation of Dipterocarpoideae, and suggested that the current small population size of Dipterocarpoideae species was primarily driven by human disturbances rather than climatic changes. Our results will be of considerable value in future efforts aimed at conservation of this emblematic group of trees.

## Results

### De novo genome assembly and annotation

To provide high-quality reference genomes for evolutionary and population genomics analyses, we assembled the genomes of the seven Dipterocarpoideae species using PacBio sequencing, Illumina sequencing and high-throughput chromatin conformation capture (Hi-C) technologies (see Methods). *k*-mer analysis revealed that all of the species sequenced in this study are diploid, except for *H. hainanensis* and *H. reticulata* (Supplementary Fig. 1a). Karyotype analysis also supported that *H. hainanensis* is autotetraploid (Supplementary Fig. 1b). Moreover, by using *Solanum lycopersicum* (2C = 2.12) as a reference, the flow cytometry results revealed the C-values of *H. chinensis, H. hainanensis* and *H. reticulata* are 2C equal to 0.81, 1.60, and 1.64 (Supplementary Fig. 1c and Supplementary Table 1). When testing samples of *H. hainanensis* or *H. reticulata* together with the diploid *H. chinensis*, the clear peak on the right side of the diploid peak confirmed the tested species as tetraploidy (Supplementary Fig. 1c). We also detected that homologous chromosomes in each of these two species shared highly conserved syntenic blocks (Fig. 1b, c), further supporting that they are autotetraploids. Besides the trees sampled for de novo assembly, we performed *k*-mer analysis for all (30) sampled *H. hainanensis* and four selected sampled *H. reticulata* trees for resequencing and found evidence for their tetraploidy (Supplementary Fig. 1d).

**Table 1 | Summary of genome assembly and annotation statistics of seven Dipterocarpoideae genomes**

| | Shoreeae | | | | | Dipterocarpeae | |
| --- | --- | --- | --- | --- | --- | --- | --- |
| | Hopea hainanensis | Hopea reticulata | Hopea chinensis | Parashorea chinensis | Shorea robusta | Dipterocarpus turbinatus | Vatica mangachapoi |
| Basic chromosome number/Ploidy | 7/4 | 7/4 | 7/2 | 7/2 | 7/2 | 11/2 | 11/2 |
| k-mer estimated genome size (Mb) | 1563.72 | 1500.49 | 346.47 | 313.14 | 328.97 | 373.17 | 452.80 |
| Assembled genome size (Mb) | 1099.23 | 1237.93 | 339.45 | 313.85 | 327.27 | 387.22 | 447.80 |
| Contig N50 (Mb) | 0.22 | 0.10 | 1.36 | 9.23 | 11.78 | 32.75 | 6.66 |
| Hi-C anchored rate (%) | 95.77 | 99.36 | 99.65 | 98.46 | 99.84 | 95.29 | 98.15 |
| Scaffold N50 (Mb) | 44.28 | 48.13 | 50.75 | 46.06 | 48.47 | 32.79 | 39.50 |
| GC content (% of genome) | 32.79 | 33.14 | 33.32 | 32.32 | 33.02 | 34.55 | 35.12 |
| Repetitive sequences (% of genome) | 49.39 | 47.48 | 44.22 | 43.64 | 45.40 | 50.53 | 51.51 |
| Number of annotated genes | 88,341 | 84,932 | 29,234 | 29,498 | 28,950 | 31,604 | 30,807 |
| Non-coding RNAs (% of genome) | 0.06 | 0.08 | 0.11 | 0.09 | 0.07 | 1.58 | 0.32 |

The k-mer estimated genome size and assembled genomes of H. hainanensis and H. reticulata include four sets of chromosomes, and only monoploid genomes are estimated and assembled for the other five species.

The estimated whole genome sizes of the two autotetraploid species, *H. hainanensis* and *H. reticulata*, were 1.56 Gb and 1.50 Gb, with the estimated monoploid genome sizes of the five diploid species ranging from 313.14 to 452.80 Mb (Table 1, Supplementary Fig. 1 and Supplementary Table 2). We generated 42.63–124.88 Gb (52.81–398.80 X) Pacbio CLR (continuous long-read) reads for *Hopea chinensis*, *Parashorea chinensis* and the two autotetraploid (Supplementary Table 2). The assembled whole genome sizes for the two auto-tetraploid species were 1.10 Gb and 1.24 Gb, and the assembled monoploid genome sizes for *H. chinensis* and *P. chinensis* are 339.45 and 313.85 Mb (Table 1). The difference between estimated and assembled genome sizes of the two autotetraploid species was largely due to the presence of highly homologous regions among the four monoploid genomes (361.86 Mb for *H. hainanensis* and 245.66 Mb for *H. reticulata*) (Supplementary Table 3). The assembled genome size plus the overlapping homologous regions account for 93.6% and 99.1% of the estimated genome size of *H. hainanensis* and *H. reticulata*. The assembled monoploid genome sizes for *Shorea robusta*, *Dipterocarpus turbinatus* and *Vatica mangachapoi* were 327.27–447.80 Mb based on 6.64–20.72 Gb Pacbio HiFi reads (Table 1 and Supplementary Table 2).

A total of 47.11–100.29 Gb (49.45–153.88 X) Hi-C reads were produced for the seven Dipterocarpoideae species (Supplementary Table 2). We generated Hi-C-based physical maps for each auto-tetraploid species to assemble 28 pseudo-chromosomes, with the anchored rate of 95.77% (1.05 Gb) for *H. hainanensis* and 99.36% (1.23 Gb) for *H. reticulata* (Fig. 1b, c, Table 1, Supplementary Fig. 2a, b and Supplementary Table 4). The scaffold N50 of the assembled genomes reached 44.28 Mb and 48.13 Mb, respectively (Table 1). In addition, 95.29–99.84% of the assembled monoploid genomes of the other five species were anchored to 7 or 11 pseudo-chromosomes (Fig. 1d–h, Table 1, Supplementary Fig. 2c–g and Supplementary Table 4). CEGMA and BUSCO assessments detected complete structure of 92.3–98.8% core eukaryotic genes and 95.0–99.0% conserved embryophyta proteins in our assembled genomes, and the mapping ratio of Illumina short reads was at least 94.45%, covering 92.38–99.99% regions of genomes (Supplementary Table 5).

We annotated 88,703 and 85,031 protein-coding genes (including alleles, see Supplementary Table 6) in the assembled genomes of *H. hainanensis* and *H. reticulata* and 29,104–31,744 protein-coding genes in the assembled monoploid genomes of five diploid species (Table 1 and Supplementary Table 7). Repetitive sequences accounted for 43.6–51.5% of the assembled genomes (Supplementary Table 8).

## Phylogenomics and genome evolution

To investigate the phylogenetic relationships among Dipterocarpoideae species and their divergence with the closest family Malvaceae, we performed phylogenomic analysis including 19 Dipterocarpoideae species (seven Dipterocarpoideae species assembled in this study plus 12 Dipterocarpoideae species from Tian et al.[13]) and other 6 plant species (see Supplementary Table 9). The maximum likelihood (ML) tree was constructed with 304 single-copy orthologs (from 21,109–48,040 clustered gene families (Supplementary Fig. 3 and Supplementary Table 9)). Using MCMCTree based on fossils and secondary calibrations[15,17,25,26], the time-calibrated phylogeny dated the divergence of Dipterocarpoideae and Malvaceae at 108.2 (95% HPD: 97.8–118.2) MYA, and the split within Dipterocarpoideae was estimated to have occurred at 88.4 (95% HPD: 77.7–102.9) MYA (Fig. 2a).

To test the reliability of this estimation, we used different subsets of calibrations in independent analyses. The calibrated trees based on the calibrations from either Bansal et al. [17] or Ng et al.[15] revealed similar divergence estimates between Dipterocarpoideae and Malvaceae (106.8 (95% HPD: 93.8–116.8) MYA and 102.5 (95% HPD: 83.6–122.2) MYA). Although the median crown age

of Dipterocarpoideae from the calibrations of Bansal et al. [17] (89.3 MYA) was earlier than that from Ng et al. [15] (65.8 MYA), the 95% HPD estimated from the two analyses overlapped (76.4–99.7 and 48.1–109.8 MYA) (Supplementary Fig. 4a).

Both the synonymous substitution rate (*Ks*) distribution and fourfold degenerate synonymous site (4DTv) analyses detected a whole genome duplication (WGD) event (WGD-1) across all focal Dipterocarpoideae species (Fig. 2b and Supplementary Fig. 4). Quota tests showed that most homologous regions in the genome of *Vitis vinifera* (Vitaceae), which only experienced the ancestral whole-genome triplication of the eudicots, were doubled in our assembled genomes, suggesting that WGD-1 was a paleotetraploid event (Supplementary Fig. 5 and Supplementary Table 10). The genetic distance of paralogs within these species was found to be greater than the distance of orthologs between pairs of species, indicating that WGD-1 occurred before the diversification of Dipterocarpoideae species (Fig. 2b and Supplementary Fig. 4b). This finding is consistent with previous genomic studies[13–15], suggesting that WGD facilitated the diversification of Dipterocarpoideae species.

The distribution of *Ks* and 4DTV distances showed that the genetic distance between paralogs within each of the two tetraploid species (*H. reticulata* and *H. hainanensis*) was smaller than the distance between orthologs in any pairwise comparison of the three *Hopea* species (*H. chinensis*, *H. reticulata* and *H. hainanensis*) (Fig. 2b and Supplementary Fig. 4b). This result suggests the whole genome duplication event (WGD-2), corresponding to the tetraploid species, occurred more recently than the divergence of these *Hopea* species. Since the whole genome duplication in *H. reticulata* and *H. hainanensis* took place after their divergence, the occurence of tetraploidy is lineage-specific.

## Genomic signatures associated with adaptation

To explore the molecular footprints associated with the adaptation of Dipterocarpoideae, we conducted comparative genomic analyses between five diploid Dipterocarpoideae species and five species of temperate trees (i.e., temperate tree comparison) (Supplementary Table 9). This comparative analysis was expected to identify the positively selected genes adapted to the stresses of tropical environments.

The comparative analysis identified 192 positively selected genes (Fig. 3a). As the number of genes was limited and the corresponding KEGG and GO enrichment analysis yielded no significant results, we presented the functional annotation of GO and KEGG of these genes instead (Supplementary Tables 11 and Supplementary Data 1). After searching published literature, 20 positively selected genes were found to be associated with plants response to environmental stresses (Supplementary Figs. 6 and 7; Supplementary Data 2). These genes were involved in DNA repair, abnormal mRNA degradation, antioxidation, anaerobic respiration, endoplasmic reticulum associated degradation, pathogen resistance and cellular energy homeostasis (Supplementary Data 2). Among these genes, seven (e.g. DNA repair protein RAD51D[27] and structural maintenance of chromosomes protein 5 (*SMC5*)[28]) were detected to be associated with UV adaptation; nine (e.g., carotenoid ε-hydroxylase (*LUT1*)[29], and riboflavin kinase (*RFK*)[30]) and five genes (e.g., molecular chaperone GrpE (*GRPE*)[31]) were related with drought and heat tolerance; we also found three and two genes participating in the plants' responses to salt stress and flooding. These results thus revealed the molecular footprints likely contributing to the adaptation of dipterocarps to environmental stresses in Asian rainforests.

All these genes were expressed in all the five diploid Dipterocarpoideae species (Supplementary Fig. 8). Furthermore, to validate the function of the positively selected genes, we selected UDP-sugar pyrophosphorylase (*USP*), and to further validate the results of functional annotation, the gene pyridoxine 4-dehydrogenase (*PLR1*)) was also chosen. We used the sequences of these two genes from the

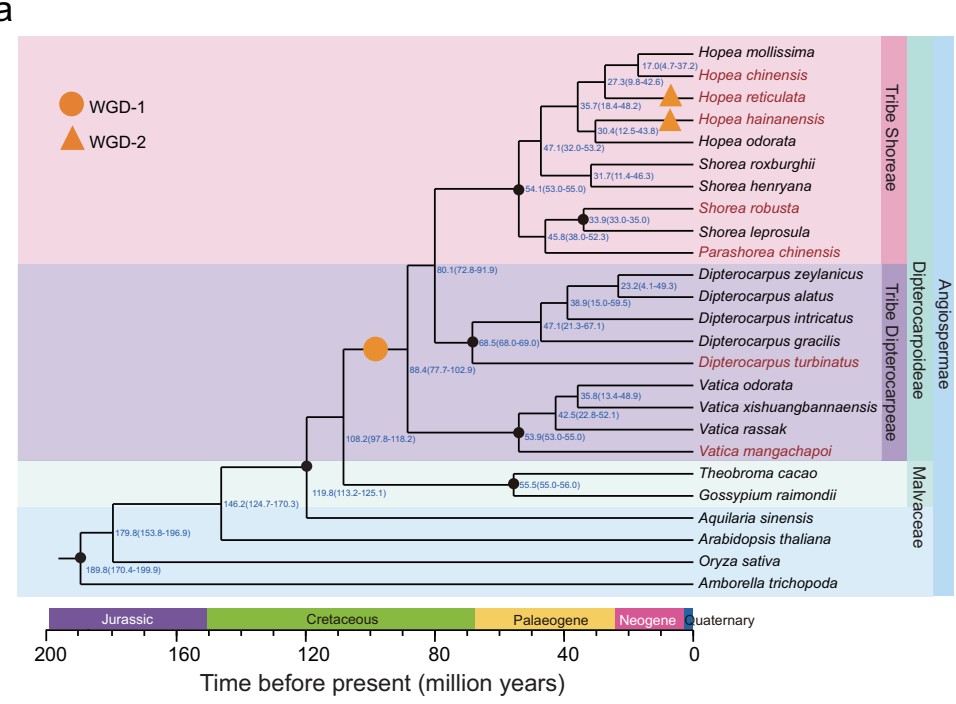

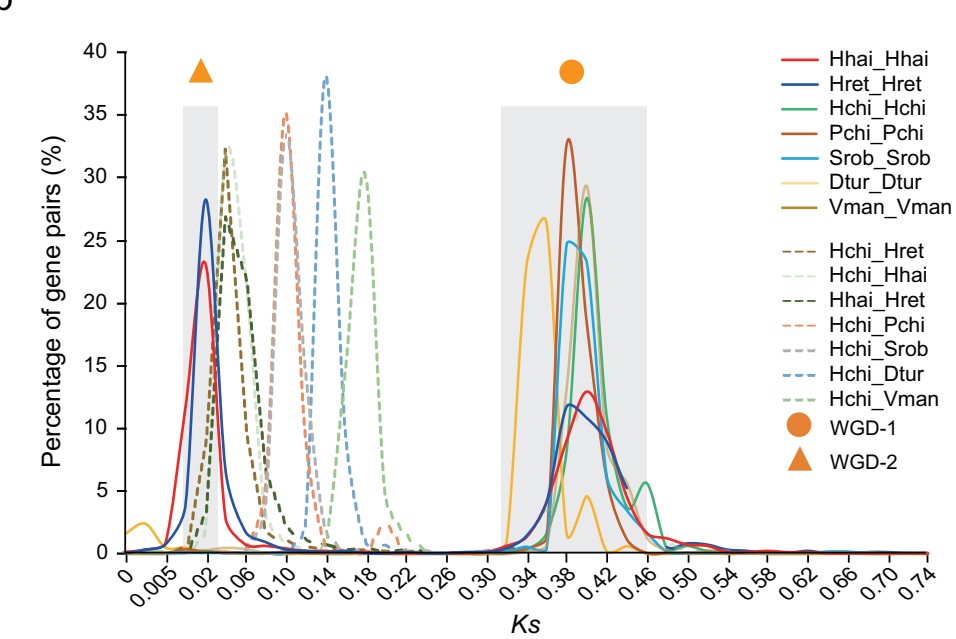

**Fig. 2 | Phylogenomics and genome evolution of our focal species.** Results of phylogenomic (**a**) and whole genome duplication (WGD) (**b**) analysis. **a** The phylogenetic tree was constructed using the genomes of 19 Dipterocarpoideae species and two Malvaceae species (with our focal Dipterocarpoideae species highlighted in red and their samaras are shown to distinguish different species) (Supplementary Fig. 4b). The black dots represent the time calibrations for estimating the divergence time from Bansal et al. [17], Ng et al. [15], Bell et al. [25] and Vega et al. [26], respectively. WGD events are represented by an orange circle (WGD-1) and orange triangles (WGD-2). **b** *Ks* distributions of the seven Dipterocarpoideae species are represented by different colors, and abbreviations of species are shown in Supplementary Table 2.

---

genome of *H. chinensis* and synthesized their recombinants (Supplementary Table 12), which were used to conduct in vitro enzyme activity assay (see Methods). The final products of the in vitro reactions identified by LC-MS are consistent with the standards (Fig. 3b), validating the enzyme activity of the two genes in synthesizing UDP-glucose and pyridoxine, respectively.

The comparison revealed three expanded and 53 contracted gene families (Fig. 3a; Supplementary Table 13). Among the expanded gene families, we identified one family of disease-resistance proteins (Supplementary Tables 14 and 15), which is absent in the temperate tree species and is specific to Dipterocarpoideae species (Fig. 3c). Ten contracted families were annotated in the plant-pathogen interaction pathway, including seven families of leucine-rich repeat-receptor-like kinases (LRR-RLKs), two families of coiled coil-nucleotide binding site-leucine rich repeat (CC-NBS-LRR) and one toll interleukin receptor-nucleotide binding site-leucine rich repeat family (TIR-NBS-LRR) (Fig. 3c, Supplementary Tables 14 and 15). These gene families are

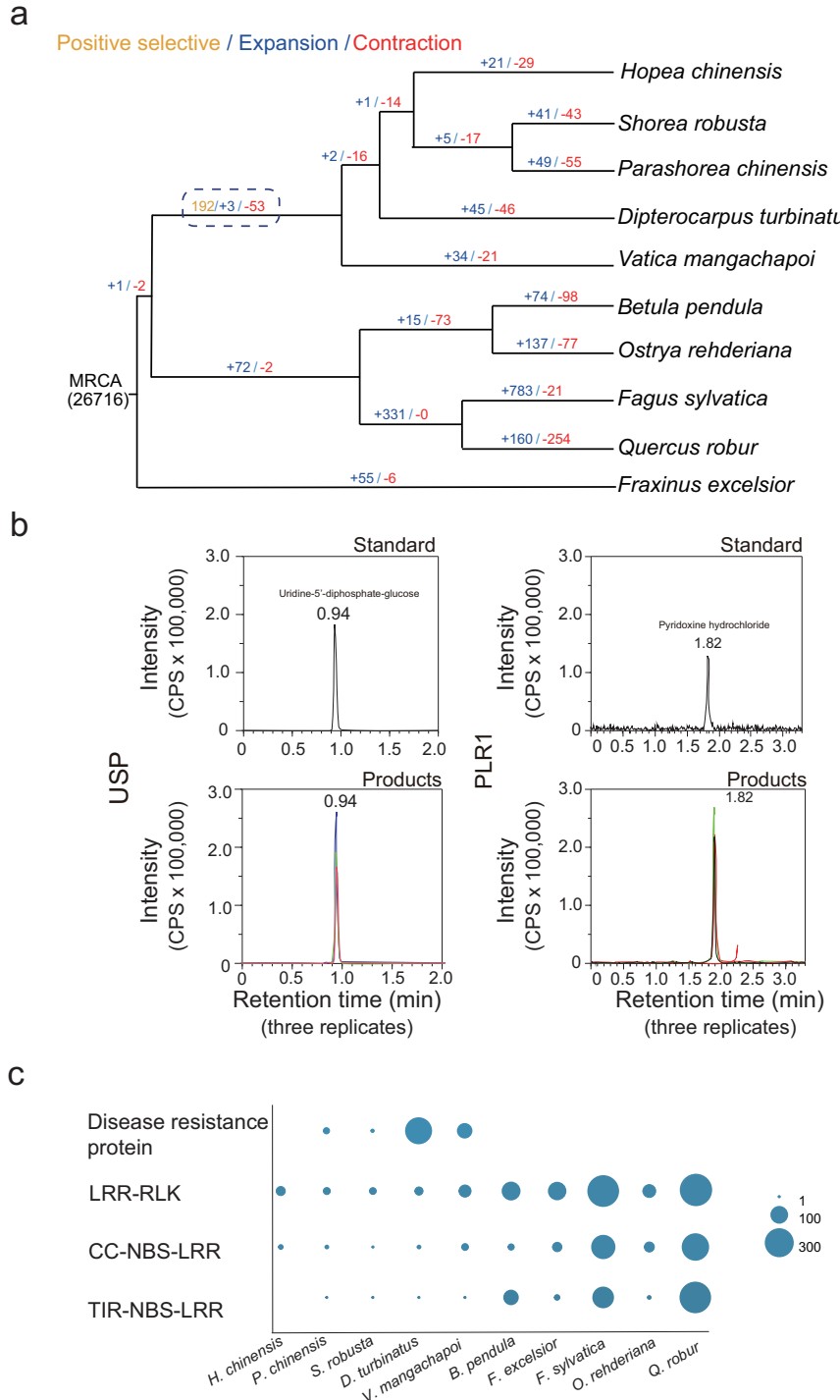

**Fig. 3 | Results of comparative genomics.** Comparative genomics between five focal Dipterocarpoideae species and five temperate tree species (**a**), results of in vitro functional characterization of two antioxidation-related genes (**b**), the number of genes in some significantly expanded and contracted gene families (see Supplementary Table 15, **c**). **a** The number of genes with significant signals of positive selection and the number of significantly expanded/contracted gene families are represented by yellow, blue and red, respectively. MRCA the most recent common ancestor. **b** The synthesized standards (at the concentration of 10 ng/ml) and reaction products (treatments with enzyme added for three replicates) for each gene (Supplementary Table 12) were identified using LC−MS. **c** Abbreviations of species are shown in Supplementary Table 2, and source data are provided as a Source Data file.

the major protein receptors through which plants recognize pathogens[32,33]. Given that the contraction of gene families is often related to specialization in biotic interaction[34], the observed contraction in gene families relating to plant-pathogen interaction is likely to be the molecular footprint of adaptation to specific pathogens endemic in Asian rainforests.

## Demographic dynamics and genetic load of *H. hainanensis* and *H. reticulata* populations

To reveal population demographic dynamics and evaluate the genetic load of Dipterocarpoideae species, we collected samples of *H. hainanensis* and *H. reticulata* from their remnant populations on Hainan Island, China, where we collected 30 and 32 wild trees

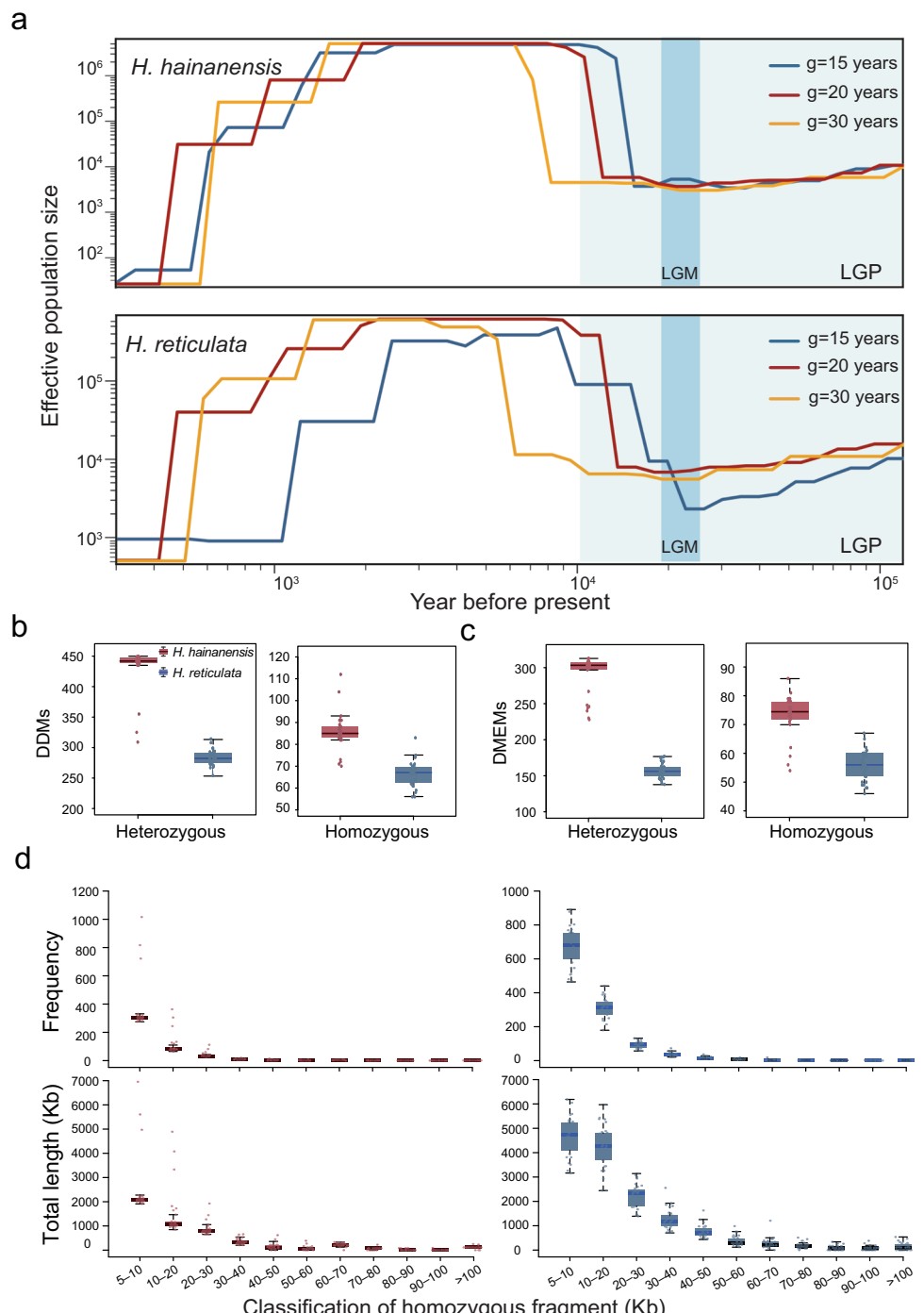

**Fig. 4 | Demographic history and genetic load of *H. hainanensis* and *H. reticulata*.** Demographic history of *H. hainanensis* and *H. reticulata* (**a**), genetic load metrics of *H. hainanensis* and *H. reticulata* (**b, c**), and box plots of frequency distribution and total lengths of homozygous fragments in different length categories in the genome of each sampled tree of *H. hainanensis* and *H. reticulata* (**d**). **a** Demographic dynamics of *H. hainanensis* and *H. reticulata* were inferred with three different values of generation time (15, 20, and 30 years) using SMC++, and the Last Glacial Period (LGP) and the Last Glacial Maximum (LGM) is highlighted in light cyan and cyan, respectively. **b, c** Boxplots show the accumulation of the derived deleterious mutations (DDMs) and the derived major-effect mutations (MEMs) in individuals sampled from populations of *H. hainanensis* and *H. reticulata*. Boxplots (**b–d**) for both *H. hainanensis* and *H. reticulata* display biologically independent samples ($n = 30$ and $32$) as data points. The center lines, box edges, and whiskers represent medians, the 25% and 75% quartiles, and the upper and lower distribution of 1.5 times of quartile range. Source data of **b–d** are provided as a Source Data file.

respectively (see Methods). Resequencing data (average depth >37 X; Supplementary Table 16) revealed a total of 4,386,932 and 9,623,768 variants for *H. hainanensis* and *H. reticulata*, of which 99.35% and 98.13% were bi-allelic, including 2,922,653 and 4,477,756 single nucleotide polymorphism loci (SNPs) (Supplementary Table 17). We found relatively similar genetic diversity in these two species

($H. hainanensis$: $\pi = 3.31 \times 10^{-3}$, $H. reticulata$: $\pi = 3.97 \times 10^{-3}$, Supplementary Fig. 9).

The demographic analysis with PSMC[35] showed a minor declining trend in effective population sizes ($N_e$) for both *H. hainanensis* and *H. reticulata* from one hundred thousand years ago and continuing throughout Last Glacial Maximum (LGM, *c.* 19,000 years ago)

(Supplementary Fig. 10). For more recent demography, the SMC++[36] analysis based on three different generation times (15, 20, and 30 years; Supplementary Table 18) revealed that the $N_e$ of both species increased after the LGM, reaching a maximum six to ten thousand years ago for *H. hainanensis* and three to nine thousand years ago for *H. reticulata* (Fig. 4a). However, a recent sharp decline started from around 1,500–2,400 years ago for *H. hainanensis* and 1,200–2,400 years ago for *H. reticulata*. During this period, effective population sizes decreased from the maximum (*c.* 5.0 ×10⁶ for *H. hainanensis* and *c.* 4.0 ×10⁵–6.0 ×10⁵ for *H. reticulata*) to less than 1000 (Fig. 4a). Since human activities in Hainan Island can be traced back to 3,000 years ago[37], the above evidence suggests that human intervention may have disrupted the maintenance of their populations after the glacial period.

To explore the consequences of small populations, we estimated the genetic loads and inbreeding for *H. hainanensis* and *H. reticulata*. Overall, we identified the 678 derived deleterious mutations (DDMs) from 647 genes in *H. hainanensis* and 617 DDMs from 581 genes in *H. reticulata* (Supplementary Fig. 11; Supplementary Table 19). We also characterized the derived major-effect mutations (DMEMs), including 416 DMEMs from 407 genes in *H. hainanensis* and 304 from 300 genes in *H. reticulata* (Supplementary Table 19). Among these deleterious mutations, 200 DDMs (195 genes) and 180 DMEMs (175 genes) in *H. hainanensis* and 197 DDMs (195 genes) and 152 DMEMs (150 genes) in *H. reticulata* were homozygous in at least one individual, and the corresponding genes were mainly enriched in pathways like starch and sucrose metabolism (map00500) (Supplementary Table 20). In particular, our analysis revealed that individuals of both species carried a very low number of homozygous deleterious mutations, with an average of 85 ± 1.5 DDMs and 73 ± 1.3 DMEMs for *H. hainanensis* and 66 ± 1.0 DDMs and 56 ± 0.9 DMEMs for *H. reticulata* (Fig. 4b, c). In comparison to another endangered plant species, *Ostrya rehderiana*, which has a similar monoploid genome size (366.2 Mb) and an average of 5350 homozygous DDMs in an individual[9], our results showed a relatively low genetic load in the two *Hopea* species.

The population genetic structure analysis revealed that all samples of *H. reticulata* were clustered into only one group (optimal $K = 1$, Supplementary Fig. 12), but samples of *H. hainanensis* were assigned into two groups (optimal $K = 2$, Supplementary Fig. 13). Intriguingly, we found samples collected from isolated sites (BW and WZ) were assigned into the same group (Supplementary Fig. 13). The runs of homozygosity (ROH) analysis revealed very few long homozygous fragments (>30 Kb) in all sampled trees, indicating low levels of inbreeding in both species (Fig. 4d). Consistent with the dynamics of effective population sizes, these findings suggest that these populations have undergone a decline in population size quite recently.

## Discussion

Exploring the processes underlying adaptive diversification and population demography can provide critical insights into the conservation of endangered species[3,38,39]. In this study, we assembled high-quality genomes of seven Dipterocarpoideae species, and identified the divergence time between Dipterocarpoideae and Malvaceae and within Dipterocarpoideae as 108.2 MYA and 88.4 MYA (Fig. 2a). This comprehensive phylogenomic analysis suggests the tropical-African origin of both Dipterocarpaceae and Dipterocarpoideae followed by Africa-India floristic interchange[17]. Positive selection was identified in a few genes potentially associated with the adaptation of Dipterocarpoideae to tropical environments. Notably, two autotetraploid *Hopea* species showed evidence for a post-LGP increase in effective population sizes. It is highly likely that human disturbances in the last few thousand years have driven population declines in these species. Despite their small population sizes, two autotetraploid *Hopea* populations accumulated low levels of deleterious mutations, probably because a bottleneck occurred recently (less than 170 generations).

Our results thus provide novel insights into how Dipterocarpoideae species have adapted to Asian rainforests yet eventually become endangered.

The estimated divergent time between Dipterocarpoideae and Malvaceae in this study appears to be earlier than the results from Ng et al.[15] and Wang et al.[14], and later than Tian et al.[13]. The estimated crown age of Dipterocarpoideae overlaps with the previous estimation by Bansal et al.[17] (~94.6 MYA). Although the estimations in this study did not contradict the crown age of eudicots (*c.* 132 MYA)[40], it indicates an earlier origin of Malvales than previously[40], and appears to support the ancient origin of Dipterocarpaceae (*c.*120 MYA) in Western Gondwana[16]. This origin is more recent than the separation of Eastern and Western Gondwana, and the dispersal of dipterocarps to India likely occurred through the Kohistan-Ladakh Island Arc, which collided with the Indian Plate in the late Jurassic[17]. The dispersal to Southeast Asia was likely associated with the collision between the Indian and the Asian Plates in the late Eocene[17]. In particular, the genus *Hopea* was primarily distributed across Southeast Asia, and the estimated crown age of this genus (35.7 MYA) in our study suggests its diversification after the India-Asia collision. Thus, by integrating more genomes, our results support the tropical-African origin of Dipterocarpoideae and the role of Africa-India floristic interchange in forming the current distribution of Dipterocarpoideae.

The divergence time between Dipterocarpaceae and Malvaceae seems to be not strongly impacted by the calibrations within Dipterocarpoideae, while the median crown age of Dipterocarpoideae appears to be sensitive to the within-subfamily calibration(s). The broader range of 95% HPD using the calibrations of Ng et al.[15] is likely due to the inclusion of only one calibration within this subfamily. Intriguingly, the emergence of new fossil records of Dipterocarpaceae from the late Cretaceous, Paleocene, and Miocene[17] supports an earlier timeframe for phylogenomic dating, resulting in an earlier crown age estimation. This phenomenon mirrors the extension of the history of angiosperms[41], where the discovery of more ancient fossils provides novel opportunities to test the biogeographic hypotheses. In addition, it is worth noting that our results are only based on Dipterocarpoideae genomes, and novel insights associated with the origin of Dipterocarpaceae are likely to be provided when high-quality genomes from the other two subfamilies (Monotoideae and Pakaraimaeoideae) are available.

WGDs and polyploidization are thought to be connected to environmental stresses such as drastic variations in temperature[42,43]. Detection of WGD-1 and WGD-2 probably reflects a genome-scale adaptation to ancient climatic/environmental changes. We found that the WGD-1 preceded the diversification of Dipterocarpoideae species, consistent with several previously published studies on genomes in this clade[13–15]. This event was previously thought to coincide with the Cretaceous–Paleogene extinction event (*c.* 66 MYA)[15], but our phylogenomic results showed a much earlier occurrence in the Mid-Cretaceous. Although it is still difficult to confirm the specific climatic/geological events associated with this WGD event, this is likely to happen in tropical Africa, where Dipterocarpoideae originated[17]. Karyotype analysis reveals that multiple species within the genus *Hopea* have polyploid populations[44–46], including *Hopea odorata* (3X), *Hopea subalata* (3X) and *Hopea nutans* (4X). Polyploidy appeared to be commonly observed in habitats characterized by climatic and edaphic fluctuations, and most documented polyploidization events to date are closely linked to environmental changes[47]. The diversification of this genus is quite recent in Dipterocarpoideae, with a crown age of 35.7 (95% HPD: 18.4–48.2) MYA (Fig. 2a), corresponding to the late Eocene[17]. The temperature decline and fluctuations in the Pliocene, along with irregular drought events in this area[15,44] may have contributed to genome duplications or polyploidizations of this lineage. While the previous study suggested the existence of diploid populations of *H. hainanensis*[14], *k*-mer analysis revealed all individuals of *H.*

*hainanensis* sampled in this study from Hainan Island as tetraploids (Supplementary Fig. 1d). Such distribution ranges of *H. hainanensis* and *H. reticulata* are located at the northern boundary of Asian tropical zone, where significant fluctuations in climates (e.g., cooler winters and highly variable precipitation) are anticipated compared with central regions of Asian rainforests. These environmental stresses may be relevant to the polyploidization of these two *Hopea* species.

Rapid evolution of functional genes usually takes place in response to environmental stresses[7,48,49]. Dipterocarpoideae species benefit greatly from the height of their adult trees, by which they preempt light resources and facilitate pollen and samara dispersal[16,50], but these advantages come with costs, such as direct exposure of adult trees to intense UV and high vulnerability to droughts. As the positively selected genes are functionally associated with these stresses[27–30], our results offer the molecular basis for further exploring the adaptation of Dipterocarpoideae to environmental stresses in Asian rainforests. The contracted gene families in plant-pathogen interaction suggest that these species may be specialized to endemic pathogens in Asian rainforests. Future studies should explore the function of these gene families of Dipterocarpoideae and their specialized interaction with pathogens.

Consistent with the relatively small effective population sizes of many temperate and subtropical species during the last glacial period[5], the post-LGM population expansion of two autotetraploid *Hopea* species suggests they prefer humid and hot climate[17]. Importantly, our results indicate the dominant role of human disturbances in driving population decline, in common with many species[51]. As Dipterocarpaceae is well known for fragrant oleoresins and high-quality timber since ancient ages[13,14], many Dipterocarpaceae species have long-term suffered overexploitation and illegal logging. Moreover, the land use changes due to human colonization and development have caused serious deforestation of Asian rainforests[11,19]. All of these disturbances are likely to have hastened the recent and drastic decline in population sizes. Note that the SMC++ curves of these two *Hopea* species showed slightly shifts in time, which is likely to reflect heterozygosity differences between them rather than real demographic/geological-event differences[52].

Small effective population sizes are expected to threaten the long-term maintenance of endangered species[39]. In accordance with this, we found relatively low genetic diversity, comparable to some other endangered species (Supplementary Fig. 9). Nevertheless, we did not observe the accumulation of deleterious mutations. Due to the masking effect of tetraploidy on deleterious mutations, tetraploids are predicted to accumulate more deleterious mutations than diploids over time[53]. One possible explanation is that the population decline of *H. hainanensis* and *H. reticulata* occurred only recently, over the last 2000 years, and deleterious mutations have not yet accumulated. Furthermore, Dipterocarpaceae species are primarily pollinated by small-sized insects, such as tiny thrips, and the gene flow is anticipated to occur within a few hundred meters[16,54]. This suggests that the geographically isolated but genetically similar populations of *H. hainanensis* were historically connected, providing further evidence that fragmentation and population size decline occurred only very recently. Hence, although the effects of small population sizes have not yet manifested, they will pose a major challenge for future conservation as deleterious mutations accumulate over time.

Ongoing global changes are leading to the rapid deterioration of habitats in which species have long evolved[2,55]. Our results reveal the phylogenetic history of Dipterocarpoideae and genomic signatures underlying their adaptive diversification, and suggest that anthropogenic disturbances, rather than glacial climate fluctuations, have led to population decline. These findings thus provide the historical context for one of the world's most charismatic tree groups. Our work emphasizes the importance of reducing human activities associated with habitat loss and contributes to biodiversity conservation in Asian rainforests.

## Methods
### Genome assembly

For de novo assembly of the seven Dipterocarpoideae species, we collected fresh leaves from one individual of each species in Hainan, China (*H. hainanensis* and *H. reticulata* were sampled in Bawangling (109°07′E, 19°06′N) and Ganshiling (109°39′E, 18°22′N), respectively. The remaining five species were sampled at the Research Institute of Tropical Forestry, Chinese Academy of Forestry (108°47′E, 18°42′N). Genomic DNA was extracted using DNAsecure Plant Kits (TIANGEN), and at least 10 μg of sheared DNA for each sampled tree was obtained for sequencing. For each species, one pair-end library was prepared with an insert size of 350 bp for sequencing on an Illumina HiSeqX platform. Genome size and ploidy were estimated using $k$-mer analysis as implemented in Jellyfish v2.1.3[56] (https://github.com/jamesturk/Jellyfish).

The number of peaks and the ratio of depth of the homozygous peak to that of the leftmost heterozygous peak for each species suggested that *H. hainanensis* and *H. reticulata* are autotetraploid species. To confirm the chromosome number and ploidy of *H. hainanensis*, we conducted karyotype analysis based on fluorescence in situ hybridization. Root tips collected from a seedling were treated with nitrous oxide and were fixed in ice-cold 90% acetic acid. The root tips were then digested by cellulase and pectinase, and dividing cells were collected by centrifugation at $1700 \times g$ for 2 min. The cell suspension was dropped onto glass slides, and we used the probe solution containing 1× TE buffer (pH7.0), the telomere specific probe Telo (TTTAGGGTT-TAGGGTTTAGGG) labeled with 6-carboxyfluorescein (6-FAM) and the probe of 18S rDNA (a 9 Kb EcoRI fragment of 18SrDNA in the pTa71 plasmid) labeled with fluorescein-12-dUTP (green) for in situ hybridization on each glass slide. After hybridization, the glass slides were mounted with the mounting medium containing 4′,6-diamidino-2-phenylindole (DAPI). Images were captured using an Olympus BX-53 microscope equipped with a DP-70CCD camera.

To further test whether *H. hainanensis* and *H. reticulata* are tetraploid species, we estimated their genome sizes and ploidy using flow cytometry. For these two species and *H. chinensis* (a diploid congener), we collected the fresh young leaves from the sampled trees used for the de novo genome assembly. We also sampled young leaves of *Solanum lycopersicum* (genome size: 2.07 Gb; 2C = 2.12 pg[57]) as the reference for genome size estimation. We first estimated the genome sizes and 2C values of the three *Hopea* species by measuring the fluorescence of nuclei from the mixture of each *Hopea* species and *S. lycopersicum* and those from each single species. Then, we assessed the ploidy for *H. hainanensis* and *H. reticulata* based on their fluorescence of nuclei setting *H. chinensis* as the diploid reference. The nuclei suspension was prepared for each species following the methods described in Ng et al.[44]. We measured the fluorescence from the nuclei suspension using a Sysmex CyFlow Cube6 flow cytometer, and data analysis was performed using FCS Express 7plus.

We carried out PacBio single-molecule real-time sequencing based on 20 Kb SMRTbell libraries using a PacBio Sequel platform. Continuous long-read (CLR) mode was used for *H. hainanensis*, *H. reticulata*, *H. chinensis*, and *P. chinensis*, and circular consensus sequencing (CCS) mode was used for *S. rubusta*, *D. turbinatus* and *V. mangachapoi* to generate Pacbio long reads. The assembly of draft contigs were performed for *H. hainanensis*, *H. reticulata*, *H. chinensis* and *P. chinensis*, based on Pacbio CLR reads using Falcon and Falcon Unzip v0.4.0[58]. Assembly was followed by error correction using Quiver (Smrtlink v6.0.1)[59] and further polishing and correction using Illumina reads as implemented in Pilon v1.22[60] (https://github.com/broadinstitute/pilon). For the other three species, Hifiasm (0.8-dirty-

r280)[61] (https://github.com/chhylp123/hifiasm) was used to assemble Pacbio HiFi reads into contigs. Purge_haplogs v1.1.0[62] (https://github.com/FullHuman/purgecss) was then used to remove the heterozygous regions of Pacbio assemblies for all species, except the two auto-tetraploid species (*H. hainanensis* and *H. reticulata*).

We then used the Hi-C technique to scaffold contigs into pseudo-chromosomes for each species. We constructed Hi-C libraries according to the protocol described in Belton et al. [63]. Fresh leaves sampled from the same trees used for initial sequencing were used for chromatin fixation by 4% formaldehyde solution, followed by an overnight digestion with a four-cutter restriction enzyme MboI (400 units) at 37 °C, we repaired DNA ends using biotin-14-dCTP and blunt-end ligation of the cross-linked fragments. Next, the proximal chromatin DNA was re-ligated by ligation enzyme, with a reverse cross-link by proteinase K. Following this, we extracted and purified DNA to remove biotin from non-ligated fragment ends using T4 DNA polymerase. After end repair and ligation by Illumina paired-end sequencing adapters, we generated a Hi-C library with insert size of 350 bp, where biotin-labeled DNA fragments were isolated using streptavidin beads and enriched by PCR amplification. The Hi-C library was sequenced on an Illumina HiSeqX platform. High-quality sequencing data were mapped to genome using the Burrows–Wheeler-alignment tool (BWA) v0.7.8[64], to identify the uniquely mapped reads. Based on the uniquely mapped Hi-C reads, we carried out pseudo-chromosome clustering and scaffolded the contigs into pseudo-chromosomes using ALLHIC v0.9.8[65] (https://github.com/tangerzhang/ALLHiC) (for the two autotetraploids, *S. rubusta*, *D. turbinatus* and *V. mangachapoi*) and Lachesis v201701[66] (for *H. chinensis*, and *P. chinensis*).

We assessed completeness of genome assembles using CEGMA (Core Eukaryotic Genes Mapping Approach) v2.5[67] and BUSCO (Benchmarking Universal Single-Copy Orthologs) v5.1.2[68]. The quality of genome assemblies was further tested by mapping Illumina paired-end reads to the genome assemblies using BWA. For each genome assembly, we performed synteny analysis using MCscanX[69], to identify syntenic blocks between pseudo-chromosomes.

To explain the difference between the estimated and assembled genome sizes in the two autotetraploid species (*H. hainanensis* and *H. reticulata*), we quantified the length of overlapping regions due to high homology according to coverage depth. Illumina paired-end reads for *k*-mer analysis were mapped to the assembled genomes using BWA-MEM (https://github.com/bwa-mem2/bwa-mem2), and coverage depth was calculated with a sliding window size of 10 Kb across all pseudo-chromosomes. The status of overlaps was determined by the ratio of coverage depth of a window to overall coverage depth in each genome, and regions with ratios of 1.5–2.5, 2.5–3.5 and larger than 3.5 were considered as overlapping of two, three and four sets of homologous chromosomes, respectively. The total overlapping regions for each genome were estimated to be the sum of overlapping regions of two sets chromosomes, two times of overlapping regions of three sets chromosomes and three times of overlapping regions of four sets chromosomes.

### Genome annotation

For each species, genome annotation comprised annotation of repeat sequences, non-coding RNA identification and gene prediction and annotation. Repeat sequences include transposable elements (TEs) and tandem repeats. For TE identification, a de novo repeat library was built first using RepeatModeler v1.0.5[70], LTR_FINDER v1.0.7[71] and RepeatScout v1.0.5[72], with corrections by the predication using RepeatMasker v4.0.5[73]. In addition, TEs were also identified by searching against the Repbase TE library (http://www.girinst.org/repbase) using RepeatMasker and RepeatProteinMask v4.0.5[73]. Tandem repeats were determined using Tandem Repeats Finder (TRF) v4.07b[74].

For non-coding RNA annotation, tRNAscan-SE v1.4[75] was used to identify tRNAs, and rRNAs were detected by aligning with the rRNAs of *Arabidopsis thaliana*, *Oryza sativa* and *Populus trichocarpa* using BlastN, with a threshold of $e < 1 \times 10^{-10}$. The miRNAs and snRNAs were determined by searching against the Rfam database (release 9.1) using INFERNAL v1.1.3[76].

To obtain high-quality annotation of protein-coding genes, we predicted gene models using multiple methods, including transcriptome-based predictions, de novo predictions, and homology-based predictions. In transcriptome-based predictions, RNA sequencing data from 12 tissues of *H. hainanensis* (with both PacBio Iso-seq and Illumina RNA-seq data), 3 tissues of *H. reticulata* (with only Iso-seq data) and 1–4 tissues of the other five species (with only RNA-seq data) (Supplementary Table 21) were mapped to their respective assembled genomes using PASA v2.3.3[77] (for Iso-seq data) and Tophat v2.0.13[78] (for RNA-seq data), and gene models were initially predicted using PASA v2.3.3[77] and Cufflinks v2.1.1[79]. Then, the PASA-assembled transcripts were used as training sets for ab initio gene prediction, using Augustus v3.0.2[80], Genscan v1.0[81], GlimmerHMM v3.0.2[82], Geneid v1.4[83] and SNAP v11-29-2013[84]. Moreover, protein sequences from six plant genomes (*A. thaliana*, *Coffea canephora*, *Populus trichocarpa*, *Camellia sinensis*, *Eucalyptus grandis*, *Prunus persica*) were used for homology-based predictions. The query sequences were searched in the assembled genomes of three *Hopea* species, using TblastN setting a criterion of $e < 1 \times 10^{-5}$, and protein sequences of nine genomes (the six genomes plus three *Hopea* genomes) were used for the other four species. BLAST hits were conjoined using Solar v0.9.6[85], and gene structure was predicted using GeneWise v2.2.0[86]. After that, EvidenceModeler v1.1.1[87] was used to integrate evidence from ab initio gene prediction, transcripts and homologous protein evidence for generating a non-redundant set of protein-coding genes. BUSCO assessment was used to evaluate the quality of gene annotation for each genome.

Functions of the protein-coding genes were annotated using BlastP (using a threshold of $e < 1 \times 10^{-5}$) to search against the databases of NCBI NR and SwissProt. Protein domains were detected using InterProScan v4.7[88] and HMMER v3.1b2[89] by searching against InterPro v29.0[90] and Pfam databases[91]. Gene Ontology (GO) terms for each protein-coding gene were achieved based on the corresponding description from InterPro and/or Pfam. Moreover, protein-coding genes were also mapped to the pathways in Kyoto encyclopedia of genes and genomes (KEGG)[92].

Alleles in the two autotetraploid genomes were identified by the detection of inter-haplotype syntenic blocks using MCScanX. Sequence similarities were checked among alleles using reciprocal BLAST (with a threshold of $e < 1 \times 10^{-7}$), and all genes were then classified into four types (genes with four, three and two alleles and those with only one allele).

### Phylogenomic analysis

To estimate the phylogenetic relationships of Dipterocarpoideae species and the divergence time between Dipterocarpaceae and its closest family Malvaceae, we first carried out gene family clustering. We downloaded the genomes and annotation information of 12 other dipterocarp species from Tian et al. [13], two Malvaceae species[93,94], and six species from other plant taxa (Supplementary Table 9) and formed a species pool containing these 25 species. For either autotetraploid species, we incorporated genes from the longest monoploid genome with all one-allele genes from other monoploids in the phylogenomic analysis, and this set of genes was also used in the following analyses. The gene models with open reading frames (ORFs) shorter than 30 amino acids were removed, and only the longest transcript was chosen to represent a gene when multiple transcripts existed. Gene family clustering based on the 25 plant genomes was performed using OrthoMCL v2.0.9[95], with a criterion of $e < 1 \times 10^{-5}$ in 'all-versus-all' BlastP.

We then estimated the phylogenetic relationships among these species using the single-copy orthologs identified by gene family clustering. Corresponding coding sequences of the single-copy orthologs from the 25 species were aligned according to protein sequences using MUSCLE v3.8.31[96], and codon position two of aligned coding sequences were chosen and concatenated to construct maximum likelihood tree using RAxML v8.0.19[97] setting a general time reversible substitution model with a gamma distribution and 100 bootstrap replicates (available at https://github.com/ Molecology/Dipterocarpoideae-genome/Supplementary_Code1.RAxML.sh). We used MCMC tree program of PAML v4.9[98] with correlated molecular clock and JC69 model to estimate divergence time, setting 1,000,000 Markov chain Monte Carlo iterations and a burn-in of 100,000 iterations. The MCMC trees were constructed using different sets of calibrations and their combination. The first set included three calibrations listed in Bansal et al.[17] (the divergence time among *Vatica* species (54 MYA), that among *Shorea* species (54 MYA), and that among *Dipterocarpus* species (68.5 MYA)), and the second set comprised two calibrations in Ng et al. [15] (the divergence time between *Theobroma cacao* and *Gossypium raimondii* (55.8 MYA), and that between *Shorea* and *Dryobalanops* (34 MYA)). We also constrained the root age using the age of Angiospermae crown-group (167–199 MYA)[25], and calibrated the crown age of Malvales as 113 MYA[26]. For the full list of parameter settings, please refer to the code (https://github.com/Molecology/Dipterocarpoideae-genome/Supplementary_Code2. mcmctree).

### Whole genome duplication analysis

To test whether the genomes of our studied species experienced whole genome duplication (WGD), homologous gene pairs within each assembled genome and those between the assembled genome were identified using MCScanX, with a threshold of $e < 1 \times 10^{-5}$ in 'all-versus-all' BlastP. Synonymous substitutions per synonymous sites (*Ks*) were calculated for each gene pair using the KaKs_Calculator v2[99], and the frequency distribution of syntenic blocks across different *Ks* values was used to reveal the potential speciation or WGD events (https://github.com/ Molecology/Dipterocarpoideae-genome/Supplementary_Code3.KaKs_Calculator). Moreover, to examine if a WGD was a polyploidization event, a quota test was conducted to compare the homologous regions in each assembled genome with those in the genome of *Vitis vinifera* (which has only experienced WGT-γ event)[100], using QUOTA-ALIGN script[101] (https://github.com/ Molecology/Dipterocarpoideae-genome/Supplementary_Code4.QUOTA-ALIGN).

### Comparative genomics

To explore the molecular footprints relevant to the adaptation of our studied species, we conducted the comparison between the genomes of the five diploid Dipterocarpoideae species with those of five temperate tree species to screen the genes with signal of positive selection and expanded/contracted gene families. Based on the temperate tree comparison, we may find the genomic signals of adaptation of Dipterocarpoideae species to tropical habitats.

Gene family clustering and construction of maximum likelihood tree were performed following the methods outlined above. Branch-site model was chosen to detect the single-copy orthologs experiencing significant positive selection using PAML (https://github.com/HLTCHKUST/PAML). Gene family expansion and contraction were examined using the Computational Analysis of gene Family Evolution (CAFE) v2.1[102], employing a stochastic birth-and-death process to model the evolution of gene family sizes over a known phylogeny. The birth-and-death parameter (λ) was estimated using 10,000 Monte Carlo random samples. The family-wise method was used to statistically test if a gene family experienced expansion/contraction (https://github.com/hahnlab/CAFE). We then assigned positively selected

genes++ and genes in expanded/contracted families into different KEGG pathways and GO terms according to their functional annotations.

Genes in LRR-RLK (leucine-rich repeat receptor-like kinase) and NBS (nucleotide binding site) gene families were identified using RGAugury pipeline[103] (https://github.com/ Molecology/Dipterocarpoideae-genome/Supplementary_Code5.RGAugury.sh). For LRR-RLKs, after searching against database RGAdb by BlastP with a threshold of $e < 1 \times 10^{-5}$, candidate proteins were determined and their domains (LRR and RLK) were predicted using InterProScan to search against databases Pfam and SMART[104]. NBS proteins were identified and categorized by checking the NBS domain, coiled-coil/toll interleukin receptor domain in the N end, and LRR domain in the C end. Coiled-coil domain was predicted by paircoil2[105], and HMMsearch v3.0[106] was used to predict the other domains based on the HMM model (PF00931) downloaded from Pfam database[60,61].

### In vitro functional characterization of key positively selected genes

The full length of the ORFs of two positively selected genes (an UDP-sugar pyrophosphorylase (*USP*) and a pyridoxine 4-dehydrogenase (*PLR1*)) from the genome of *H. chinensis* (Supplementary Table 12) were confirmed by reverse transcription PCRs and Sanger sequencing. These two genes were then artificially synthesized and cloned into pET-22b(+) plasmids (MilliporeSigma), expressed in *E. coli* strains BL21 (DE3). Following expression, the cells were harvested and lysed, and the lysates were then centrifuged at 13,523 g for 10 min to pellet the cell debris. The produced recombinant proteins were purified (purity >90%) using modified nickel-nitrilotriacetic acid agarose (Thermo Fisher Scientific).

For enzyme activity assays of the two genes, we used the reaction system (500 µl) mainly composed of 50 mM MOPS-KOH (pH 7.4)/ citrate (pH 6.5) buffer, 0.01% (g/ml) bovine serum albumin (BSA), 0.4–1.0 mM substrate(s) (Supplementary Table 12), and 10 µl of purified recombinant protein (0.2 mg ml$^{-1}$). After 5 min of incubation and 30 min of reaction at 35 °C, the reaction products were collected by extraction using diethyl ether and were analyzed using LC−MS. For each gene, enzyme activity assay was repeated three times. In addition, three replications of negative controls (only adding the substrates and BSA into buffer) were conducted for each gene.

### Resequencing and variant calling

To characterize the population genomics of the two autotetraploid species, we collected fresh leaves from a total of 30 (for *H. hainanensis*) and 32 (for *H. reticulata*) wild trees for resequencing. All of these individuals were found in Hainan, China (Supplementary Table 16), with a minimum sampling interval of 50 m between the nearest conspecific trees. Though scattered *H. hainanensis* trees may exist in northern Vietnam, precise locations of wild populations are only available on Hainan Island, and 23, 1, 1, and 5 trees were sampled from Bawangling, Wuzhi Mountain, Danzhou and Jianfengling, respectively. All trees of *H. reticulata* were sampled from Ganshiling, the only location where wild *H. reticulata* populations have been recorded[107].

For each tree sampled, an Illumina pair-end library with insert size of 350 bp was prepared and sequenced using an Illumina HiSeqX platform. The quality control of raw data was performed using fastp v0.19.7[108], and raw pair-end reads were trimmed by removing adaptors and low-quality reads (the proportion of unknown bases exceeds 10% or the proportion of low-quality bases (quality score ≤5) is larger than 50%). We mapped the filtered reads to the longest monoploid genome of each species to generate BAM files using BWA-MEM with default parameters, and SAMTOOLS v1.3[109] was used to remove duplicated reads (https://github.com/Molecology/Dipterocarpoideae-genome/Supplementary_ Code6.mapping.varcall.sh). To meet the assumptions of diploid genomes in population demography analyses using SMC++[36] and PSMC[35], we adopted the longest two monoploid

genomes of the autoploid species as mapping references (see below for the details).

The Realigner Target Creator program in the Genome Analysis Toolkit (GATK) v4.0.4.0[110] (https://github.com/broadinstitute/gatk). Variants were then detected using HaplotypeCaller setting the default parameters. We filtered out the variants within this quality range: OD < 2.0, QUAL < 30.0, SOR > 3.0, FS > 60.0, MQ < 40.0, MOR-ankSum < −12.5 and ReadPosRankSum <−8.0. Furthermore, single nucleotide polymorphism loci (SNPs) were filtered by VCFtools with minimum depth ≥13, minor allele frequency (MAF) ≥ 0.05 and missing rate ≤0.1 (https://github.com/Molecology/Dipterocarpoideae-genome/Supplementary_Code6.mapping. varcall.sh). SNPs were annotated using ANNOVAR[111] on the basis of genome annotation, and were assigned into intergenic and genic regions (i.e., synonymous, nonsynonymous, intronic variants). In addition, to test if our sampled trees are auto-tetraploid, $k$-mer analysis using Jellyfish was carried out using the four trees sampled with the highest calling depth per species (see Supplementary Table 17).

## Population demography

To infer the demographic history for the two species, we estimated variation in effective population sizes across time based on both Sequentially Markovian Coalescent (SMC) and Pairwise Sequentially Markovian Coalescent (PSMC) models. We first determined generation time ($g$) and mutation rate ($\mu$) for both species. Because the planted trees began to flower and produce seeds at $c.$ 10 years old (according to the records from local forestry stations) and a previous study has reported that trees of a congeneric species $H. chinensis$ become mature at about 12 years old[112] (which provide the minimum estimation of generation time for these two species), we set multiple generation time of 15, 20 and 30 years for both species. We then assessed $\mu$ using the formula $\mu = (D \times g)/2\,T$, where $D$ is the observed frequency of pairwise site differences between genomes of the two species and $T$ is the estimated divergence time that had been evaluated in the phylogenomic analyses outlined above. $D$ was calculated manually based on the four-fold degenerate sites extracted from the aligned single copy orthologs between any two of the three $Hopea$ genomes (see Supplementary Table 18), which were identified using InParanoid[113].

The challenge of demographic analysis for autopolyploid species is mainly caused by the lack of generalized methodologies[53,114]. To date, the algorithms and software (e.g. SMC++[36] and PSMC[35]) developed for effective population size analysis are restricted to diploid analysis. To meet the requirements of these analyses, we made an attempt to treat the autotetraploid species as diploids by using the longest two monoploid genomes as the reference genome.

We estimated the recent-past effective population sizes ($N_e$) of $H. hainanensis$ and $H. reticulata$ separately using the coalescent-based inferences in SMC++ v 1.15.5[36]. The VCF files, including 30 $H. hainanensis$ samples and 32 $H. reticulata$ samples, were converted to the SMC++ input files using the 'vcf2smc' command. The estimate was run with generation times of 15, 20 and 30 separately and the corresponding estimated mutation rates (Supplementary Table 18). Codes are available at https://github.com/Molecology/Dipterocarpoideae-genome/Supplementary _Code7.SMC.sh.

In addition, we inferred historical changes in $N_e$ of $H. hainanensis$ and $H. reticulata$ using PSMC v0.6.4-r49[35]. For each of 30 $H. hainanensis$ samples and 32 $H. reticulata$ samples, we performed the analysis with generation times of 15, 20 and 30 and the corresponding estimated mutation rates separately (Supplementary Table 18). The whole-genome consensus sequence was generated by SAMTOOLS v1.16, bcftools, and vcfutils.pl, setting the minimum read depth to 10, maximum read depth to 100, and minimum mapping quality 20. We ran PSMC with parameters (psmc -N30 -t15 -r5 -p 4+25*2+4+ 6) (https://github.com/Molecology/Dipterocarpoideae-genome/Supplementary_Code8. PSMC.sh).

## Assessment of genetic diversity

To evaluate genome-wide genetic diversity in our focal populations of $H. hainanensis$ and $H. reticulata$ based on SNP data, the population genetic diversity index $\pi$ was calculated directly from VCF files setting the sliding window size of 10 Kb using VCFtools v0.1.14[115].

## Evaluation of genetic load

To explore the potential genetic load in the two species, we detected the derived deleterious mutations (DDMs) and the derived major-effect mutations (DMEMs) using SNP data. We first chosen non-synonymous SNPs based on genome annotation, and deleterious mutations (DMs) were identified using PROVEAN v1.1.5[116] (https://github.com/Molecology/Dipterocarpoideae-genome/Supplementary_Code9.proven.sh) and SIFT4G v2.0.0[117] (https://github.com/ Molecology/Dipterocarpoideae-genome/Supplementary_Code10.SIFT.sh) using the default settings. Nonsynonymous SNPs with scores < −2.5 in PROVEAN and <0.05 in SIFT4G were assumed to be deleterious, and we only retained deleterious mutations supported by both analyses. Moreover, we searched through each genome to demark mutations that caused start loss, stop gain, stop loss and changes in splice sites with the help of annotation information (i.e., major-effect mutations (MEMs)). To further select the DMs and MEMs that were derived after speciation, we identified ancestral alleles of segregating sites, using the genomes of $H. reticulata$ and $H. chinensis$ as the outgroup for $H. hainanensis$ and those of $H. hainanensis$ and $H. chinensis$ as the outgroup for $H. reticulata$. We only called the polarity of variants when both species in the outgroups had identical homozygous states. Then, DDMs and DMEMs were determined according to the identification of derived alleles (https://github.com/Molecology/Dipterocarpoideae-genome/Supplementary_Code11.derived), and their distributions within studied populations were described.

## Genetic structure

To reveal the genetic structure within each species, we carried out genetic clustering, phylogenetic analysis and principal component analysis (PCA). We used PLINK v1.07[118] to remove SNPs located in coding regions and those with linkage disequilibrium coefficient ($r^2$) larger than 0.1. Filtered SNP data were used for the following analysis. Genetic clustering was performed using ADMIXTURE v1.23[119]. The optimal number of clusters ($K$) was determined by running ADMIXTURE from $K = 1$ to $K = 8$ and choosing the $K$ value with the minimum cross-validation error. When assigning the sampled trees into different genetic clusters, the standard errors of parameters were estimated using bootstrapping (bootstrap = 200). A neighbor-joining (NJ) tree was constructed to uncover the phylogentic relastionships among the individuals sampled using TreeBeST v1.9.2[120], setting 1000 bootstraps. Figtree v.1.4 was used to visualize the NJ tree. PCA analysis was conducted using GCTA v1.24.2[121] to calculate eigenvectors and eigenvalues, and the top two principal components were extracted for the assignment of our sampled trees.

## Estimation of inbreeding

To assess the inbreeding level of $H. hainanensis$ and $H. reticulata$ populations, based on SNP data, we estimated the runs of homozygosity (ROH) for each sampled tree using PLINK with the slide window size of 20 SNPs (https://github.com/Molecology/Dipterocarpoideae-genome/ Supplementary _Code12.ROH.sh). We calculated the frequency distribution and total length of ROHs that were longer than 5 Kb and have at least 10 SNPs.

## Reporting summary

Further information on research design is available in the Nature Portfolio Reporting Summary linked to this article.

## Data availability

The whole-genome sequences (WGS) of our seven studied Dipterocarpoideae species and the 62 individuals used in the population genomic analysis have been deposited in the NCBI GenBank database, under the BioProject accession number PRJNA1056647 (https://www.ncbi.nlm.nih.gov/bioproject/PRJNA1056647). The annotation files of the seven Dipterocarpoideae species have been deposited in the Figshare (https://figshare.com/articles/dataset/_/24936075). The genome assemblies of other species used in our analyses were downloaded from the sources as below: CNGBdb: CNP0002104 (13 Dipterocarpoideae species); NCBI: PRJNA788082 (*Gossypium raimondii*); DDBJ: PRJDB8161 and PRJDB8182 (*Shorea leprosula*); EMBL, GenBank, DDBJ: CACC01000001–CACC01025912 (*Theobroma cacao*); http://ftp.ensemblgenomes.org/pub/plants/release-45/genbank/oryza_sativa (*Oryza sativa*); www.arabidopsis.org/cereon (*Arabidopsis thaliana*); iProX: PXD015597 (*Aquilaria sinensis*); NCBI: PRJNA212863 (*Amborella trichopoda*); ENA: PRJEB4958 (*Fraxinus excelsior*); ENA: PRJEB24056 (*Fagus sylvatica*); NCBI: PRJNA428013 (*Ostrya rehderiana*); ENA: PRJEB19898 (*Quercus robur*). Source data are provided with this paper.

## Code availability

The in-house analysis scripts have been deposited in Github (https://github.com/Molecology /Dipterocarpoideae-genome).

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

## Acknowledgements

We thank Yao-Bin Song, Shui-Xing Luo, Shou-Qian Nong, Qi-Chong Zhu, and Qian-Ya Li for their kind helps in field sampling and analysis, and Xiu-Guang Mao, and Pan-Yu Hua for constructive suggestions in data analysis. We especially thank Chen Hui and Xishuangbanna Station for Tropical Rainforest Ecosystem Studies, XTBG, CAS, for supplying the photograph showing the Asian rainforest. This work is supported by the National Key Research & Development Program (2016YFC0503102), NSFC grant 32261123001 (X.-Y.C.) and "Ecology +" program of East

China Normal University. Simon Segar acknowledges departmental funding from Harper Adams University.

## Author contributions

Conceptualization: R.W., X.Y.C.; Methodology: R.W., Y.Y.Z., C.N.L., S.S., Y.T.J., K.J.Z., K.J., G.W., J.Cai, J.Cheng, S.G.C., X.D.H., W.W.X., Y.T.J., Z.X.Z., L.K., L.F.C., J.Y.D., F.K.D., Y.Y.L., R.G.Z.; Investigation: C.N.L., R.W., X.H.H., X.T., Y.Y.D., L.T., D.H.L., L.L., Y.Y., S.C., Z.S.W.; Funding acquisition: X.Y.C., R.G.Z.; Project administration: X.Y.C., R.W.; Writing: all authors contribute to writing the manuscript.

## Competing interests

The authors declare no competing interests.

## Additional information

[1]Zhejiang Tiantong Forest Ecosystem National Observation and Research Station, Shanghai Key Lab for Urban Ecological Processes and Eco-Restoration, School of Ecological and Environmental Sciences, East China Normal University, Shanghai, China. [2]Agriculture & Environment Department, Harper Adams University, Newport, United Kingdom. [3]Novogene Bioinformatics Institute, Beijing, China. [4]Shanghai Key Laboratory of Plant Functional Genomics and Resources, Shanghai Chenshan Botanical Garden, Shanghai, China. [5]CAS Key Laboratory of Tropical Forest Ecology, Xishuangbanna Tropical Botanical Garden, Chinese Academy of Sciences, Mengla, Yunnan, China. [6]School of Ecology and Environment, Northwestern Polytechnical University, Xi'an 710072, China. [7]School of Biology, University of Leeds, Leeds, United Kingdom. [8]School of Ecology and Nature Conservation, Beijing Forestry University, Beijing, China. [9]Guangxi Institute of Botany, Guangxi Zhuang Autonomous Region and the Chinese Academy of Sciences, Guilin, China. [10]College of Ecology and Environment, Hainan University, Haikou, China. [11]Key Laboratory of Genetics and Germplasm Innovation of Tropical Special Forest Trees and Ornamental Plants, Hainan University, Haikou, China. [12]Ecology and Nature Conservation Institute, Chinese Academy of Forestry, Beijing, China. [13]Key Laboratory of the Ministry of Education for Coastal and Wetland Ecosystems, College of the Environment and Ecology, Xiamen University, Xiamen, Fujian, China. [14]Shanghai Engineering Research Center of Sustainable Plant Innovation, Shanghai, China. [15]Shanghai Institute of Pollution Control and Ecological Security, Shanghai, China. [16]Institute of Eco-Chongming, Shanghai, China. [17]These authors contributed equally: Rong Wang, Chao-Nan Liu, Simon T. Segar, Yu-Ting Jiang, Kai-Jian Zhang. ✉e-mail: rwang@des.ecnu.edu.cn; zhangyuanye@xmu.edu.cn; xychen@des.ecnu.edu.cn

