## [Peer Review File · Nature Communications]

REVIEWER COMMENTS

Reviewer #1 (Remarks to the Author):

Dipterocarp trees are dominant in Southeast Asian tropical forests and important for environments, forestry, and medicine. Recently, a few genomic studies reactivated long-standing debates about its evolutionary history centered on its Gondwanan vicariance hypothesis. The authors reported chromosome-level assemblies of seven dipterocarp tree species. Two of them are autotetraploid tree species, which can be a unique resource to study plant evolution in general. However, a major revision would be necessary to place the study in the context of previous research.

1. The genome assemblies of dipterocarps, i.e., *Shorea leprosula* (Ng et al. *Commun Biol* 7:1166, 2021) with transcriptome and resequencing of other dipterocarps, *Dipterocarpus turbinatus* and *Hopea hainanensis* (Wang et al. *Plant Biotechnol* 20:538, 2022, Epub 2021 Dec), were already reported some time ago. Many results of this manuscript, such as ancient polyploidy and time estimation, were reported. The same species, *Hopea hainanensis*, was sequenced before. The current manuscript would give a false impression of novelty. This is not scientifically proper. Rather, I would suggest that the comparison with previous studies and the introduction explaining unsolved issues would highlight the importance of this new study as explained below. Their report should be fully integrated fully in the introduction and discussion. Furthermore, Tian et al. (*Plant Communications* 2022, 100464, 2022) reported 13 dipterocarp genomes in October 2022. I am not sure if this is before or after the submission of this manuscript and may depend on the journal policy, but integrating it would further strengthen this manuscript.

2. Novelty in autopolyploidy

Wang et al. reported the assembly of the same species *Hopea hainanensis* but they did not mention autotetraploidy at all and treated it as a normal diploid species. This manuscript provided chromosome count (Supplementary Figure 1) and careful k-mer analysis to provide strong evidence of polyploidy. Furthermore, they seem to have successfully provided phased four assemblies, although further quality checks in this aspect should be reported. In general, the genome assembly of autopolyploid species remains a major challenge in genomics. For example, the assembly of autotetraploid *Arabidopsis arenosa* is difficult, and instead, self-fertilizing allotetraploid *A. suecica* derived from *A. arenosa* was sequenced instead (Burns et al. *Nature Ecology and Evolution*, 10:1367, 2021; Jiang et al. *Nature Ecology and Evolution*, 10:1382, 2021). Supposing the quality of the phasing is quantified, the new assembly of *H. hainanensis* would provide interesting insights into the autopolyploid genome.

To support the tetraploidy, please add the empirical data of genome size measured by flow cytometry. Ng et al. (*Plant Ecology and Diversity* 9:437, 2016) explained a standard method of dipterocarps.

The methods to analyze the demographic analysis of the autotetraploid need to be thoroughly revised. As far as reading the Methods, software developed for diploid species is applied to the data. However, many assumptions are not filled. Please refer to the population genetic analysis of autotetraploid *A. arenosa* (Monnahan et al. *Nature Ecol Evol* 3: 457, 2019) and theoretical studies cited there (particularly Otto and Whitton, *Ann Rev Genet* 34:401, 2000). Here I list a few examples below but the authors should investigate the effect of autopolyploidy thoroughly. As far as I understand, SMC++ and PSMC assume diploid inheritance like a human being. Intuitively, it may be usable if you randomly choose two haplotypes, but a thorough theoretical basis should be explained if it would be included in the revised manuscript. Furthermore, the generation time of 10 and 15 years (line 741) are very likely minimum estimates, because the trees will produce seeds for a long until a gap may be formed in the forest or other reasons. Relative time estimation of N_e would be interesting but the usage of absolute time to compare the demography with glacial cycles should be omitted. In line 239, a comparison with cherry-picking several tree species does not make sense and so should be omitted. In line 221, Tajima's D, CLR, and iHS are not designed for autopolyploid species, although they can work for allopolyploid species.

3. Ancient polyploidy was already reported

Ng et al. (2021) and Wang et al. (2022) reported a genome duplication event preceding the divergence of Dipterocarpoideae species. The authors did find it out, but the original paper should

be cited. Otherwise, it will give a false impression of novelty.

4. Time estimation

There has been a long dispute on when Dipterocarpaceae/Dipterocarpoideae originated (see references cited in the literature below). Traditionally, the Gondwanan vicariance hypothesis based on broad geographic distribution proposed an old origin of >120 million years ago. However, the estimates are variable among phylogenetic studies, and most importantly, the Gondwanan hypothesis contradicts the general time scale of angiosperm evolution. Ng et al. conducted a detailed analysis using the genome data and fossil calibration and reported much younger divergence. In 2022, new microfossils of dipterocarps were reported, which can push the divergence time even younger (Bansal et al. *Science* 375:455, 2022). The issue is currently unsolved and can be potentially a major contribution to this manuscript. The authors cited Bansal et al. 2022 but did not discuss the dispute at all.

I should note that the methodologies the author used are primitive. The authors should examine the complexity and many assumptions in the time estimation. For calibration, authors took values from the TimeTree webpage, but this does not seem sophisticated. Particularly, little data on dipterocarp and its relatives were available. For example, when "Hopea" and *Theobroma cacao* was filled in on its webpage, the outcome was 80 MYA (CI 38.7-85.0 MYA). In line 139, the authors reported c. 68.2 MYA without providing a confidence interval. This illustrates the insufficient quality of the current analysis. At the moment, it would be wise to use two different ways of fossil calibration, the

new ones by Bansal et al., and traditional ones used by Ng et al. Otherwise, it may be better to remove time estimation from this manuscript.

5. Cherry-picking of genes under selection

Although the authors said "GO enrichment analysis" (line 174), I could not find any statistics. Rather, it seems that the authors arbitrarily picked up genes relevant to UV, oxidative stress, and whatever they want to discuss. The authors can find standard GO enrichment analysis in many genome papers. "Temperate tree comparison" or "Malvaceae comparison" does not seem to make sense because dipterocarp species and Malvaceae species would form a single clade, respectively, and after all, there is only a single comparison. If this should be retained, the validity of the methods should be explained in detail. Moreover, their hypothesis in line 62 "At the emergent layer, these species are challenged with intensive ultraviolet (UV) radiation and associated oxidative stress. Adaptation to such conditions is thus critical for these species to occupy this niche" does not seem to be well defended. For example, tree species living in forest gaps may be subjected to stronger stress, while the seedlings of canopy species may grow understory. For example, Ng et al. 2021 tested the importance of drought stress using the retained duplicated genes, and here the authors could test it using the signature of selection on each gene. Even if it is supported or not, the test can be interesting.

6. Data availability section should state more specifically the availability of the assemblies and annotation files.

Minor issues

line 184, which is involved

Reviewer #2 (Remarks to the Author):

Comments for authors

NCOMMS-22-41572-T: Liu et al. Life up high: the adaptations and demography of the Asian rainforest titans, Dipterocarpoideae.

In this manuscript, Liu et al. report seven dipterocarps genomes assembled to the chromosome level; 5 were diploid (*Shorea robusta*, *Dipterocarpus turbinatus*, *Hopea chinensis*, *Vatica mangachapoi* & *Parashorea chinensis*) and 2 were determined as autotetraploid (*Hopea hainanensis* & *H. reticulata*). The dipterocarp genomes shows evidence of whole-genome duplication (WGD) in its evolutionary past, preceding their lineage diversification at Oligocene and

two lineage-specific autotetraploidization. Comparative genomics using 5 of the diploid dipterocarp species with 5 temperate species and 5 Malvaceae species, they found positively selected genes that relate to the adaptation above emergent canopy layer. Using resequencing samples of ~30 individuals each from the two *Hopea* species (autotetraploid) they found expansion of effective population size during last glacial period but suffers shape decline during post glacial period due to rising sea levels.

The assembled genomes quality are good, (except for the 2 autotetraploids – see below) but interesting observation was made by the authors on these genomes, e.g. regarding the involvement of DNA repair in contributing to the adaptation to the UV-rich above emergent canopy layer.

Several concerns on this manuscript:

This manuscript adds important genome resources to the currently growing list of sequenced dipterocarp genomes. Here are the list:

1. Ng et al. 2021 Communications Biology – *Shorea leprosula*
2. Wang et al. 2022 Plant Biotechnology Journal – *Dipterocarpus turbinatus* & *Hopea hainanensis*
3. Tang et al. 2022 DNA Research – *Vatica mangachapoi*
4. Tian et al. 2022 Plant Communications – *Dipterocarpus alatus*, *D. gracilis*, *D. intricatus*, *D. zeylanicus*, *Hopea mollissima*, *H. odorata*, *Parashorea chinensis*, *Shorea roxburghii*, *S. leprosula*, *S. henryana*, *S. xishuangbannaensis*, *Vatica odorata* & *Vatica rassak*

I would strongly suggest that the authors to review and include them in the introduction and discussion. If not too much, include them in their analysis to get better conclusion particularly on the phylogenomics and genome evolution.

In addition, four out of the seven species presented in the present manuscript have also been published. Particularly, study by Wang et al 2022 (above) reported that *Hopea hainanensis* as diploid and their sequenced assembly at the chromosome scale does not showed any autotetraploidization genome size as reported in this study. However, in the present study, the authors justified their finding of tetraploidization in *H. hainanensis* by karyotyping (but did not discuss what Wang et al. has reported). A better resolution for Supplementary Fig 1q (FISH probe 18S rDNA) would really help. Besides, why similar approach was not done to confirm the autotetraploidization of *H. reticulata*? In addition, why didn't consider estimating the genome content and ploidy of the two autotetraploid *Hopea* species by measuring the C-value (amount of DNA in the unreplicated gametic nucleus) using flow cytometry?

The known obstacle in the de novo assembly of an autopolyploid genome is distinguishing and separating very similar haplotypes, which are often assembled as highly fragmented sequences until recently with the use of chromosome-scale and haplotype-resolve genome assembly approach done in tetraploid potato (Sun et al. 2022 Nat Genet). Will the selection of the longest monoploid with most annotated protein-coding genes as reported in this study meaningful? What were the proportions of annotated protein-coding genes for the other monoploids?

The authors also suggest that lineage-specific polyploidization after speciation but from result clearly such event does not occur on *H. chinensis* which are closely related to *H. reticulata*. Further elaboration and explanation are necessary to support this statement.

Specifically:

Page 5, lines 78-81: The sentence here was so confusing to read. Please revise.

Page 5, lines 94-95: The sentence was very confusing.

Supplementary Fig 1a does not show similar kmer pattern as to Supplementary Fig 1h-k. Please clarify this.

Page 8, lines 154-158: Why the two tetraploid species were not included in the comparative

genomics analysis? Plus, there were several others sequenced and published dipterocarp genomes, it would be much more interesting to know if they share the similar genomic signature of adaptation.

Page 10, line 205: Only 3 expanded gene families were observed in Suppl Table 12 not 4 expanded gene families.

Page 11, lines 217-219: statements are very speculative, please provide support or examples.

Page 11, line 237: ...relatively similar genetic diversity...

Page 12, line 260: 'endangerment' change to "decrease in population sizes"

Supplementary Fig 12: Description was not found in the main text

Page 13, line 281: "extensive gene flow" – do you know what are the pollinators within the distribution areas? As in the aseasonal tropics, tiny thrips and medium size stingless bees (*Trigona* spp) were reported but they do not strongly suggest extensive gene flow, if any, it will be at the lower proportion. Perhaps, this fundamental information is critical to justify conservation and management of the species.

Supplementary Fig 14: no indication of a-d

Page 14, lines 295-296: "...probably resulting from their tetraploidy." Could you elaborate more on how this was resulted from tetraploidy?

Page 14, lines 298-304: This should be best discussed along with previous findings on the other published dipterocarp genomes.

Page 14, lines 309-310: The two cited references only mentioned DNA repair but what about antioxidation?

Page 15, lines 310-315: This is a rather speculative statement. It would be interesting to consider checking the previously published genomes for confirmation.

Page 15, lines 319-321: This is speculative.

Page 15, lines 325-326: extensive gene flow? insect pollinators? Do you know what insect pollinators?

RESPONSE TO REVIEWERS' COMMENTS

Reviewer #1 (Remarks to the Author):

Dipterocarp trees are dominant in Southeast Asian tropical forests and important for environments, forestry, and medicine. Recently, a few genomic studies reactivated long-standing debates about its evolutionary history centered on its Gondwanan vicariance hypothesis. The authors reported chromosome-level assemblies of seven dipterocarp tree species. Two of them are autotetraploid tree species, which can be a unique resource to study plant evolution in general. However, a major revision would be necessary to place the study in the context of previous research.

>>R: Many thanks for your positive comments and your constructive suggestions. Following your suggestions, we have revised the manuscript thoroughly; please find the point-to-point reply below. We hope you will find that novelty and the significance of this study have been substantially strengthened.

1. The genome assemblies of dipterocarps

1. The genome assemblies of dipterocarps, i.e., *Shorea leprosula* (Ng et al. *Commun Biol* 7:1166, 2021) with transcriptome and resequencing of other dipterocarps, *Dipterocarpus turbinatus* and *Hopea hainanensis* (Wang et al. *Plant Biotechnol* 20:538, 2022, Epub 2021 Dec), were already reported some time ago. Many results of this manuscript, such as ancient polyploidy and time estimation, were reported. The same species, *Hopea hainanensis*, was sequenced before. The current manuscript would give a false impression of novelty. This is not scientifically proper. Rather, I would suggest that the comparison with previous studies and the introduction explaining unsolved issues would highlight the importance of this new study as explained below. Their report should be integrated fully in the introduction and discussion. Furthermore, Tian et al. (*Plant Communications* 2022, 100464, 2022) reported 13 dipterocarp genomes in October 2022. I am not sure if this is before or after the submission of this manuscript and may depend on the journal policy, but integrating it would further strengthen this manuscript.

>>**R:** Many thanks for your suggestions and the references. Indeed, we agree that it is important to acknowledge the previous discoveries. Following your suggestions, we have highlighted the contributions of Ng et al. 2021, Wang et al. 2022, and Tian et al. 2022 while introducing the debates surrounding the origin of the Dipterocarpaceae. We have rephrased the paragraphs in the introduction and discussion sections (The revision is highlighted in blue in the main text, please also see below).

In the introduction, we added:

Earlier studies proposed that Dipterocarpaceae originated in Western Gondwanaland in Early Cretaceous (*c.*120 million years ago (MYA)), before the separation of Africa and South America¹⁴. Recent evidence from pollen fossils¹⁵ and a phylogenomic study¹³ supported such a Western Gondwanaland origin, albeit with a slightly recent dating (*c.*102.9 MYA¹⁵ and 105.0 MYA¹³). These studies suggested that Dipterocarpaceae dispersed to India during the mid-Cretaceous and subsequent to Southeast Asia facilitated by India-Asia collision. However, genomic studies involving different sets of dipterocarp species revealed a much more recent divergence between Dipterocarpaceae and its closest family Malvaceae (*c.* 84 MYA¹⁶ and 86–98 MYA¹⁷). Therefore, there is still a controversy surrounding the evolution of these keystone rainforest species; resolving it will provide valuable insights into the origins of megathermal angiosperms.

In the discussion, we added:

Here, the phylogenetic history constructed using a total of 19 Dipterocarpoideae genomes agrees with the results from the early view¹⁴ and some recent phylogenomic/phylogenetic studies^{13, 15}, thus supporting the Western Gondwanaland origin of Dipterocarpaceae. Different calibrations yielded consistent results, indicating this estimation of divergent time is robust. This finding is largely consistent with the evolutionary history of angiosperms revealed by genomic data³⁵ and new fossil evidence³⁶, showing far earlier splits between major clades of angiosperms (occurring at the Jurassic and Lower Cretaceous) than traditionally expected.

Furthermore, as you suggested, we have integrated all published Dipterocarpaceae genomes, including those in Tian et al. 2022 in the revised phylogenomic analysis, and

thus form a dataset of 19 Dipterocarp genomes. These suggestions significantly improve the novelty and solidity of our study. Please refer to the response for this specific issue below.

2. Novelty in autopolyploidy

Wang et al. reported the assembly of the same species *Hopea hainanensis* but they did not mention autotetraploidy at all and treated it as a normal diploid species. This manuscript provided chromosome count (Supplementary Figure 1) and careful k-mer analysis to provide strong evidence of polyploidy. Furthermore, they seem to have successfully provided phased four assemblies, although further quality checks in this aspect should be reported. In general, the genome assembly of autopolyploid species remains a major challenge in genomics. For example, the assembly of autotetraploid *Arabidopsis arenosa* is difficult, and instead, self-fertilizing allotetraploid *A. suecica* derived from *A. arenosa* was sequenced instead (Burns et al. Nature Ecology and Evolution, 10:1367, 2021; Jiang et al. Nature Ecology and Evolution, 10:1382, 2021). Supposing the quality of the phasing is quantified, the new assembly of *H. hainanensis* would provide interesting insights into the autopolyploid genome.

To support the tetraploidy, please add the empirical data of genome size measured by flow cytometry. Ng et al. (Plant Ecology and Diversity 9;437, 2016) explained a standard method of dipterocarps.

>>R: Many thanks for acknowledging the significance of our tetraploid assembly and for the references. The autotetraploid assembly presented a major challenge until recently. Zhang et al. 2018 Nat Genet provided the first allele-aware assembly of sugarcane and corresponding methods were published by Zhang et al. 2019 Nat Plants. We followed their approach and maintained close communication with the authors since we started the project in 2019, and these efforts resulted in the assembly of comparable quality to Zhang et al. 2018 Nat Genet. We also provided evidence showing the quality of the two autotetraploid genomes (Lines 133-140, 144-147, and 150-154; Supplementary Tables 3-5). So far, the allele-aware genome assembly of autotetraploid

was still limited, in particular for wild species, and we hope that our work will inspire further involvement in studies of Dipterocarpaceae and wild tetraploid species.

Furthermore, we highly appreciate the methods you provided. We conducted flow cytometry analysis and estimated the C-values accordingly. The corresponding results have been added to Figure S1 and Table S1 (please also see below), providing robust evidence in support of tetraploidization. We have also revised the results section accordingly.

Sample	Genometric mean	CV error (%)	2C (pg)	Genome size (Mb)
Hopea chinensis	10647.2	9.66	0.63	0.62
Solanum lycopersicum	35604.3	5.03	2.12	
Hopea reticulata	15993.9	8.57	1.36	1.33
Solanum lycopersicum	25019.7	4.02	2.12	
Hopea hainanensis	846.9	6.78	1.66	1.62
Solanum lycopersicum	1082.3	3.16	2.12	

In the results, we added:

Moreover, by using *Solanum lycopersicum* (2C = 2.12) as a reference, the flow cytometry results revealed the C-values of *H. chinensis*, *H. reticulata* and *H. hainanensis* are 2C equal to 0.63, 1.36 and 1.66 (Supplementary Fig. 1j-l and

Supplementary Table 1). Compared to the diploid *H. chinensis*, the cytometry analysis identified a clear peak confirming the tetraploidy of these two species (Supplementary Fig. 1m-n).

In the methods, we added:

To further test whether *H. hainanensis* and *H. reticulata* are tetraploid species, we estimated their genome sizes and ploidy using flow cytometry. For these two species and *H. chinensis* (a diploid congener), we collected the fresh young leaves from the sampled trees used for the *de novo* genome assembly. We also sampled young leaves of *Solanum lycopersicum* (genome size: 2.07 Gb; $2C = 2.12 \text{ pg}^{53}$) as the reference for genome size estimation. We first estimated the genome sizes and $2C$ values of the three *Hopea* species by measuring the fluorescence of nuclei from the mixture of each *Hopea* species and *S. lycopersicum* and those from each single species. Then, we assessed the ploidy for *H. hainanensis* and *H. reticulata* based on their fluorescence of nuclei setting *H. chinensis* as the diploid reference. The nuclei suspension was prepared for each species following the methods described in Ng et al.⁵⁴. We measured the fluorescence from the nuclei suspension using a Sysmex CyFlow Cube6 flow cytometer, and data analysis was performed using FCSExpress 7plus.

The methods to analyze the demographic analysis of the autotetraploid need to be thoroughly revised. As far as reading the Methods, software developed for diploid species is applied to the data. However, many assumptions are not filled. Please refer to the population genetic analysis of autotetraploid *A. arenosa* (Monnahan et al. Nature Ecol Evol 3: 457, 2019) and theoretical studies cited there (particularly Otto and Whitton, Ann Rev Genet 34:401, 2000). Here I list a few examples below but the authors should investigate the effect of autopolyploidy thoroughly. As far as I understand, SMC++ and PSMC assume diploid inheritance like a human being. Intuitively, it may be usable if you randomly choose two haplotypes, but a thorough theoretical basis should be explained if it would be included in the revised manuscript.

Furthermore, the generation time of 10 and 15 years (line 741) are very likely minimum estimates, because the trees will produce seeds for a long until a gap may be

formed in the forest or other reasons. Relative time estimation of N_e would be interesting but the usage of absolute time to compare the demography with glacial cycles should be omitted.

>>R: Thank you for your suggestions, especially those regarding tetraploid analysis. We may not have fully understood the methods presented in Monnahan et al. (2019) *Nature Ecol Evol* 3: 457, as they did not seem to conduct a demographic analysis using those tools. Instead, we appreciate your suggestion of extracting and analyzing two haplotypes. This approach treats the tetraploid genome as a diploid genome and aligns with the model assumptions. Accordingly, we extracted the two longest monoploid genomes for analysis (Reviewer 2 recommended this approach as it would include the most genomic information). Furthermore, we also adjusted the generation times to 15, 20, and 30 according to your suggestions.

Surprisingly, different generation times and methods (SMC++ and PSMC) yielded highly consistent results. Please see below for the revised Figure 4a and Supplementary Figure 11. The results section was also revised thoroughly (please also see below). It is indeed a challenge to provide a clear theoretical explanation for the mechanism behind these robust results. Intuitively, your suggestion that treating the tetraploid genome as diploid to fit the model assumption likely contributes significantly to the robustness of the results. Many thanks again for the suggestion. (Discussion about Otto and Whitton, 2000; please see the reply below)

We revised the results as:

The demographic history for both species was performed using SMC++³⁰ (see Methods), based on generation times of 15, 20, and 30 years (Supplementary Table 21). We found that the effective population sizes of both species increased after the Last Glacial Maximum (LGM, c. 19,000 years ago), and reached a maximum around 6,000 – 10,000 years ago for *H. hainanensis* and 3,000 – 9,000 years ago for *H. reticulata*

(Fig. 4a). However, both species experienced a sharp decline recently around 1,500 – 2,400 years ago for *H. hainanensis* and 1,200 – 2,400 years ago for *H. reticulata*. During this period, effective population sizes decreased from the maximum (*c.* 5.0×10^6 for *H. hainanensis* and *c.* $4.0 \times 10^6 - 6.0 \times 10^6$ for *H. reticulata*) to less than 1,000 (Fig. 4a). The demographic analysis with PSMC³¹ showed consistent results (Supplementary Fig. 11). Since human activities in Hainan Island can be traced back to 3000 years ago³², the above evidence suggests that human intervention may have disrupted the maintenance of their populations after the glacial period, thus serving as a primary major factor contributing to species endangerment.

In line 239, a comparison with cherry-picking several tree species does not make sense and so should be omitted. In line 221, Tajima's D, CLR, and iHS are not designed for autopolyploid species, although they can work for allopolyploid species.

>>R: Thanks. Accordingly, we have removed the relevant descriptions of genetic diversity of several tree species from the main text. Indeed, Tajima's D, CLR, and iHS are not designed for autopolyploid species. Monnahan et al. (2019) and Zhang et al. (2018) converted the tetraploid data into diploid data by mapping against one monoploid genome and considering two-allelic sites only. Our previous approach followed their methods. However, their analyses mainly focused on neutral processes and did not include selection analysis. Therefore, there is currently no reference for population genomic analysis regarding the signature of selection in autopolyploids.

To explore the effect of autopolyploidy and determine appropriate methods, we followed your previous advice and re-ran the entire population genomics analysis, including structure, selection analysis, and ROH, using the two longest monoploids as templates for mapping. This approach treats the tetraploid genome as a diploid genome (referred to as the diploid model analysis). To facilitate comparison, we have compiled all the results together (please see the figure below).

In the structure analysis of the neutral process, we found consistent results between the two methods. However, in the selection analysis, the diploid model analysis identified a greater number of positively selected genes than the tetraploid model. Otto

and Whitton, 2000 predicted that the efficiency of positive selection is reduced in polyploids, particularly when the mutation is not dominant. Our observation is consistent with their prediction that beneficial mutations are less efficient in reaching high frequencies in polyploid, leading to less prominent positive selection signatures in the tetraploid model analysis. On the other hand, having four alleles per locus (a characteristic of autotetraploidy) increased the genomic heterozygosity. Therefore, although the previous method of mapping against the longest monoploid genome was not perfect, it remains the optimal solution. We retained our previous methods in this manuscript. Thank you for the fruitful discussion, which has greatly enhanced our understanding of the theory and analysis on tetraploid species.

The two longest monoploid genomes as reference

The longest monoploid genome as reference

3. Ancient polyploidy was already reported

Ng et al. (2021) and Wang et al. (2022) reported a genome duplication event preceding the divergence of Dipterocarpoideae species. The authors did find it out, but the original paper should be cited. Otherwise, it will give a false impression of novelty.

>>R: Many thanks. We agree it is critically important to acknowledge previous findings. We therefore referred to these excellent studies in the results and discussion, indicating our findings were consistent with them.

In the results, we revised and added:

The genetic distance of paralogs within these species was found to be greater than the distance of orthologs between pairs of species, indicating that WGD-1 occurred before the diversification of Dipterocarpoideae species (Fig. 2b and Supplementary Fig. 4). This finding is consistent with previous genomic studies^{13, 16, 17}, suggesting that WGD facilitated the diversification of Dipterocarpoideae species.

In the discussion, we revised and added:

Detection of WGD-1 and WGD-2 probably reflect a genome-scale adaptation to ancient climate/environmental changes. We found that the WGD-1 preceded the diversification of Dipterocarpoideae species, which is consistent with several previously published studies on genomes in this clade^{13, 16, 17}.

4. Time estimation

There has been a long dispute on when Dipterocarpaceae/Dipterocarpoideae originated (see references cited in the literature below). Traditionally, the Gondwanan vicariance hypothesis based on broad geographic distribution proposed an old origin of >120 million years ago. However, the estimates are variable among phylogenetic studies, and most importantly, the Gondwanan hypothesis contradicts the general time scale of angiosperm evolution. Ng et al. conducted a detailed analysis using the genome data and fossil calibration and reported much younger divergence. In 2022, new microfossils of dipterocarps were reported, which can push the divergence time even younger (Bansal et al. *Science* 375:455, 2022). The issue is currently unsolved and can be potentially a major contribution to this manuscript. The authors cited Bansal et al. 2022 but did not discuss the dispute at all.

I should note that the methodologies the author used are primitive. The authors should examine the complexity and many assumptions in the time estimation. For calibration, authors took values from the TimeTree webpage, but this does not seem sophisticated. Particularly, little data on dipterocarp and its relatives were available. For example, when "Hopea" and *Theobroma cacao* was filled in on its webpage, the outcome was 80 MYA (CI 38.7-85.0 MYA). In line 139, the authors reported c. 68.2

MYA without providing a confidence interval. This illustrates the insufficient quality of the current analysis. At the moment, it would be wise to use two different ways of fossil calibration, the new ones by Bansal et al., and traditional ones used by Ng et al. Otherwise, it may be better to remove time estimation from this manuscript.

>>R: Many thanks for introducing the Gondwana hypothesis into this study, which has substantially enhanced the significance of this study. Regarding the controversies surrounding this topic, we have referred to these previous studies and made a throughout revision in the introduction and discussion sections. Please refer to our responses to the first major comment.

Following your suggestions, we identified 12 dipterocarp genomes from Ng et al. 2021, Wang et al. 2022 and Tian et al. 2022, and included them all in our phylogenomic analysis. Furthermore, we calibrated the phylogenetic tree using the fossil points from Bansal et al. and the calibration provided by Ng et al. Both calibrations yielded consistent estimates of divergent times, supporting the Gondwana origin hypothesis. Please refer to the revised Fig. 2a and Supplementary Fig. 4a attached below. We have also updated the methods and results selections (Please also see below).

We would like to note that the gene family expansion/contraction analysis re-run using the new dating, and the results were largely consistent with the previous version (Fig. 4e; Supplementary Fig. 6), which revealed one expanded and 36 contracted gene families in the comparison with Malvaceae species, and the comparison with temperate trees identified three expanded and 53 contracted gene families.

We revised the maintext as:

To investigate the phylogenetic relationships among Dipterocarpoideae species and their divergence with the closest family Malvaceae, we performed phylogenomic analysis including 19 Dipterocarpoideae species and two Malvaceae species. These are seven Dipterocarpoideae species assembled in this study, 12 Dipterocarpoideae species (*Dipterocarpus alatus*, *Dipterocarpus gracilis*, *Dipterocarpus intricatus*, *Dipterocarpus zeylanicus*, *Hopea mollissima*, *Hopea odorata*, *Shorea leprosula*, *Shorea roxburghii*, *Shorea henryana*, *Vatica xishuangbannaensis*, *Vatica odorata* and *Vatica rassak*) from Tian et al.¹³, *Theobroma cacao* and *Gossypium rainmondii*^{25, 26}. The maximum likelihood (ML) tree constructed with 918 single-copy orthologs (from

21,109 – 48,040 clustered gene families (Supplementary Fig. 3 and Supplementary Table 9)). We adopted the set of calibrations in Bansal et al.¹⁵, including the divergence time of *Vatica* species (54 MYA), *Shorea* species (54 MYA), and *Dipterocarpus* species (68.5 MYA)) (Fig. 2a). The calibrated tree dated the divergence between Dipterocarpoideae and Malvaceae to 108.0 (95% PHD: 97.4-117.3) MYA, and the earliest divergence within Dipterocarpoideae was estimated to occur at 101.7 (95% PHD: 91.5-115.1) MYA (Fig. 2a). To test the robustness of this estimation, we adopted a different set of calibration in Ng et al.¹⁷; including the divergence time between *T. cacao* and *G. raimondii* (55.8 MYA), and that between *Shorea* and *Dryobalanops* (34 MYA). The calibrated tree revealed the divergent time between Dipterocarpoideae and Malvaceae as 125.7 (95% PHD: 102.1-173.3) MYA) and that among Dipterocarpoideae species being 94.6 (95% PHD: 73.3-128.1) MYA) (Supplementary Fig. 4b). As these two calibration sets reveal generally consistent results, our study represents a most comprehensive phylogenomic analysis of Dipterocarpoideae, supporting the origin of Dipterocarpaceae in Western Gondwanaland.

5. Cherry-picking of genes under selection

Although the authors said "GO enrichment analysis" (line 174), I could not find any statistics. Rather, it seems that the authors arbitrarily picked up genes relevant to UV, oxidative stress, and whatever they want to discuss. The authors can find standard GO enrichment analysis in many genome papers. "Temperate tree comparison" or "Malvaceae comparison" does not seem to make sense because dipterocarp species and Malvaceae species would form a single clade, respectively, and after all, there is only a single comparison. If this should be retained, the validity of the methods should be explained in detail. Moreover, their hypothesis in line 62 "At the emergent layer, these species are challenged with intensive ultraviolet (UV) radiation and associated oxidative stress. Adaptation to such conditions is thus critical for these species to occupy this niche" does not seem to be well defended. For example, tree species living in forest gaps may be subjected to stronger stress, while the seedlings of canopy species

may grow understory. For example, Ng et al. 2021 tested the importance of drought stress using the retained duplicated genes, and here the authors could test it using the signature of selection on each gene. Even if it is supported or not, the test can be interesting.

>>R: Manty thanks for the comments. In the previous version, perhaps we did not make it clear why we conducted these two comparisons and how we identified these genes. This may lead to some misunderstandings and give the impression that we were cherry-picking these genes. The two comparisons were intended to find genes responsible for the adaptation to the emergent layer of tropical rainforest. We did not report GO or KEGG enrichment results as none terms were significant due to the limited number of total genes. The genes were located systematically by intersecting the functional annotation of comparisons. We have made a thorough revision to clarify these points. Please see below for the related revision in the results section. We hope these revisions convince you that our approach was systematic and unbiased.

In the results, we revised and added:

Since Dipterocarpoideae and Malvaceae species involved were both tropical tree species but occupying different canopy layers, the comparative analysis between them aimed to identify positively selected genes of dipterocarps specifically adapted to the emergent canopy layers. The comparative analysis against temperate tree species was then to identify the positively selected genes adapted to tropical environments. The intersection of these two sets of genes thus would provide robust signatures of positive selection regarding the adaptation of Dipterocarpoideae to the emergent canopy layer of tropical rainforests.

The comparative analysis against Malvaceae identified 171 positively selected genes, whereas that against temperate tree species identified 191 (Supplementary Fig 6). As the number of genes was limited and the corresponding KEGG and GO enrichment analysis yielded no significant results, we presented the functional annotation of GO and KEGG of these genes instead (Supplementary Tables 11 and 12). We found that both comparative analyses annotated functions relating to DNA repair and antioxidation, including Nucleotide excision repair (KEGG), Riboflavin

metabolism (KEGG), DNA repair (GO), Double-strand break repair via homologous recombination (GO), Nucleotide-excision repair (GO) and Response to oxidative stress (GO) (Fig. 3a,b; Supplementary Tables 11 and 12). Such a consistent functional annotation arose from different genes identified in either analysis, such as Bloom syndrome protein (*BLM*) in DNA repair, UDP-sugar pyrophosphorylase (*USP*) in antioxidation, DNA mismatch repair protein (*MSH6*) in DNA repair and Pyridoxine 4-dehydrogenase (*PLRI*) in antioxidation (Supplementary Tables 13 and 14). More importantly, such a consistent functional annotation also arose from the same genes identified by both analyses, including DNA repair protein RAD51D (in homologous recombination), Structural maintenance of chromosomes protein 5 (*SMC5*), and riboflavin kinase (*RFK*) in antioxidation (Fig. 3c, Supplementary Fig. 7, Supplementary Tables 13 and 14).

As you suggested, adaptation to drought may also be an important mechanism for dipterocarps to thrive in tropical environments. This adaptation is not specific to the emergent canopy layer but to the overall tropical environments. In our comparison with temperate tree species, we have identified two key candidate genes associated with this adaptation. Please see below for the related revision in the results section.

In the comparative analysis against temperate tree species, we identified two positively selected genes (O-succinylbenzoate-CoA ligase (AEE14) and L-ascorbate peroxidase relevant to the plants' response to drought stress (Supplementary Table 13), which is involved in KEGG pathways of ubiquinone and other terpenoid-quinone biosynthesis (map00130) and glutathione metabolism (map00480), respectively. The finding of these positively selected genes is consistent with a previous study indicating that dipterocarps have adapted to the rare and irregular drought events in tropical environments¹⁷.

6. data availability

Data availability section should state more specifically the availability of the assemblies and annotation files.

>>**R:** We have revised the data availability statement as follows:

All data used in the analysis of this study have been deposited in the CNSA (<https://db.cngb.org/cnsa/>) in the BIG Data Center, Beijing Institute of Genomics (BIG), Chinese Academy of Sciences, BioProject: ID: PRJCA017262.

Minor issues

line 184, which is involved

>>**R:** Revised. Line 242.

Reviewer #2 (Remarks to the Author):

Comments for authors

NCOMMS-22-41572-T: Liu et al. Life up high: the adaptations and demography of the Asian rainforest titans, Dipterocarpoideae.

In this manuscript, Liu et al. report seven dipterocarps genomes assembled to the chromosome level; 5 were diploid (*Shorea robusta*, *Dipterocarpus turbinatus*, *Hopea chinensis*, *Vatica mangachapoi* & *Parashorea chinensis*) and 2 were determined as autotetraploid (*Hopea hainanensis* & *H. reticulata*). The dipterocarp genomes shows evidence of whole-genome duplication (WGD) in its evolutionary past, preceding their lineage diversification at Oligocene and two lineage-specific autotetraploidization. Comparative genomics using 5 of the diploid dipterocarp species with 5 temperate species and 5 Malvaceae species, they found positively selected genes that relate to the adaptation above emergent canopy layer. Using resequencing samples of ~30 individuals each from the two *Hopea species* (autotetraploid) they found expansion of effective population size during last glacial period but suffers shape decline during post glacial period due to rising sea levels.

The assembled genomes quality are good, (except for the 2 autotetraploids – see below) but interesting observation was made by the authors on these genomes, e.g. regarding the involvement of DNA repair in contributing to the adaptation to the UV-rich above emergent canopy layer.

>>**R:** We highly appreciate your positive comments on the significance of our study and the constructive suggestions. We have conducted thorough revisions according to your suggestions, and please refer to our point-by-point responses for details. We hope you find that the significance and solidity of this study have been further enhanced after these revisions.

Several concerns on this manuscript:

Sequence genomes

This manuscript adds important genome resources to the currently growing list of sequenced dipterocarp genomes. Here are the list:

1. Ng et al. 2021 Communications Biology – *Shorea leprosula*
2. Wang et al. 2022 Plant Biotechnology Journal – *Dipterocarpus turbinatus* & *Hopea hainanensis*
3. Tang et al. 2022 DNA Research – *Vatica mangachapoi*
4. Tian et al. 2022 Plant Communications – *Dipterocarpus alatus*, *D. gracilis*, *D. intricatus*, *D. zeylanicus*, *Hopea mollissima*, *H. odorata*, *Parashorea chinensis*, *Shorea roxburghii*, *S. leprosula*, *S. henryana*, *S. xishuangbannaensis*, *Vatica odorata* & *Vatica rassak*

I would strongly suggest that the authors to review and include them in the introduction and discussion. If not too much, include them in their analysis to get better conclusion particularly on the phylogenomics and genome evolution.

>>**R:** Thank you for your suggestions and references. We highly acknowledge the importance of recognizing previous studies. Following your suggestions, as well as those from Reviewer 1, we have thoroughly revised the introduction and discussion to summarize the four above-mentioned studies from the perspective of the Gondwana hypothesis. Further, we have identified 12 genomes from these studies that were not previously included in our sequencing efforts, and incorporated them into our revised phylogenomic analysis. We greatly appreciate your valuable suggestions, and our study, now encompassing 19 dipterocarp species, represents the most comprehensive

comparative genomic analysis of the Dipterocarpaceae to date. These findings provide robust evidence supporting the Gondwana origin of this clade. Please refer to the revised Fig. 2a presented below. We have carefully highlighted all the corresponding revisions in the main text and included them in this reply for your convenience (Please see below).

In the introduction, we added:

Earlier studies proposed that Dipterocarpaceae originated in Western Gondwanaland in Early Cretaceous (c.120 million years ago (MYA)), before the separation of Africa and South America¹⁴. Recent evidence from pollen fossils¹⁵ and a phylogenomic study¹³ supported such a Western Gondwanaland origin, albeit with a slightly recent dating (c.102.9 MYA¹⁵ and 105.0 MYA¹³). These studies suggested that Dipterocarpaceae dispersed to India during the mid-Cretaceous and subsequent to Southeast Asia facilitated by India-Asia collision. However, genomic studies involving different sets of dipterocarp species revealed a much more recent divergence between Dipterocarpaceae and its closest family Malvaceae (c. 84 MYA¹⁶ and 86–98 MYA¹⁷). Therefore, there is still a controversy surrounding the evolution of these keystone rainforest species; resolving it will provide valuable insights into the origins of megathermal angiosperms.

We would like to acknowledge that previous genomic studies have sequenced several Dipterocarpoideae species. Taking advantage of these genomic studies, we aim to (1) conduct a more comprehensive phylogenomic analysis of Dipterocarpoideae to resolve the controversy about their origin; (2) identify positively selected genes underlying the adaptation to the emergent layer and tropical forest environments; and (3) reveal the demographic history and infer factors contributing to endangerment.

In the results, we revised:

To investigate the phylogenetic relationships among Dipterocarpoideae species and their divergence with the closest family Malvaceae, we performed phylogenomic analysis including 19 Dipterocarpoideae species and two Malvaceae species. These are seven Dipterocarpoideae species assembled in this study, 12 Dipterocarpoideae species (*Dipterocarpus alatus*, *Dipterocarpus gracilis*, *Dipterocarpus intricatus*, *Dipterocarpus zeylanicus*, *Hopea mollissima*, *Hopea odorata*, *Shorea leprosula*, *Shorea roxburghii*, *Shorea henryana*, *Vatica xishuangbannaensis*, *Vatica odorata* and *Vatica rassak*) from Tian et al.¹³, *Theobroma cacao* and *Gossypium raimondii*^{25, 26}. The maximum likelihood (ML) tree constructed with 918 single-copy orthologs (from 21,109 – 48,040 clustered gene families (Supplementary Fig. 3 and Supplementary Table 9)). We adopted the set of calibrations in Bansal et al.¹⁵, including the divergence time of *Vatica* species (54 MYA), *Shorea* species (54 MYA), and *Dipterocarpus* species (68.5 MYA) (Fig. 2a). The calibrated tree dated the divergence between Dipterocarpoideae and Malvaceae to 108.0 (95% PHD: 97.4-117.3) MYA, and the earliest divergence within Dipterocarpoideae was estimated to occur at 101.7 (95% PHD: 91.5-115.1) MYA (Fig. 2a). To test the robustness of this estimation, we adopted a different set of calibration in Ng et al.¹⁷, including the divergence time between *T. cacao* and *G. raimondii* (55.8 MYA), and that between *Shorea* and *Dryobalanops* (34 MYA). The calibrated tree revealed the divergent time between Dipterocarpoideae and Malvaceae as 125.7 (95% PHD: 102.1-173.3) MYA and that among Dipterocarpoideae species being 94.6 (95% PHD: 73.3-128.1) MYA (Supplementary Fig. 4b). As these two calibration sets reveal generally consistent results, our study represents a most

comprehensive phylogenomic analysis of Dipteroocarpoideae, supporting the origin of Dipteroocarpaceae in Western Gondwanaland.

In the discussion, we added:

Here, the phylogenetic history constructed using a total of 19 Dipteroocarpoideae genomes agrees with the results from the early view¹⁴ and some recent phylogenomic/phylogenetic studies^{13, 15}, thus supporting the Western Gondwanaland origin of Dipteroocarpaceae. Different calibrations yielded consistent results, indicating this estimation of divergent time is robust. This finding is largely consistent with the evolutionary history of angiosperms revealed by genomic data³⁵ and new fossil evidence³⁶, showing far earlier splits between major clades of angiosperms (occurring at the Jurassic and Lower Cretaceous) than traditionally expected.

C-value

In addition, four out of the seven species presented in the present manuscript have also been published. Particularly, study by Wang et al 2022 (above) reported that *Hopea hainanensis* as diploid and their sequenced assembly at the chromosome scale does not showed any autotetraploidization genome size as reported in this study. However, in the present study, the authors justified their finding of tetraploidization in *H. hainanensis* by karyotyping (but did not discuss what Wang et al. has reported). A better resolution for Supplementary Fig 1q (FISH probe 18S rDNA) would really help. Besides, why similar approach was not done to confirm the autotetraploidization of *H. reticulata*? In addition, why didn't consider estimating the genome content and ploidy of the two autotetraploid *Hopea* species by measuring the C-value (amount of DNA in the unreplicated gametic nucleus) using flow cytometry?

>>**R:** Many thanks for your questions. Regarding the karyotype analysis of *H. reticulata*, it was quite unfortunate that we were unable to collect the necessary materials of this species. Karyotype analysis requires the root tips of freshly germinated seedlings. During our field survey from 2018-2021 in Ganshiling (Hainan, China, the only recorded wild population) of *H. reticulata*, we did not find any seeds. This lack of seed

production is probably due to the mast seeding of *H. reticulata*, as such extreme mast seeding has been recorded in many Dipterocarpoideae species (Ghazoul, *Dipterocarp Biology, Ecology, and Conservation*, Oxford University Press, 2016). We have provided an explanation for this in the Methods section.

Regarding your second question, it is an excellent idea to conduct flow cytometry to measure the C-value, which only requires fresh leaves. Therefore, we collected fresh leaves for *H. hainanensis*, *H. reticulata* and *H. chinensis*. These leaves were all obtained from the same plant used for genome assembly. We used diploid *H. chinensis* and *Solanum lycopersicum* as reference to analyze the tetraploid peak and estimate the C-value. The corresponding results have been added to Figure S1 and Table S1 (please also see below), providing robust evidence supporting tetraploidization. We have also revised the results section accordingly (Please see below).

Sample	Genometric mean	CV error (%)	2C (pg)	Genome size (Mb)
Hopea chinensis	10647.2	9.66	0.63	0.62
Solanum lycopersicum	35604.3	5.03	2.12	
Hopea reticulata	15993.9	8.57	1.36	1.33
Solanum lycopersicum	25019.7	4.02	2.12	
Hopea hainanensis	846.9	6.78	1.66	1.62
Solanum lycopersicum	1082.3	3.16	2.12	

In the results, we added:

Moreover, by using *Solanum lycopersicum* (2C = 2.12) as a reference, the flow cytometry results revealed the C-values of *H. chinensis*, *H. reticulata* and *H. hainanensis* are 2C equal to 0.63, 1.36, and 1.66 (Supplementary Fig. 1j-l and Supplementary Table 1). Compared to the diploid *H. chinensis*, the cytometry analysis identified a clear peak confirming the tetraploidy of these two species (Supplementary Fig. 1m-n).

In addition, following your suggestions, we acknowledged Wang et al 2022's *H. hainanensis* assembly in the introduction, and in particular discussed the possibility of ploidy variation among different populations of this species.

In the discussion, we added:

The genome of *H. hainanensis* was previously reported as diploidy¹⁶, and this is probably because this species may also have both tetraploid and diploid populations, with the former distributed on Hainan Island and the latter scattered on the mainland of Southeast Asia.

Monoploid

The known obstacle in the de novo assembly of an autopolyploid genome is distinguishing and separating very similar haplotypes, which are often assembled as highly fragmented sequences until recently with the use of chromosome-scale and haplotype-resolve genome assembly approach done in tetraploid potato (Sun et al. 2022 Nat Genet). Will the selection of the longest monoploid with most annotated protein-coding genes as reported in this study meaningful? What were the proportions of

annotated protein-coding genes for the other monoploids?

>>**R:** Many thanks for your questions. Yes, assembling allele-aware autotetraploid genomes presented a significant challenge in genome study. Sun et al. (2022) proposed a new assembly strategy using haploid pollen data, which achieved a contig N50 of ~6M. This was an impressive achievement. When we started our project in 2019, we primarily referenced to Zhang et al. 2018 Nat Genet who provided the first allele-aware assembly of sugarcane, their corresponding methods (Zhang et al. 2019 Nat Plants), and a close personal communication with the authors. These collaborative efforts resulted in an assembly of comparable quality to Zhang et al. (2018). In terms of gene completeness, our assembly results were acceptable (Supplementary Table 5). However, we acknowledge that some very similar sequences were not assembled into allele-aware genomes, which could be noticed from the mapping results of resequencing data (Supplementary Table 3).

Regarding the longest monoploid, it contains 2978 (100%) four-allele genes, 9913 (88.5%) three-allele genes, 8579 (73.4%) two-allele genes, and 4430 (26.6%) one-allele genes. Therefore, if our analysis only included monoploid, a portion of genes would indeed be neglected. To address this issue, we incorporated genes from the longest monoploid genome with all one-allele genes from other monoploids in the phylogenomic analysis. The revised phylogenomic results, mentioned earlier in this response, had included these changes. Furthermore, in the effective population size analysis, we revised the analysis to include the two longest monoploids (see the figure below). This adjustment aligns with the assumptions of the methods, as suggested by Reviewer 1. For the population genetic analysis of resequencing, through discussions with Reviewer 1, we found that using the longest monoploid as the mapping reference is still the optimal solution, and thus we retained the previous results in this part. We hope that our efforts have successfully captured a maximum amount of information from the tetraploid genome while fulfilling the assumptions of the analysis methods.

lineage-specific polyploidization

The authors also suggest that lineage-specific polyploidization after speciation but from result clearly such event does not occur on *H. chinensis* which are closely related to *H. reticulata*. Further elaboration and explanation are necessary to support this statement.

>>R: Many thanks for you constructive suggestion. Indeed, the lineage-specific polyploidization was not clearly explained in the previous version. We have thoroughly revised this paragraph in the results section to clarify this point. Please find the revised paragraph below.

In the results, we added:

The phylogenomic results revealed that the diploid *H. chinensis*, and tetraploid *H. reticulata* and *H. hainanensis* shared a most common ancestor (Fig. 2a). The distribution of K_s and 4DTV distances showed that the genetic distance between paralogs in the two tetraploid species was smaller than the distance between orthologs in any pairwise comparison of the three species (Fig. 2b and Supplementary Fig. 4b). This observation suggests the whole genome duplication event (WGD-2) corresponding to the tetraploid species occurred more recently than the divergence of the three species. Since the whole genome duplication in *H. reticulata* and *H. hainanensis* took place after their speciation events, the formation of tetraploid is lineage-specific.

Specifically:

Page 5, lines 78-81: The sentence here was so confusing to read. Please revise.

>>**R:** Thanks. This sentence is revised as follows.

Here, we performed *de novo* genome assembly of seven Dipterocarpoideae species, including five species from the Shoreeae tribe (basic chromosome number $n=7$) and two species from the Dipterocarpeae tribe ($n=11$)²⁴(Table 1).

Page 5, lines 94-95: The sentence was very confusing.

>>**R:** Thanks. This sentence is revised as follows.

k -mer analysis revealed that all the species sequenced in this study are diploid, except for *H. hainanensis* and *H. reticulata* (Supplementary Fig. 1a-g).

Besides the sampled trees for *de novo* assembly, we performed k -mer analysis for eight trees chosen for resequencing (four trees for each autotetraploid species) and found evidence for their tetraploidy (Supplementary Fig. 1o-v).

Supplementary Fig 1a does not show similar kmer pattern as to Supplementary Fig 1h-k. Please clarify this.

>>**R:** Thanks for this helpful suggestion. The k -mer analysis estimates the ploidy of a sequenced individual via calculating the ratio of the depth of the last peak to that of the

first peak, e.g., the depths of the last and the first peak in Supplementary Fig. 1a were *c.* 500 and *c.* 120, and thus the ratio approximates to 4, indicating it is a tetraploid organism (see the figure below). Similarly, the depths of the last and the first peak in the sampled trees for resequencing were about 80-100 and about 20-25, also suggesting that they are tetraploid. The differences in the number of peaks and depths of peaks between Supplementary Fig. 1a and Supplementary Fig. 1h-k are the result of distinct Illumina sequencing data between the *de novo* assembly (264 Gb for *H. hainanensis*) and resequencing (< 37 Gb for each sampled tree for population genomics).

Figure. The results of *k*-mer analysis of the *de novo* assembly (a) and a resequencing tree of *H. hainanensis* (b).

Page 8, lines 154-158: Why the two tetraploid species were not included in the comparative genomics analysis? Plus, there were several others sequenced and published dipterocarp genomes, it would be much more interesting to know if they share the similar genomic signature of adaptation.

>>**R:** Many thanks for the question. We fully agree that including all published dipterocarp genomes in the positive selection analysis would yield interesting results. This approach is challenging, because the number of orthologs available for analysis decreased dramatically with an increasing number of genomes. When all 19 Dipterocarpaceae genomes were included, only less than 100 single-copy orthologs

were retained for analysis.

Incorporating the two tetraploid species into the analysis is feasible in the sense of the number of available orthologs. We did not include them, mainly because some genes are anticipated to acquire new functions after tetraploidization. This could disturb the consistent signature of positive selection observed from diploids. To gain a deeper understanding of this issue, we included the two tetraploid genomes in the analysis. The comparative analysis against temperate tree species yielded 93 positively selected genes, and with Malvaceae species yielded 92 positively selected genes (please see the figure below). Intriguingly, for the genes of interest, the results of the tetraploid analysis were a subset of the diploid analysis (Please see the tables below, where we have highlighted the consistent results with Supplementary Figures 13 and 14). This subset relationship reflects the robustness of the previous diploid analysis, and it is possibly because the several genes in tetraploids have new functions and were captured by the analysis. Since the results of both analyses are largely consistent, indicating genes involved in DNA repair and antioxidation were under positive selection, and diploid estimates may be more robust, we presented the results of the diploid analysis in the main text.

The temperate tree comparison

Gene/Gene family ID	Encoded protein	KEGG pathway/GO term	Functional category	Reference
evm.model.000049F.44	DNA excision repair protein (XPF)	map03440 Homologous recombination	DNA repair	Manova et al. Front. Plant Sci. 6, 885, 2015
evm.model.000026F.212	Bloom syndrome protein (BLM)	GO:0006281 DNA repair	DNA repair	Manova et al. Front. Plant Sci. 6, 885, 2015
evm.model.000148F.47	5'-nucleotidase	map03440 Homologous recombination	Antioxidation	Berger et al. Trends Biochem. Sci. 29, 111–118, 2004.
31 (expanded)	Disease resistance proteins	GO:0006952 Defense response	Plant immunity	NA
26, 34, 41, 64, 72, 355 (contracted)	Leucine-rich repeat receptor-like kinases (LRR-RLKs)	map04626 Plant-pathogen interaction	Plant immunity	29
8, 80 (contracted)	TIR-NBS-LRR	map04626 Plant-pathogen interaction	Plant immunity	30
23 (contracted)	CC-NBS-LRR (NBS-LRR: nucleotide binding site-leucine rich repeat)	map04626 Plant-pathogen interaction	Plant immunity	30

The Malvaceae comparison

Gene/Gene family ID	Encoded protein	KEGG pathway/GO term	Functional category	Reference
evm.model.000006F.266	Lipoyl synthase (LIAS)	map00785 Lipoic acid metabolism	Antioxidation	Navari-Izzo et al. Plant Physiol. Biochem. 40, 463–470, 2002.
evm.model.000011F.38	Pyridoxine 4-dehydrogenase (PLR1)	map00750 Vitamin B6 metabolism	Antioxidation	Berger et al. Trends Biochem. Sci. 29, 111–118, 2004.
evm.model.000224F.68	Uridine nucleosidase (URH1)	map00760 Nicotinate and nicotinamide metabolism	Antioxidation	Berger et al. Trends Biochem. Sci. 29, 111–118, 2004.
evm.model.000123F.61	DNA polymerase I (polA)	map00230 Purine metabolism	DNA repair	

		map03410 Base excision repair		Manova et al. Front. Plant Sci.
		map03420 Nucleotide excision repair		6, 885, 2015
evm.model.000055F.237	DNA mismatch repair protein (MSH6)	map03430 Mismatch repair	DNA repair	Manova et al. Front. Plant Sci.
				6, 885, 2015
11 (expanded)	Disease resistance proteins	GO:0006952 Defense response	Plant immunity	NA
10, 48, 51, 68, 85, 209	Leucine-rich repeat receptor-like kinases	map04626 Plant-pathogen interaction	Plant immunity	29
(contracted)	(LRR-RLKs)			
8, 150, 376 (contracted)	CC-NBS-LRR (NBS-LRR: nucleotide binding site-leucine rich repeat)	map04626 Plant-pathogen interaction	Plant immunity	30
102(contractd)	TIR-NBS-LRR	map04627 Plant-pathogen interaction	Plant immunity	30

Page 10, line 205: Only 3 expanded gene families were observed in Suppl Table 12 not 4 expanded gene families.

>>**R**: Many thanks. As the phylogeny and divergent time were re-estimated, the gene family analysis was also updated. Please refer to the revised Supplementary Fig. 6 below, and this sentence here is revised as follows.

The comparison with Malvaceae species revealed one expanded and 36 contracted gene families, and the comparison with temperate trees identified three expanded and 53 contracted gene families (Supplementary Fig. 6).

Page 11, lines 217-219: statements are very speculative, please provide support or examples.

>>**R**: Thanks. We added a reference to this statement and refined this statement as follows.

Given that contraction of gene families is often related to specialization in biotic interaction²⁹, the observed contraction in gene families relating to plant-pathogen interaction is likely to be the molecular footprint of adaptation to specific pathogens endemic in Asian rainforests.

Page 11, line 237: ...relatively similar genetic diversity...

>>**R**: Thanks. We revised it accordingly. Line 279.

Page 12, line 260: 'endangerment' change to "decrease in population sizes"

>>**R:** Thanks. As the population demographic analysis was re-estimated, and the results were revised thoroughly. This sentence has been removed.

Supplementary Fig 12: Description was not found in the main text

>>**R:** Thanks. This figure is now cited in the sentence in Line 296.

Page 13, line 281: “extensive gene flow” – do you know what are the pollinators within the distribution areas? As in the aseasonal tropics, tiny thrips and medium size stingless bees (*Trigona* spp) were reported but they do not strongly suggest extensive gene flow, if any, it will be at the lower proportion. Perhaps, this fundamental information is critical to justify conservation and management of the species.

>>**R:** Thank you very much for this valuable information. Because Dipterocarpoideae species are exceptionally tall, conducting pollination observation on these species is quite a challenge in the field. The information you provided is thus extremely precious and valuable. We have also consulted our colleagues in Hainan studying *Vatica mangachapoi*, and they also have such a view that the gene flow of Dipterocarpoideae is often quite limited. Therefore, the observation that geographically isolated populations belong to the same genetic cluster is likely due to historical connectivity between these populations, with fragmentation and population size decline occurring only very recently. Therefore, we revised the discussion accordingly as follows.

Consistent with the dynamics of effective population sizes, these findings suggest that these populations have undergone fragmentation and a decline in population size quite recently (also please see discussion).

Supplementary Fig 14: no indication of a-d

>>**R:** Revised, and this figure has been moved into the maintext as Fig. 4d.

Page 14, lines 295-296: “...probably resulting from their tetraploidy.” Could you elaborate more on how this was resulted from tetraploidy?

>>**R:** Many thanks for the question. To answer it, we carefully consulted Otto and

Whitton's theoretical paper (Ann. Rev. Genet. 2000.34:401-437) and realized our previous explanation was incorrect. Due to the masking effect of tetraploidy on deleterious mutations, tetraploids are predicted to accumulate more deleterious mutations over time than diploids. Here, The observed lower accumulation of deleterious mutations is probably associated with population sizes declining only recently. Thus, we have revised the expression in the discussion section as follows.

Despite their small population sizes, two autotetraploid *Hopea* populations accumulated low levels of deleterious mutations, probably because the bottleneck occurred recently (less than 170 generations).

Page 14, lines 298-304: This should be best discussed along with previous findings on the other published dipterocarp genomes.

>>**R:** Revised, and we now have cited the publications of other dipterocarp genomes.

Detection of WGD-1 and WGD-2 probably reflect a genome-scale adaptation to ancient climate/environmental changes. We found that the WGD-1 preceded the diversification of Dipterocarpoideae species, which is consistent with several previously published studies on genomes in this clade^{13, 16, 17}.

Page 14, lines 309-310: The two cited references only mentioned DNA repair but what about antioxidation?

>>**R:** We have added a reference relevant to antioxidation (Zeng et al. 13: 112 Molecular Plant, 2020). Line 357.

Page 15, lines 310-315: This is a rather speculative statement. It would be interesting to consider checking the previously published genomes for confirmation.

>>**R:** Revised, and we re-organized our representation and addressed the importance to carry out further research as follows:

The contracted gene families in plant-pathogen interaction suggest that the species may be specialized to endemic pathogens in Asian rainforests. Future studies should explore the function of these gene families of Dipterocarpoideae and their

specialized interaction with pathogens.

Page 15, lines 319-321: This is speculative.

>>**R:** We have removed this sentence based on the new results of demographic analysis.

Page 15, lines 325-326: extensive gene flow? insect pollinators? Do you know what insect pollinators?

>>**R:** Thanks for your questions. Based on the evidence you mentioned above and our discussion in the reply, we both agree that these species exhibit limited gene flow. Therefore, the corresponding sentences have been revised as follows.

Furthermore, Dipterocarpaceae species are primarily pollinated by small-sized insects, such as tiny thrips, and the gene flow is anticipated to occur within a few hundred meters^{14, 47}. Here, we observed that geographically isolated populations of *H. hainanensis* belonging to the same genetic cluster was likely due to historical connectivity between these populations. This observation provides another piece of evidence that fragmentation and population size decline occurred very recently.

REVIEWER COMMENTS

Reviewer #1 (Remarks to the Author):

General comments

The comments by reviewer 2 were highly similar to mine. The authors provided additional meaningful experimental data using flow cytometry. However, in bioinformatic analysis, many additional analyses were superficial and did not address the fundamental theoretical issues in phylogenetic and population genetic analysis. The methods are so short that I cannot judge if the analyses were appropriate. Alternatively, code can be informative on the methods, but the main text does not have a statement of code availability, although the Editorial Check List said "We have provided a full code availability statement in the manuscript". The authors should consider reproducibility seriously.

My comments on each point will be shown by >.

Response to reviewers' comments

Reviewer #1 (Remarks to the Author):

Dipterocarp trees are dominant in Southeast Asian tropical forests and important for environments, forestry, and medicine. Recently, a few genomic studies reactivated long-standing debates about its evolutionary history centered on its Gondwanan vicariance hypothesis. The authors reported chromosome-level assemblies of seven dipterocarp tree species. Two of them are autotetraploid tree species, which can be a unique resource to study plant evolution in general. However, a major revision would be necessary to place the study in the context of previous research.

>>R: Many thanks for your positive comments and your constructive suggestions. Following your suggestions, we have revised the manuscript thoroughly; please find the point-to-point reply below. We hope you will find that novelty and the significance of this study have been substantially strengthened.

1. The genome assemblies of dipterocarps

1. The genome assemblies of dipterocarps, i.e., *Shorea leprosula* (Ng et al. *Commun Biol* 7:1166, 2021) with transcriptome and resequencing of other dipterocarps, *Dipterocarpus turbinatus* and *Hopea hainanensis* (Wang et al. *Plant Biotechnol* 20:538, 2022, Epub 2021 Dec), were already reported some time ago. Many results of this manuscript, such as ancient polyploidy and time estimation, were reported. The same species, *Hopea hainanensis*, was sequenced before. The current manuscript would give a false impression of novelty. This is not scientifically proper. Rather, I would suggest that the comparison with previous studies and the introduction explaining unsolved issues would highlight the importance of this new study as explained below. Their report should be integrated fully in the introduction and discussion. Furthermore, Tian et al. (*Plant Communications* 2022, 100464, 2022) reported 13 dipterocarp genomes in October 2022. I am not sure if this is before or after the submission of this manuscript and may depend on the journal policy, but integrating it would further strengthen this manuscript.

>>R: Many thanks for your suggestions and the references. Indeed, we agree that it is important to acknowledge the previous discoveries. Following your suggestions, we have highlighted the contributions of Ng et al. 2021, Wang et al. 2022, and Tian et al. 2022 while introducing the debates surrounding the origin of the Dipterocarpaceae. We have rephrased the paragraphs in the introduction and discussion sections (The revision is highlighted in blue in the main text, please also see below).

In the introduction, we added:

Earlier studies proposed that Dipterocarpaceae originated in Western Gondwanaland in Early Cretaceous (c.120 million years ago (MYA)), before the separation of Africa and South America¹⁴. Recent evidence from pollen fossils¹⁵ and a phylogenomic study¹³ supported such a Western Gondwanaland origin, albeit with a slightly recent dating (c.102.9 MYA¹⁵ and 105.0 MYA¹³). These studies suggested that Dipterocarpaceae dispersed to India during the mid-Cretaceous and subsequent to Southeast Asia facilitated by India-Asia collision. However, genomic studies involving different sets of dipterocarp species revealed a much more recent divergence between Dipterocarpaceae and its closest family Malvaceae (c. 84 MYA¹⁶ and 86–98 MYA¹⁷). Therefore, there is still a controversy surrounding the evolution of these keystone rainforest species; resolving it will provide valuable insights into the origins of megathermal angiosperms.

In the discussion, we added:

Here, the phylogenetic history constructed using a total of 19 Dipterocarpoideae genomes agrees with the results from the early view¹⁴ and some recent phylogenomic/phylogenetic studies^{13, 15}, thus supporting the Western Gondwanaland origin of Dipterocarpaceae. Different calibrations yielded consistent results, indicating this estimation of divergent time is robust. This finding is largely consistent with the evolutionary history of angiosperms revealed by genomic data³⁵ and new fossil evidence³⁶, showing far earlier splits between major clades of angiosperms (occurring at the Jurassic and Lower Cretaceous) than traditionally expected. Furthermore, as you suggested, we have integrated all published Dipterocarpaceae genomes, including those in Tian et al. 2022 in the revised phylogenomic analysis, and thus form a dataset of 19 Dipterocarp genomes. These suggestions significantly improve the novelty and solidity of our study. Please refer to the response for this specific issue below.

>The new paragraphs partly solved the lack of introduction. However, other important conclusions of the manuscript include genome duplication. A more thorough description of previous publications is necessary.

2. Novelty in autopolyploidy

Wang et al. reported the assembly of the same species *Hopea hainanensis* but they did not mention autotetraploidy at all and treated it as a normal diploid species. This manuscript provided chromosome count (Supplementary Figure 1) and careful k-mer analysis to provide strong evidence of polyploidy. Furthermore, they seem to have successfully provided phased four assemblies, although further quality checks in this aspect should be reported. In general, the genome assembly of autopolyploid species remains a major challenge in genomics. For example, the assembly of autotetraploid *Arabidopsis arenosa* is difficult, and instead, self-fertilizing allotetraploid *A. suecica* derived from *A. arenosa* was sequenced instead (Burns et al. *Nature Ecology and Evolution*, 10:1367, 2021; Jiang et al. *Nature Ecology and Evolution*, 10:1382, 2021). Supposing the quality of the phasing is quantified, the new assembly of *H. hainanensis* would provide interesting insights into the autopolyploid genome.

To support the tetraploidy, please add the empirical data of genome size measured by flow cytometry. Ng et al. (*Plant Ecology and Diversity* 9:437, 2016) explained a standard method of dipterocarps.

>>R: Many thanks for acknowledging the significance of our tetraploid assembly and for the references. The autotetraploid assembly presented a major challenge until recently. Zhang et al. 2018 *Nat Genet* provided the first allele-aware assembly of sugarcane and corresponding methods were published by Zhang et al. 2019 *Nat Plants*. We followed their approach and maintained close communication with the authors since we started the project in 2019, and these efforts resulted in the assembly of comparable quality to Zhang et al. 2018 *Nat Genet*. We also provided evidence showing the quality of the two autotetraploid genomes (Lines 133-140, 144-147, and 150-154; Supplementary Tables 3-5). So far, the allele-aware genome assembly of autotetraploid was still limited, in particular for wild species, and we hope that our work will inspire further involvement in studies of Dipterocarpaceae and wild tetraploid species.

Furthermore, we highly appreciate the methods you provided. We conducted flow cytometry analysis and estimated the C-values accordingly. The corresponding results have been added to Figure S1 and Table S1 (please also see below), providing robust evidence in support of tetraploidization. We have also revised the results section accordingly.

In the results, we added:

Moreover, by using *Solanum lycopersicum* ($2C = 2.12$) as a reference, the flow cytometry results revealed the C-values of *H. chinensis*, *H. reticulata* and *H. hainanensis* are $2C$ equal to 0.63, 1.36 and 1.66 (Supplementary Fig. 1j-l and Supplementary Table 1). Compared to the diploid *H. chinensis*, the cytometry analysis identified a clear peak confirming the tetraploidy of these two species (Supplementary Fig. 1m-n).

In the methods, we added:

To further test whether *H. hainanensis* and *H. reticulata* are tetraploid species, we estimated their genome sizes and ploidy using flow cytometry. For these two species and *H. chinensis* (a diploid congener), we collected the fresh young leaves from the sampled trees used for the de novo genome assembly. We also sampled young leaves of *Solanum lycopersicum* (genome size: 2.07

Gb; $2C = 2.12 \text{ pg}53$) as the reference for genome size estimation. We first estimated the genome sizes and $2C$ values of the three *Hopea* species by measuring the fluorescence of nuclei from the mixture of each *Hopea* species and *S. lycopersicum* and those from each single species. Then, we assessed the ploidy for *H. hainanensis* and *H. reticulata* based on their fluorescence of nuclei setting *H. chinensis* as the diploid reference. The nuclei suspension was prepared for each species following the methods described in Ng et al.⁵⁴. We measured the fluorescence from the nuclei suspension using a Sysmex CyFlow Cube6 flow cytometer, and data analysis was performed using FCSExpress 7plus.

>Flow cytometry data are satisfactory. This also solved a similar issue raised by Reviewer 2.

The methods to analyze the demographic analysis of the autotetraploid need to be thoroughly revised. As far as reading the Methods, software developed for diploid species is applied to the data. However, many assumptions are not filled. Please refer to the population genetic analysis of autotetraploid *A. arenosa* (Monnahan et al. *Nature Ecol Evol* 3: 457, 2019) and theoretical studies cited there (particularly Otto and Whitton, *Ann Rev Genet* 34:401, 2000). Here I list a few examples below but the authors should investigate the effect of autopolyploidy thoroughly. As far as I understand, SMC++ and PSMC assume diploid inheritance like a human being. Intuitively, it may be usable if you randomly choose two haplotypes, but a thorough theoretical basis should be explained if it would be included in the revised manuscript.

Furthermore, the generation time of 10 and 15 years (line 741) are very likely minimum estimates, because the trees will produce seeds for a long until a gap may be formed in the forest or other reasons. Relative time estimation of N_e would be interesting but the usage of absolute time to compare the demography with glacial cycles should be omitted.

>>R: Thank you for your suggestions, especially those regarding tetraploid analysis. We may not have fully understood the methods presented in Monnahan et al. (2019) *Nature Ecol Evol* 3: 457, as they did not seem to conduct a demographic analysis using those tools. Instead, we appreciate your suggestion of extracting and analyzing two haplotypes. This approach treats the tetraploid genome as a diploid genome and aligns with the model assumptions. Accordingly, we extracted the two longest monoploid genomes for analysis (Reviewer 2 recommended this approach as it would include the most genomic information). Furthermore, we also adjusted the generation times to 15, 20, and 30 according to your suggestions.

>The newly added method is just a single sentence without a reference.

: Line 779 "To meet the standards of population demography analyses (using SMC++ and PSMC), we mapped the filtered reads to the longest two monoploid genomes of each species to form an approximation of diploidy."

This is far too obscure. Furthermore, no codes were provided. What is the motivation to map reads to two monoploid genomes? Then, most genes have two duplicated copies. Mapping to duplicated genes is a complex issue. As a reviewer, I cannot be sure if the analysis was appropriate.

Surprisingly, different generation times and methods (SMC++ and PSMC) yielded highly consistent results. Please see below for the revised Figure 4a and Supplementary Figure 11. The results section was also revised thoroughly (please also see below). It is indeed a challenge to provide a clear theoretical explanation for the mechanism behind these robust results. Intuitively, your suggestion that treating the tetraploid genome as diploid to fit the model assumption likely contributes significantly to the robustness of the results. Many thanks again for the suggestion. (Discussion about Otto and Whitton, 2000; please see the reply below)

We revised the results as:

The demographic history for both species was performed using SMC++30 (see Methods), based on generation times of 15, 20, and 30 years (Supplementary Table 21). We found that the effective population sizes of both species increased after the Last Glacial Maximum (LGM, c. 19,000 years ago), and reached a maximum around 6,000 – 10,000 years ago for *H. hainanensis* and 3,000 – 9,000 years ago for *H. reticulata* (Fig. 4a). However, both species experienced a sharp decline recently around 1,500 – 2,400 years ago for *H. hainanensis* and 1,200 – 2,400 years ago for *H. reticulata*. During this period, effective population sizes decreased from the maximum (c. 5.0×10^6 for *H. hainanensis* and c. 4.0×10^6 – 6.0×10^6 for *H. reticulata*) to less than 1,000 (Fig. 4a).

The demographic analysis with PSMC31 showed consistent results (Supplementary Fig. 11). Since human activities in Hainan Island can be traced back to 3000 years ago³², the above evidence suggests that human intervention may have disrupted the maintenance of their populations after the glacial period, thus serving as a primary major factor contributing to species endangerment.

>Seeing the Figures, I do not understand why the authors stated "The demographic analysis with PSMC31 showed consistent results". As far as I saw, they were different in the important aspects. In the SMC analysis of *H. hainanensis* (left, top), the population size increased after the band of LGM, then drop. However, in the PSMC analysis, the vast majority of the lines did not show an increase. A subsequent drop was not observed, either. I wonder if the authors carefully examined the results. By the way, it is well known that the estimation in very recent times using SMC and PSMC may not be reliable, so should not be interpreted much.

In line 239, a comparison with cherry-picking several tree species does not make sense and so should be omitted. In line 221, Tajima's D, CLR, and iHS are not designed for autopolyploid species, although they can work for allopolyploid species.

>>R: Thanks. Accordingly, we have removed the relevant descriptions of genetic diversity of several tree species from the main text. Indeed, Tajima's D, CLR, and iHS are not designed for autopolyploid species. Monnahan et al. (2019) and Zhang et al. (2018) converted the tetraploid data into diploid data by mapping against one monoploid genome and considering two-allelic sites only. Our previous approach followed their methods. However, their analyses mainly focused on neutral processes and did not include selection analysis. Therefore, there is currently no reference for population genomic analysis regarding the signature of selection in autopolyploids. To explore the effect of autopolyploidy and determine appropriate methods, we followed your previous advice and re-ran the entire population genomics analysis, including structure, selection analysis, and ROH, using the two longest monoploids as templates for mapping. This approach treats the tetraploid genome as a diploid genome (referred to as the diploid model analysis). To facilitate comparison, we have compiled all the results together (please see the figure below). In the structure analysis of the neutral process, we found consistent results between the two methods. However, in the selection analysis, the diploid model analysis identified a greater number of positively selected genes than the tetraploid model. Otto and Whitton, 2000 predicted that the efficiency of positive selection is reduced in polyploids, particularly when the mutation is not dominant. Our observation is consistent with their prediction that beneficial mutations are less efficient in reaching high frequencies in polyploid, leading to less prominent positive selection signatures in the tetraploid model analysis. On the other hand, having four alleles per locus (a characteristic of autotetraploidy) increased the genomic heterozygosity. Therefore, although the previous method of mapping against the longest monoploid genome was not perfect, it remains the optimal solution. We retained our previous methods in this manuscript. Thank you for the fruitful discussion, which has greatly enhanced our understanding of the theory and analysis on tetraploid species.

>I do not think the authors addressed the point. A deep understanding of population genetics is lacking.

3. Ancient polyploidy was already reported

Ng et al. (2021) and Wang et al. (2022) reported a genome duplication event preceding the divergence of Dipterocarpoideae species. The authors did find it out, but the original paper should be cited. Otherwise, it will give a false impression of novelty.

>>R: Many thanks. We agree it is critically important to acknowledge previous findings. We therefore referred to these excellent studies in the results and discussion, indicating our findings were consistent with them.

In the results, we revised and added:

The genetic distance of paralogs within these species was found to be greater than the distance of orthologs between pairs of species, indicating that WGD-1 occurred before the diversification of Dipterocarpoideae species (Fig. 2b and Supplementary Fig. 4). This finding is consistent with

previous genomic studies^{13, 16, 17}, suggesting that WGD facilitated the diversification of Dipterocarpoideae species.

In the discussion, we revised and added:

Detection of WGD-1 and WGD-2 probably reflect a genome-scale adaptation to ancient climate/environmental changes. We found that the WGD-1 preceded the diversification of Dipterocarpoideae species, which is consistent with several previously published studies on genomes in this clade^{13, 16, 17}.

>The content is fine but please add this to the introduction as mentioned above.

4. Time estimation

There has been a long dispute on when Dipterocarpaceae/Dipterocarpoideae originated (see references cited in the literature below). Traditionally, the Gondwanan vicariance hypothesis based on broad geographic distribution proposed an old origin of >120 million years ago. However, the estimates are variable among phylogenetic studies, and most importantly, the Gondwanan hypothesis contradicts the general time scale of angiosperm evolution. Ng et al. conducted a detailed analysis using the genome data and fossil calibration and reported much younger divergence. In 2022, new microfossils of dipterocarps were reported, which can push the divergence time even younger (Bansal et al. *Science* 375:455, 2022). The issue is currently unsolved and can be potentially a major contribution to this manuscript. The authors cited Bansal et al. 2022 but did not discuss the dispute at all.

I should note that the methodologies the author used are primitive. The authors should examine the complexity and many assumptions in the time estimation. For calibration, authors took values from the TimeTree webpage, but this does not seem sophisticated. Particularly, little data on dipterocarp and its relatives were available. For example, when "Hopea" and *Theobroma cacao* was filled in on its webpage, the outcome was 80 MYA (CI 38.7-85.0 MYA). In line 139, the authors reported c. 68.2 MYA without providing a confidence interval. This illustrates the insufficient quality of the current analysis. At the moment, it would be wise to use two different ways of fossil calibration, the new ones by Bansal et al., and traditional ones used by Ng et al. Otherwise, it may be better to remove time estimation from this manuscript.

>>R: Many thanks for introducing the Gondwana hypothesis into this study, which has substantially enhanced the significance of this study. Regarding the controversies surrounding this topic, we have referred to these previous studies and made a throughout revision in the introduction and discussion sections. Please refer to our responses to the first major comment. Following your suggestions, we identified 12 dipterocarp genomes from Ng et al. 2021, Wang et al. 2022 and Tian et al. 2022, and included them all in our phylogenomic analysis. Furthermore, we calibrated the phylogenetic tree using the fossil points from Bansal et al. and the calibration provided by Ng et al. Both calibrations yielded consistent estimates of divergent times, supporting the Gondwana origin hypothesis. Please refer to the revised Fig. 2a and Supplementary Fig. 4a attached below. We have also updated the methods and results selections (Please also see below). We would like to note that the gene family expansion/contraction analysis re-run using the new dating, and the results were largely consistent with the previous version (Fig. 4e; Supplementary Fig. 6), which revealed one expanded and 36 contracted gene families in the comparison with Malvaceae species, and the comparison with temperate trees identified three expanded and 53 contracted gene families.

We revised the maintext as :

To investigate the phylogenetic relationships among Dipterocarpoideae species and their divergence with the closest family Malvaceae, we performed phylogenomic analysis including 19 Dipterocarpoideae species and two Malvaceae species. These are seven Dipterocarpoideae species assembled in this study, 12 Dipterocarpoideae species (*Dipterocarpus alatus*, *Dipterocarpus gracilis*, *Dipterocarpus intricatus*, *Dipterocarpus zeylanicus*, *Hopea mollissima*, *Hopea odorata*, *Shorea leprosula*, *Shorea roxburghii*, *Shorea henryana*, *Vatica xishuangbannaensis*, *Vatica odorata* and *Vatica rassak*) from Tian et al.¹³, *Theobroma cacao* and *Gossypium rainmondii*^{25, 26}. The maximum likelihood (ML) tree constructed with 918 single-copy orthologs (from 21,109 – 48,040 clustered gene families (Supplementary Fig. 3 and Supplementary Table 9)). We adopted the set of calibrations in Bansal et al.¹⁵, including the divergence time of *Vatica* species (54 MYA), *Shorea* species (54 MYA), and *Dipterocarpus* species (68.5 MYA) (Fig. 2a). The calibrated tree dated the

divergence between Dipterocarpoideae and Malvaceae to 108.0 (95% PHD: 97.4-117.3) MYA, and the earliest divergence within Dipterocarpoideae was estimated to occur at 101.7 (95% PHD: 91.5-115.1) MYA) (Fig. 2a). To test the robustness of this estimation, we adopted a different set of calibration in Ng et al. 2017 including the divergence time between *T. cacao* and *G. raimondii* (55.8 MYA), and that between *Shorea* and *Dryobalanops* (34 MYA). The calibrated tree revealed the divergent time between Dipterocarpoideae and Malvaceae as 125.7 (95% PHD: 102.1-173.3) MYA and that among Dipterocarpoideae species being 94.6 (95% PHD: 73.3-128.1) MYA) (Supplementary Fig. 4b). As these two calibration sets reveal generally consistent results, our study represents a most comprehensive phylogenomic analysis of Dipterocarpoideae, supporting the origin of Dipterocarpaceae in Western Gondwanaland.

>The details of the methods of the new analyses were not described. No codes were provided, either. Seeing the results, it is strange that similar times were estimated by two analyses using very distinct fossil calibrations. It cannot be excluded that the analysis may be fine, but for example, it is possible that priors specifying the lower and upper boundaries in the time in the Bayesian phylogenetic analysis may have biased the results. Without the details of the analysis, I cannot be sure if the analysis was appropriate.

5. Cherry-picking of genes under selection

Although the authors said "GO enrichment analysis" (line 174), I could not find any statistics. Rather, it seems that the authors arbitrarily picked up genes relevant to UV, oxidative stress, and whatever they want to discuss. The authors can find standard GO enrichment analysis in many genome papers. "Temperate tree comparison" or "Malvaceae comparison" does not seem to make sense because dipterocarp species and Malvaceae species would form a single clade, respectively, and after all, there is only a single comparison. If this should be retained, the validity of the methods should be explained in detail. Moreover, their hypothesis in line 62 "At the emergent layer, these species are challenged with intensive ultraviolet (UV) radiation and associated oxidative stress. Adaptation to such conditions is thus critical for these species to occupy this niche" does not seem to be well defended. For example, tree species living in forest gaps may be subjected to stronger stress, while the seedlings of canopy species may grow understory. For example, Ng et al. 2021 tested the importance of drought stress using the retained duplicated genes, and here the authors could test it using the signature of selection on each gene. Even if it is supported or not, the test can be interesting.

>>R: Manty thanks for the comments. In the previous version, perhaps we did not make it clear why we conducted these two comparisons and how we identified these genes. This may lead to some misunderstandings and give the impression that we were cherry-picking these genes. The two comparisons were intended to find genes responsible for the adaptation to the emergent layer of tropical rainforest. We did not report GO or KEGG enrichment results as none terms were significant due to the limited number of total genes. The genes were located systematically by intersecting the functional annotation of comparisons. We have made a thorough revision to clarify these points. Please see below for the related revision in the results section. We hope these revisions convince you that our approach was systematic and unbiased.

In the results, we revised and added:

Since Dipterocarpoideae and Malvaceae species involved were both tropical tree species but occupying different canopy layers, the comparative analysis between them aimed to identify positively selected genes of dipterocarps specifically adapted to the emergent canopy layers. The comparative analysis against temperate tree species was then to identify the positively selected genes adapted to tropical environments. The intersection of these two sets of genes thus would provide robust signatures of positive selection regarding the adaptation of Dipterocarpoideae to the emergent canopy layer of tropical rainforests.

The comparative analysis against Malvaceae identified 171 positively selected genes, whereas that against temperate tree species identified 191 (Supplementary Fig 6). As the number of genes was limited and the corresponding KEGG and GO enrichment analysis yielded no significant results, we presented the functional annotation of GO and KEGG of these genes instead (Supplementary Tables 11 and 12). We found that both comparative analyses annotated functions relating to DNA repair and antioxidation, including Nucleotide excision repair (KEGG), Riboflavin metabolism

(KEGG), DNA repair (GO), Double-strand break repair via homologous recombination (GO), Nucleotide-excision repair (GO) and Response to oxidative stress (GO) (Fig. 3a,b; Supplementary Tables 11 and 12). Such a consistent functional annotation arose from different genes identified in either analysis, such as Bloom syndrome protein (BLM) in DNA repair, UDP-sugar pyrophosphorylase (USP) in antioxidation, DNA mismatch repair protein (MSH6) in DNA repair and Pyridoxine 4-dehydrogenase (PLR1) in antioxidation (Supplementary Tables 13 and 14). More importantly, such a consistent functional annotation also arose from the same genes identified by both analyses, including DNA repair protein RAD51D (in homologous recombination), Structural maintenance of chromosomes protein 5 (SMC5), and riboflavin kinase (RFK) in antioxidation (Fig. 3c, Supplementary Fig. 7, Supplementary Tables 13 and 14).

As you suggested, adaptation to drought may also be an important mechanism for dipterocarps to thrive in tropical environments. This adaptation is not specific to the emergent canopy layer but to the overall tropical environments. In our comparison with temperate tree species, we have identified two key candidate genes associated with this adaptation. Please see below for the related revision in the results section.

In the comparative analysis against temperate tree species, we identified two positively selected genes (O-succinylbenzoate-CoA ligase (AEE14) and L-ascorbate peroxidase relevant to the plants' response to drought stress (Supplementary Table 13), which is involved in KEGG pathways of ubiquinone and other terpenoid-quinone biosynthesis (map00130) and glutathione metabolism (map00480), respectively. The finding of these positively selected genes is consistent with a previous study indicating that dipterocarps have adapted to the rare and irregular drought events in tropical environments¹⁷.

>I do not think the authors answered the point. Diverse genes were identified by the selection analysis. Although the lack of enrichment in Gene Ontology analysis or any other analysis, the authors picked up several genes relevant to UV responses. Seeing the same data, other authors can make their own stories. This is not considered objective.

6. data availability

Data availability section should state more specifically the availability of the assemblies and annotation files.

>>R: We have revised the data availability statement as follows:

All data used in the analysis of this study have been deposited in the CNSA (<https://db.cngb.org/cnsa/>) in the BIG Data Center, Beijing Institute of Genomics (BIG), Chinese Academy of Sciences, BioProject: ID: PRJCA017262.

>This is fine, but as noted at the beginning, code availability is missing although although the Editorial Check List said "We have provided a full code availability statement in the manuscript". Reproducibility is important.

Minor issues

line 184, which is involved

>>R: Revised. Line 242.

>Fine.

Reviewer #2 (Remarks to the Author):

Comments for authors

NCOMMS-22-41572A: Liu et al. Life up high: the adaptations and demography of the Asian rainforest titans, Dipterocarpoideae.

The authors have carefully taken up all my comments and satisfactorily addressed some of them. Really appreciate the effort for trying to improve the manuscript. However, I have several comments on the revised manuscript:

Page 6&7, lines 119-124: The histograms obtained from ploidy screening using the DNA flow cytometry were not really convincing particularly showing broad DNA peak rather than single

prominent DNA peak. Even the standard reference DNA peak was broad. Perhaps this may be due to the handling error during samples chopping/preparation. I strongly encourage this to be addressed.

Page 15: lines 319-320: "Is there sufficient intraspecific genetic variation within species to keep pace with the ongoing environmental changes?" The authors discussed it in Page 18: lines 385-391, does this really answer the question above?

Page 15: lines 324-325: "has resolved the previous debate on the origin of this clade" is a very bold statement, perhaps the jury is still out there. Hence, I would suggest that this statement be revised to, "The comprehensive comparative genomic analysis further support the Gondwanan origin of Dipterocarpoideae species"

Page 16: lines 348-351: This is crucial. The natural distribution of *H. hainanensis* only can be found in Hainan province and northern part of Vietnam. They suggested that previously reported genome of *H. hainanensis* by Wang et al. 2022 was 'probably' from diploid population. Where is the evidence of diploid populations?

If both tetraploid and diploid populations do exist, how do they know the 30 *H. hainanensis* individuals used for resequencing study were all autotetraploid individuals (minus 4 individuals tested for kmer analysis)?

In addition, why the suspected diploid genome of *H. hainanensis* (Wang et al 2022) were not included in the analysis?

RESPONSE TO REVIEWERS' COMMENTS

Reviewer #1 (Remarks to the Author):

General comments

The comments by reviewer 2 were highly similar to mine. The authors provided additional meaningful experimental data using flow cytometry. However, in bioinformatic analysis, many additional analyses were superficial and did not address the fundamental theoretical issues in phylogenetic and population genetic analysis. The methods are so short that I cannot judge if the analyses were appropriate. Alternatively, code can be informative on the methods, but the main text does not have a statement of code availability, although the Editorial Check List said "We have provided a full code availability statement in the manuscript". The authors should consider reproducibility seriously.

>>**R:** Many thanks for the constructive suggestions in both rounds of review, and for the positive comments on our previous reply. In this revision, we have improved methodological details, and in particular, we have provided the analysis codes and made them publicly available. Concerning the genomic analysis of polyploid species, we fully agree with the reviewer that this field is challenged by the lack of generalized methodologies. Our study, representing the first genome assembly of wild autotetraploid species (previous autotetraploid assemblies were sugarcane (Zhang et al. Nature Genetics, 50: 1565–1573, 2018), alfalfa (Chen et al. Nature Communications, 11: 2494, 2020), and potato (Sun et al. Nature Genetics, 54: 342–348, 2022)), has made various attempts to explore methodology following your constructive suggestions, and we hope to promote the research on polyploid genomes together with the reviewers. Please note that the comments are bold in black, and our responses are highlighted in blue, with the revised contents are in green. The contents from the previous round of revision are in grey.

My comments on each point will be shown by >.

>>**R:** We highlight these comments in bold to separate them from the previous round of revision, and our responses are in blue.

Response to reviewers' comments

Reviewer #1 (Remarks to the Author):

Dipterocarp trees are dominant in Southeast Asian tropical forests and important for environments, forestry, and medicine. Recently, a few genomic studies reactivated long-standing debates about its evolutionary history centered on its Gondwanan vicariance hypothesis. The authors reported chromosome-level assemblies of seven dipterocarp tree species. Two of them are autotetraploid tree species, which can be a unique resource to study plant evolution in general. However, a major revision would be necessary to place the study in the context of previous research.

>>R: Many thanks for your positive comments and your constructive suggestions. Following your suggestions, we have revised the manuscript thoroughly; please find the point-to-point reply below. We hope you will find that novelty and the significance of this study have been substantially strengthened.

1. The genome assemblies of dipterocarps, i.e., *Shorea leprosula* (Ng et al. *Commun Biol* 7:1166, 2021) with transcriptome and resequencing of other dipterocarps, *Dipterocarpus turbinatus* and *Hopea hainanensis* (Wang et al. *Plant Biotechnol* 20:538, 2022, Epub 2021 Dec), were already reported some time ago. Many results of this manuscript, such as ancient polyploidy and time estimation, were reported. The same species, *Hopea hainanensis*, was sequenced before. The current manuscript would give a false impression of novelty. This is not scientifically proper. Rather, I would suggest that the comparison with previous studies and the introduction explaining unsolved issues would highlight the importance of this new study as explained below. Their report should be integrated fully in the introduction and discussion. Furthermore, Tian et al. (*Plant Communications* 2022, 100464, 2022) reported 13 dipterocarp genomes in October 2022. I am not sure if this is before or after the submission of this manuscript and may depend on the journal policy, but integrating it would further strengthen this

manuscript.

>>R: Many thanks for your suggestions and the references. Indeed, we agree that it is important to acknowledge the previous discoveries. Following your suggestions, we have highlighted the contributions of Ng et al. 2021, Wang et al. 2022, and Tian et al. 2022 while introducing the debates surrounding the origin of the Dipterocarpaceae. We have rephrased the paragraphs in the introduction and discussion sections (The revision is highlighted in blue in the main text, please also see below).

In the introduction, we added:

Earlier studies proposed that Dipterocarpaceae originated in Western Gondwanaland in Early Cretaceous (c.120 million years ago (MYA)), before the separation of Africa and South America¹⁴. Recent evidence from pollen fossils¹⁵ and a phylogenomic study¹³ supported such a Western Gondwanaland origin, albeit with a slightly recent dating (c.102.9 MYA¹⁵ and 105.0 MYA¹³). These studies suggested that Dipterocarpaceae dispersed to India during the mid-Cretaceous and subsequent to Southeast Asia facilitated by India-Asia collision. However, genomic studies involving different sets of dipterocarp species revealed a much more recent divergence between Dipterocarpaceae and its closest family Malvaceae (c. 84 MYA¹⁶ and 86–98 MYA¹⁷). Therefore, there is still a controversy surrounding the evolution of these keystone rainforest species; resolving it will provide valuable insights into the origins of megathermal angiosperms.

In the discussion, we added:

Here, the phylogenetic history constructed using a total of 19 Dipterocarpoideae genomes agrees with the results from the early view¹⁴ and some recent phylogenomic/phylogenetic studies^{13, 15}, thus supporting the Western Gondwanaland origin of Dipterocarpaceae. Different calibrations yielded consistent results, indicating this estimation of divergent time is robust. This finding is largely consistent with the evolutionary history of angiosperms revealed by genomic data³⁵ and new fossil evidence³⁶, showing far earlier splits between major clades of angiosperms (occurring at the Jurassic and Lower Cretaceous) than traditionally expected.

Furthermore, as you suggested, we have integrated all published Dipterocarpaceae

genomes, including those in Tian et al. 2022 in the revised phylogenomic analysis, and thus form a dataset of 19 Dipterocarp genomes. These suggestions significantly improve the novelty and solidity of our study. Please refer to the response for this specific issue below.

>The new paragraphs partly solved the lack of introduction. However, other important conclusions of the manuscript include genome duplication. A more thorough description of previous publications is necessary.

>>**R:** Thanks. According to your suggestions, we have now included a more thorough description of previous findings, including the genome duplication event and others.

In the introduction, we added:

Recent genomic studies^{13, 14, 15} found a whole genome duplication (WGD) event preceding the divergence of Dipterocarpoideae species. This WGD event increased the chromosome number of Dipterocarpoideae ancestors to 12, and the following two steps of karyotype evolution gave rise to 11-chromosome species (Dipterocarpeae tribe) and 9-chromosome species (Shoreaeae tribe)¹³. (Lines 64 – 68)

2. Novelty in autopolyploidy

Wang et al. reported the assembly of the same species *Hopea hainanensis* but they did not mention autotetraploidy at all and treated it as a normal diploid species. This manuscript provided chromosome count (Supplementary Figure 1) and careful k-mer analysis to provide strong evidence of polyploidy. Furthermore, they seem to have successfully provided phased four assemblies, although further quality checks in this aspect should be reported. In general, the genome assembly of autopolyploid species remains a major challenge in genomics. For example, the assembly of autotetraploid *Arabidopsis arenosa* is difficult, and instead, self-fertilizing allotetraploid *A. suecica* derived from *A. arenosa* was sequenced instead (Burns et al. *Nature Ecology and Evolution*, 10:1367, 2021; Jiang et al. *Nature Ecology and Evolution*, 10:1382, 2021). Supposing the quality of the phasing is quantified, the new assembly of *H. hainanensis* would provide interesting insights into the autopolyploid genome.

To support the tetraploidy, please add the empirical data of genome size measured

by flow cytometry. Ng et al. (Plant Ecology and Diversity 9;437, 2016) explained a standard method of dipterocarps.

>>R: Many thanks for acknowledging the significance of our tetraploid assembly and for the references. The autotetraploid assembly presented a major challenge until recently. Zhang et al. 2018 Nat Genet provided the first allele-aware assembly of sugarcane and corresponding methods were published by Zhang et al. 2019 Nat Plants. We followed their approach and maintained close communication with the authors since we started the project in 2019, and these efforts resulted in the assembly of comparable quality to Zhang et al. 2018 Nat Genet. We also provided evidence showing the quality of the two autotetraploid genomes (Lines 133-140, 144-147, and 150-154; Supplementary Tables 3-5). So far, the allele-aware genome assembly of autotetraploid was still limited, in particular for wild species, and we hope that our work will inspire further involvement in studies of Dipterocarpaceae and wild tetraploid species.

Furthermore, we highly appreciate the methods you provided. We conducted flow cytometry analysis and estimated the C-values accordingly. The corresponding results have been added to Figure S1 and Table S1 (please also see below), providing robust evidence in support of tetraploidization. We have also revised the results section accordingly.

In the results, we added:

Moreover, by using *Solanum lycopersicum* ($2C = 2.12$) as a reference, the flow cytometry results revealed the C-values of *H. chinensis*, *H. reticulata* and *H. hainanensis* are $2C$ equal to 0.63, 1.36 and 1.66 (Supplementary Fig. 1j-l and Supplementary Table 1). Compared to the diploid *H. chinensis*, the cytometry analysis identified a clear peak confirming the tetraploidy of these two species (Supplementary Fig. 1m-n).

In the methods, we added:

To further test whether *H. hainanensis* and *H. reticulata* are tetraploid species, we estimated their genome sizes and ploidy using flow cytometry. For these two species and *H. chinensis* (a diploid congener), we collected the fresh young leaves from the sampled trees used for the de novo genome assembly. We also sampled young leaves

of *Solanum lycopersicum* (genome size: 2.07 Gb; $2C = 2.12 \text{ pg}$) as the reference for genome size estimation. We first estimated the genome sizes and $2C$ values of the three *Hopea* species by measuring the fluorescence of nuclei from the mixture of each *Hopea* species and *S. lycopersicum* and those from each single species. Then, we assessed the ploidy for *H. hainanensis* and *H. reticulata* based on their fluorescence of nuclei setting *H. chinensis* as the diploid reference. The nuclei suspension was prepared for each species following the methods described in Ng et al. 54. We measured the fluorescence from the nuclei suspension using a Sysmex CyFlow Cube6 flow cytometer, and data analysis was performed using FCSExpress 7plus.

>Flow cytometry data are satisfactory. This also solved a similar issue raised by Reviewer 2.

>>R: Thanks.

The methods to analyze the demographic analysis of the autotetraploid need to be thoroughly revised. As far as reading the Methods, software developed for diploid species is applied to the data. However, many assumptions are not filled. Please refer to the population genetic analysis of autotetraploid *A. arenosa* (Monnahan et al. *Nature Ecol Evol* 3: 457, 2019) and theoretical studies cited there (particularly Otto and Whitton, *Ann Rev Genet* 34:401, 2000). Here I list a few examples below but the authors should investigate the effect of autopolyploidy thoroughly. As far as I understand, SMC++ and PSMC assume diploid inheritance like a human being. Intuitively, it may be usable if you randomly choose two haplotypes, but a thorough theoretical basis should be explained if it would be included in the revised manuscript. Furthermore, the generation time of 10 and 15 years (line 741) are very likely minimum estimates, because the trees will produce seeds for a long until a gap may be formed in the forest or other reasons. Relative time estimation of N_e would be interesting but the usage of absolute time to compare the demography with glacial cycles should be omitted.

>>R: Thank you for your suggestions, especially those regarding tetraploid analysis. We may not have fully understood the methods presented in Monnahan et al. (2019)

Nature Ecol Evol 3: 457, as they did not seem to conduct a demographic analysis using those tools. Instead, we appreciate your suggestion of extracting and analyzing two haplotypes. This approach treats the tetraploid genome as a diploid genome and aligns with the model assumptions. Accordingly, we extracted the two longest monoploid genomes for analysis (Reviewer 2 recommended this approach as it would include the most genomic information). Furthermore, we also adjusted the generation times to 15, 20, and 30 according to your suggestions.

>The newly added method is just a single sentence without a reference.

Line 779 "To meet the standards of population demography analyses (using SMC++ and PSMC), we mapped the filtered reads to the longest two monoploid genomes of each species to form an approximation of diploidy."

This is far too obscure. Furthermore, no codes were provided. What is the motivation to map reads to two monoploid genomes? Then, most genes have two duplicated copies. Mapping to duplicated genes is a complex issue. As a reviewer, I cannot be sure if the analysis was appropriate.

>>R: This suggestion is gratefully accepted. The codes of PSMC (https://github.com/Molecology/Dipterocarpoideae-genome/Supplementary_Code8.PSMC.sh) and SMC++ (https://github.com/Molecology/Dipterocarpoideae-genome/Supplementary_Code7.SMC.sh) have been provided and the reference was added. Indeed, mapping to duplicated genomes is complex, and to ensure reliable demography analysis, we only included the uniquely mapped reads (BWA mapping quality > 0) in the analysis. In the PSMC analysis, we set -Q 20 in vcf2fq to indicate the minimum mapping quality is 20, and used the default --minimum-mapping-quality10 in the GATK variant calling (https://github.com/Molecology/Dipterocarpoideae-genome/Supplementary_Code6.mapping.varcall.sh) whose output .vcf was input for SMC++ analysis. We also added details of these parameters in the methods (Please see below).

Concerning the motivation to map reads to two monoploid genomes, we are inspired by your recommended references, i.e. the empirical study (Monnahan et al. Nature Ecol Evol 3: 457, 2019) and the theoretical study (Otto and Whitton, Ann Rev Genet 34:401, 2000). These studies indicate the general methodology for polyploid

genomics is currently lacking, and it is necessary to adopt novel approaches. Therefore, we decided to carry out the attempt using two haplotype genomes other than the traditional approach using only one haplotype genome, despite that the reliability of this method requires further investigation due to the lack of precedents. In the methods, we added a paragraph to interpret this motivation and cite the references.

The challenge of demographic analysis for autopolyploid species is mainly caused by the lack of generalized methodologies^{48, 108}. To date, the algorithms and software (e.g. SMC++³⁶ and PSMC³⁵) developed for effective population size analysis are restricted to diploid analysis. To meet the requirements of these analyses, we made an attempt to treat the autotetraploid species as diploids by using the longest two monoploid genomes as the reference genome. (Lines 776 – 780)

We estimated the recent-past effective population sizes (N_e) of *H. hainanensis* and *H. reticulata* separately using the coalescent-based inferences in SMC++ v 1.15.2³⁶. The VCF files, including 30 *H. hainanensis* samples and 32 *H. reticulata* samples, were converted to the SMC++ input files using the ‘vcf2smc’ command. The estimate was run with generation times of 15, 20 and 30 separately and the corresponding estimated mutation rates (Supplementary Table 20). Codes are available at https://github.com/Molecology/Dipterocarpoideae-genome/Supplementary_Code7.SMC.sh. (Lines 781 – 787)

In addition, we inferred historical changes in N_e of *H. hainanensis* and *H. reticulata* using PSMC v0.6.4³⁵. For each of 30 *H. hainanensis* samples and 32 *H. reticulata* samples, we performed the analysis with generation times of 15, 20 and 30 and the corresponding estimated mutation rates separately (Supplementary Table 20). The whole-genome consensus sequence was generated by SAMTOOLS v1.3, bcftools, and vcfutils.pl, setting the minimum read depth to 10, maximum read depth to 100, and minimum mapping quality 20. We ran PSMC with parameters (psmc -N30 -t15 -r5 -p 4+25*2+4+6) (https://github.com/Molecology/Dipterocarpoideae-genome/Supplementary_Code8.PSMC.sh). (Lines 788 – 795)

Surprisingly, different generation times and methods (SMC++ and PSMC) yielded

highly consistent results. Please see below for the revised Figure 4a and Supplementary Figure 11. The results section was also revised thoroughly (please also see below). It is indeed a challenge to provide a clear theoretical explanation for the mechanism behind these robust results. Intuitively, your suggestion that treating the tetraploid genome as diploid to fit the model assumption likely contributes significantly to the robustness of the results. Many thanks again for the suggestion. (Discussion about Otto and Whitton, 2000; please see the reply below)

We revised the results as:

The demographic history for both species was performed using SMC++30 (see Methods), based on generation times of 15, 20, and 30 years (Supplementary Table 21). We found that the effective population sizes of both species increased after the Last Glacial Maximum (LGM, c. 19,000 years ago), and reached a maximum around 6,000 – 10,000 years ago for *H. hainanensis* and 3,000 – 9,000 years ago for *H. reticulata* (Fig. 4a). However, both species experienced a sharp decline recently around 1,500 – 2,400 years ago for *H. hainanensis* and 1,200 – 2,400 years ago for *H. reticulata*. During this period, effective population sizes decreased from the maximum (c. 5.0×10^6 for *H. hainanensis* and c. 4.0×10^6 – 6.0×10^6 for *H. reticulata*) to less than 1,000 (Fig. 4a). The demographic analysis with PSMC31 showed consistent results (Supplementary Fig. 11). Since human activities in Hainan Island can be traced back to 3000 years ago³², the above evidence suggests that human intervention may have disrupted the maintenance of their populations after the glacial period, thus serving as a primary major factor contributing to species endangerment.

>Seeing the Figures, I do not understand why the authors stated "The demographic analysis with PSMC31 showed consistent results". As far as I saw, they were different in the important aspects. In the SMC analysis of *H. hainanensis* (left, top), the population size increased after the band of LGM, then drop. However, in the PSMC analysis, the vast majority of the lines did not show an increase. A subsequent drop was not observed, either. I wonder if the authors carefully examined the results. By the way, it is well known that the estimation in

very recent times using SMC and PSMC may not be reliable, so should not be interpreted much.

>>R: Many thanks for the suggestions. Indeed, PSMC analysis is not reliable within the last ten thousand years (Li H, and Durbin R, Nature, 475: 493–496 2011). Although SMC appears to provide accurate estimates across a time span of three orders of magnitude (10^3 – 10^6 years ago), it is not reliable within the last thousand years (Terhorst et al. Nature Genetics, 49: 303–312, 2017). Thus, for the period between the last ten thousand and one thousand years, corresponding to after the Last Glacial Maximum (LGM), it seems only SMC provided relatively reliable results. In the revision, we removed the comparison between the two approaches, and interpreted the results only on their reliable ranges.

In the results, we added:

The demographic analysis with PSMC³⁵ showed a minor declining trend in effective population sizes (N_e) for both *H. hainanensis* and *H. reticulata* from one hundred thousand years ago and continuing throughout Last Glacial Maximum (LGM, c. 19,000 years ago) (Supplementary Fig. 10). For more recent demography, the SMC++³⁶ analysis based on three different generation times (15, 20, and 30 years; Supplementary Table 20) revealed that the N_e of both species increased after the LGM, reaching a maximum six to ten thousand years ago for *H. hainanensis* and three to nine thousand years ago for *H. reticulata* (Fig. 4a). (Lines 263 – 269)

In line 239, a comparison with cherry-picking several tree species does not make sense and so should be omitted. In line 221, Tajima's D, CLR, and iHS are not designed for autopolyploid species, although they can work for allopolyploid species.

>>R: Thanks. Accordingly, we have removed the relevant descriptions of genetic diversity of several tree species from the main text. Indeed, Tajima's D, CLR, and iHS are not designed for autopolyploid species. Monnahan et al. (2019) and Zhang et al. (2018) converted the tetraploid data into diploid data by mapping against one monoplloid genome and considering two-allelic sites only. Our previous approach followed their methods. However, their analyses mainly focused on neutral processes

and did not include selection analysis. Therefore, there is currently no reference for population genomic analysis regarding the signature of selection in autopolyploids.

To explore the effect of autopolyploidy and determine appropriate methods, we followed your previous advice and re-ran the entire population genomics analysis, including structure, selection analysis, and ROH, using the two longest monoploids as templates for mapping. This approach treats the tetraploid genome as a diploid genome (referred to as the diploid model analysis). To facilitate comparison, we have compiled all the results together (please see the figure below).

In the structure analysis of the neutral process, we found consistent results between the two methods. However, in the selection analysis, the diploid model analysis identified a greater number of positively selected genes than the tetraploid model. Otto and Whitton, 2000 predicted that the efficiency of positive selection is reduced in polyploids, particularly when the mutation is not dominant. Our observation is consistent with their prediction that beneficial mutations are less efficient in reaching high frequencies in polyploid, leading to less prominent positive selection signatures in the tetraploid model analysis. On the other hand, having four alleles per locus (a characteristic of autotetraploidy) increased the genomic heterozygosity. Therefore, although the previous method of mapping against the longest monoploid genome was not perfect, it remains the optimal solution. We retained our previous methods in this manuscript. Thank you for the fruitful discussion, which has greatly enhanced our understanding of the theory and analysis on tetraploid species.

>I do not think the authors addressed the point. A deep understanding of population genetics is lacking.

>>R: We misunderstood your suggestion previously. We have now realized that your suggestion was to omit analysis of Tajima's *D*, CLR, and iHS, and delete the results. In this revision, we revised it accordingly.

3. Ancient polyploidy was already reported

Ng et al. (2021) and Wang et al. (2022) reported a genome duplication event preceding the divergence of Dipterocarpoideae species. The authors did find it out, but the original

paper should be cited. Otherwise, it will give a false impression of novelty.

>>R: Many thanks. We agree it is critically important to acknowledge previous findings. We therefore referred to these excellent studies in the results and discussion, indicating our findings were consistent with them.

In the results, we revised and added:

The genetic distance of paralogs within these species was found to be greater than the distance of orthologs between pairs of species, indicating that WGD-1 occurred before the diversification of Dipteroocarpoideae species (Fig. 2b and Supplementary Fig. 4). This finding is consistent with previous genomic studies^{13, 16, 17}, suggesting that WGD facilitated the diversification of Dipteroocarpoideae species.

In the discussion, we revised and added:

Detection of WGD-1 and WGD-2 probably reflect a genome-scale adaptation to ancient climate/environmental changes. We found that the WGD-1 preceded the diversification of Dipteroocarpoideae species, which is consistent with several previously published studies on genomes in this clade^{13, 16, 17}.

>The content is fine but please add this to the introduction as mentioned above.

>>R: Thanks. This is added in the introduction as:

Recent genomic studies^{13, 14, 15} found a whole genome duplication (WGD) event preceding the divergence of Dipteroocarpoideae species. This WGD event increased the chromosome number of Dipteroocarpoideae ancestors to 12, and the following two steps of karyotype evolution gave rise to 11-chromosome species (Dipteroocarpeae tribe) and 9-chromosome species (Shoreeae tribe)¹³. (Lines 64 – 68)

4. Time estimation

There has been a long dispute on when Dipteroocarpaceae/Dipteroocarpoideae originated (see references cited in the literature below). Traditionally, the Gondwanan vicariance hypothesis based on broad geographic distribution proposed an old origin of >120 million years ago. However, the estimates are variable among phylogenetic studies, and most importantly, the Gondwanan hypothesis contradicts the general time scale of angiosperm evolution. Ng et al. conducted a detailed analysis using the genome data

and fossil calibration and reported much younger divergence. In 2022, new microfossils of dipterocarps were reported, which can push the divergence time even younger (Bansal et al. *Science* 375:455, 2022). The issue is currently unsolved and can be potentially a major contribution to this manuscript. The authors cited Bansal et al. 2022 but did not discuss the dispute at all.

I should note that the methodologies the author used are primitive. The authors should examine the complexity and many assumptions in the time estimation. For calibration, authors took values from the TimeTree webpage, but this does not seem sophisticated. Particularly, little data on dipterocarp and its relatives were available. For example, when "Hoepa" and *Theobroma cacao* was filled in on its webpage, the outcome was 80 MYA (CI 38.7-85.0 MYA). In line 139, the authors reported c. 68.2 MYA without providing a confidence interval. This illustrates the insufficient quality of the current analysis. At the moment, it would be wise to use two different ways of fossil calibration, the new ones by Bansal et al., and traditional ones used by Ng et al. Otherwise, it may be better to remove time estimation from this manuscript.

>>R: Many thanks for introducing the Gondwana hypothesis into this study, which has substantially enhanced the significance of this study. Regarding the controversies surrounding this topic, we have referred to these previous studies and made a throughout revision in the introduction and discussion sections. Please refer to our responses to the first major comment.

Following your suggestions, we identified 12 dipterocarp genomes from Ng et al. 2021, Wang et al. 2022 and Tian et al. 2022, and included them all in our phylogenomic analysis. Furthermore, we calibrated the phylogenetic tree using the fossil points from Bansal et al. and the calibration provided by Ng et al. Both calibrations yielded consistent estimates of divergent times, supporting the Gondwana origin hypothesis. Please refer to the revised Fig. 2a and Supplementary Fig. 4a attached below. We have also updated the methods and results selections (Please also see below).

We would like to note that the gene family expansion/contraction analysis re-run using the new dating, and the results were largely consistent with the previous version (Fig. 4e; Supplementary Fig. 6), which revealed one expanded and 36 contracted gene

families in the comparison with Malvaceae species, and the comparison with temperate trees identified three expanded and 53 contracted gene families.

We revised the maintext as: To investigate the phylogenetic relationships among Dipterocarpoideae species and their divergence with the closest family Malvaceae, we performed phylogenomic analysis including 19 Dipterocarpoideae species and two Malvaceae species. These are seven Dipterocarpoideae species assembled in this study, 12 Dipterocarpoideae species (*Dipterocarpus alatus*, *Dipterocarpus gracilis*, *Dipterocarpus intricatus*, *Dipterocarpus zeylanicus*, *Hopea mollissima*, *Hopea odorata*, *Shorea leprosula*, *Shorea roxburghii*, *Shorea henryana*, *Vatica xishuangbannaensis*, *Vatica odorata* and *Vatica rassak*) from Tian et al.¹³, *Theobroma cacao* and *Gossypium raimondii*^{25, 26}. The maximum likelihood (ML) tree constructed with 918 single-copy orthologs (from 21,109 – 48,040 clustered gene families (Supplementary Fig. 3 and Supplementary Table 9)). We adopted the set of calibrations in Bansal et al.¹⁵, including the divergence time of *Vatica* species (54 MYA), *Shorea* species (54 MYA), and *Dipterocarpus* species (68.5 MYA)) (Fig. 2a). The calibrated tree dated the divergence between Dipterocarpoideae and Malvaceae to 108.0 (95% PHD: 97.4-117.3) MYA, and the earliest divergence within Dipterocarpoideae was estimated to occur at 101.7 (95% PHD: 91.5-115.1) MYA (Fig. 2a). To test the robustness of this estimation, we adopted a different set of calibration in Ng et al.¹⁷ including the divergence time between *T. cacao* and *G. raimondii* (55.8 MYA), and that between *Shorea* and *Dryobalanops* (34 MYA). The calibrated tree revealed the divergent time between Dipterocarpoideae and Malvaceae as 125.7 (95% PHD: 102.1-173.3) MYA) and that among Dipterocarpoideae species being 94.6 (95% PHD: 73.3-128.1) MYA (Supplementary Fig. 4b). As these two calibration sets reveal generally consistent results, our study represents a most comprehensive phylogenomic analysis of Dipterocarpoideae, supporting the origin of Dipterocarpaceae in Western Gondwanaland.

>The details of the methods of the new analyses were not described. No codes were provided, either. Seeing the results, it is strange that similar times were estimated by two analyses using very distinct fossil calibrations. It cannot be excluded that

the analysis may be fine, but for example, it is possible that priors specifying the lower and upper boundaries in the time in the Bayesian phylogenetic analysis may have biased the results. Without the details of the analysis, I cannot be sure if the analysis was appropriate.

>>**R:** Thank you very much for this instructive comment. We have now provided the codes for phylogenomic analyses (https://github.com/Moleculology/Dipterocarpoideae-genome/Supplementary_Code2.mcmctree), and revised the methods to add the details (please see below). The previous setting of the upper boundary of the root was 113 MYA, but this may not be reasonable as it is smaller than the estimation (147.3 (95% PHD: 125.8–159.4) MYA, Tian et al. 2022). We therefore revised to use a more feasible setting of upper boundary as 160 MYA. Under this setting, both calibrations and their combinations yielded consistent results (Revised Supplementary Fig. 4a and Fig. 2a, please also see below).

Revisions in the results:

The time-calibrated phylogeny dated the divergence of Dipterocarpoideae and Malvaceae at 148.7 (95% PHD: 121.9–168.1) MYA, and the split within Dipterocarpoideae was estimated to have occurred at 122.7 (95% PHD: 98.5–151.6) MYA (Fig. 2a). To test the reliability of this estimation, we used different sets of calibrations in independent analyses. The calibrated trees based on the calibrations from either Bansal et al.¹⁷ or Ng et al.¹⁵ revealed very similar divergence estimates between Dipterocarpoideae and Malvaceae (150.7 (95% PHD: 127.8–170.8) MYA and 144.8 (95% PHD: 107.6–169.3) MYA) (Supplementary Fig. 4a). (Lines 173 – 180)

Revisions in the methods:

We used MCMC tree program of PAML v4.7⁹² with correlated molecular clock and JC69 model to estimate divergence time, setting 1,000,000 Markov chain Monte Carlo iterations and a burn-in of 100,000 iterations. The MCMC trees were constructed using different sets of calibrations and their combination. The first set included three calibrations listed in Bansal et al.¹⁷ (the divergence time among *Vatica* species (54 MYA), that among *Shorea* species (54 MYA), and that among *Dipterocarpus* species (68.5 MYA)), and the second set comprised two calibrations in Ng et al.¹⁵ (the

divergence time between *Theobroma cacao* and *Gossypium raimondii* (55.8 MYA), and that between *Shorea* and *Dryobalanops* (34 MYA). Based on the earliest divergence time between Dipterocarpaceae and Malvaceae (147.3 (95% PHD: 125.8–159.4) MYA) reported previously¹³, we set the upper boundary of their divergent time as 160 MYA (RootAge = <1.6 in the mcmctree.ctl file). For the full list of parameter settings, please refer to the code (https://github.com/Molecology/Dipterocarpoideae-genome/Supplementary_Code2.mcmctree). (Lines 659 – 671)

Revised Supplementary Fig. 4a

Revised Fig. 2a

a

5. Cherry-picking of genes under selection

Although the authors said "GO enrichment analysis" (line 174), I could not find any statistics. Rather, it seems that the authors arbitrarily picked up genes relevant to UV, oxidative stress, and whatever they want to discuss. The authors can find standard GO enrichment analysis in many genome papers. "Temperate tree comparison" or "Malvaceae comparison" does not seem to make sense because dipterocarp species and Malvaceae species would form a single clade, respectively, and after all, there is only a single comparison. If this should be retained, the validity of the methods should be explained in detail. Moreover, their hypothesis in line 62 "At the emergent layer, these species are challenged with intensive ultraviolet (UV) radiation and associated oxidative stress. Adaptation to such conditions is thus critical for these species to occupy this niche" does not seem to be well defended. For example, tree species living in forest gaps may be subjected to stronger stress, while the seedlings of canopy species may grow understory. For example, Ng et al. 2021 tested the importance of drought stress using the retained duplicated genes, and here the authors could test it using the signature of selection on each gene. Even if it is supported or not, the test can be interesting.

>>R: Manty thanks for the comments. In the previous version, perhaps we did not make

it clear why we conducted these two comparisons and how we identified these genes. This may lead to some misunderstandings and give the impression that we were cherry-picking these genes. The two comparisons were intended to find genes responsible for the adaptation to the emergent layer of tropical rainforest. We did not report GO or KEGG enrichment results as none terms were significant due to the limited number of total genes. The genes were located systematically by intersecting the functional annotation of comparisons. We have made a thorough revision to clarify these points. Please see below for the related revision in the results section. We hope these revisions convince you that our approach was systematic and unbiased.

In the results, we revised and added:

Since Dipterocarpoideae and Malvaceae species involved were both tropical tree species but occupying different canopy layers, the comparative analysis between them aimed to identify positively selected genes of dipterocarps specifically adapted to the emergent canopy layers. The comparative analysis against temperate tree species was then to identify the positively selected genes adapted to tropical environments. The intersection of these two sets of genes thus would provide robust signatures of positive selection regarding the adaptation of Dipterocarpoideae to the emergent canopy layer of tropical rainforests.

The comparative analysis against Malvaceae identified 171 positively selected genes, whereas that against temperate tree species identified 191 (Supplementary Fig 6). As the number of genes was limited and the corresponding KEGG and GO enrichment analysis yielded no significant results, we presented the functional annotation of GO and KEGG of these genes instead (Supplementary Tables 11 and 12). We found that both comparative analyses annotated functions relating to DNA repair and antioxidation, including Nucleotide excision repair (KEGG), Riboflavin metabolism (KEGG), DNA repair (GO), Double-strand break repair via homologous recombination (GO), Nucleotide-excision repair (GO) and Response to oxidative stress (GO) (Fig. 3a,b; Supplementary Tables 11 and 12). Such a consistent functional annotation arose from different genes identified in either analysis, such as Bloom syndrome protein (BLM) in DNA repair, UDP-sugar pyrophosphorylase (USP) in antioxidation, DNA mismatch

repair protein (MSH6) in DNA repair and Pyridoxine 4-dehydrogenase (PLR1) in antioxidation (Supplementary Tables 13 and 14). More importantly, such a consistent functional annotation also arose from the same genes identified by both analyses, including DNA repair protein RAD51D (in homologous recombination), Structural maintenance of chromosomes protein 5 (SMC5), and riboflavin kinase (RFK) in antioxidation (Fig. 3c, Supplementary Fig. 7, Supplementary Tables 13 and 14).

As you suggested, adaptation to drought may also be an important mechanism for dipterocarps to thrive in tropical environments. This adaptation is not specific to the emergent canopy layer but to the overall tropical environments. In our comparison with temperate tree species, we have identified two key candidate genes associated with this adaptation. Please see below for the related revision in the results section.

In the comparative analysis against temperate tree species, we identified two positively selected genes (O-succinylbenzoate-CoA ligase (AEE14) and L-ascorbate peroxidase relevant to the plants' response to drought stress (Supplementary Table 13), which is involved in KEGG pathways of ubiquinone and other terpenoid-quinone biosynthesis (map00130) and glutathione metabolism (map00480), respectively. The finding of these positively selected genes is consistent with a previous study indicating that dipterocarps have adapted to the rare and irregular drought events in tropical environments¹⁷.

>I do not think the authors answered the point. Diverse genes were identified by the selection analysis. Although the lack of enrichment in Gene Ontology analysis or any other analysis, the authors picked up several genes relevant to UV responses. Seeing the same data, other authors can make their own stories. This is not considered objective.

>>R: We highly appreciate this thoughtful comment. We have now understood that we had arbitrarily selected genes and established their associations with UV adaptation. Therefore, we must increase the objectivity in identifying genes under selection. To achieve this, we intersected the positively selected genes from the two comparisons and found 37 genes (these are listed in Supplementary Table 13), four of which were related to the stress response according to previous studies. Moreover, we also have made

efforts to improve the objectivity in interpreting our results. As Dipterocarps experience a range of stresses, including UV and drought stresses, the positively selected genes should be interpreted as adaptation to such a broad range of stressors. To achieve both ends, we made a thorough revision in the abstract, introduction, results and discussion.

In the abstract, we revised the corresponding interpretation as:

Several positively selected genes were involved in antioxidation and DNA repair functions, likely facilitating adaptation to environmental stresses in Asian rainforests.

(Lines 44 – 46)

In the introduction, we added:

As a consequence of above-canopy life in Asian rainforests, these species are challenged with various abiotic stresses such as irregular droughts¹⁵ and intensive ultraviolet (UV) radiation¹⁸. (Lines 83 – 84)

In the results, we revised the corresponding part as:

There were 37 positively selected genes supported by both comparisons, among which four genes were found to be associated with plants response to environmental stresses (Supplementary Table 13). These four genes included DNA repair protein RAD51D (in homologous recombination *RAD51*), structural maintenance of chromosomes protein 5 (*SMC5*), and carotenoid ϵ -hydroxylase (*LUTI*) involved in DNA repair, and riboflavin kinase (*RFK*) involved in antioxidation (Fig. 3c, Supplementary Fig. 6 and 7, Supplementary Table 13). *RAD51*²⁷ and *SMC5*²⁸ are found to be associated with UV adaptation, and *LUTI*²⁹ and *RFK*³⁰ have been reported to be linked with drought tolerance. Moreover, according to the functional annotation in DNA repair and antioxidation, we identified several additional genes in either analysis, such as Bloom syndrome protein (*BLM*) and DNA mismatch repair protein (*MSH6*) in DNA repair, and UDP-sugar pyrophosphorylase (*USP*) and pyridoxine 4-dehydrogenase (*PLRI*) in antioxidation (Supplementary Table 14). (Lines 217 – 228)

Given that DNA repair and antioxidation are involved in the responses of plants to most environmental stresses^{18, 31}, these results effectively revealed the molecular footprints likely contributing to the adaptation of dipterocarps to environmental stresses in tropical rainforests. (Lines 235 – 238)

In the discussion, we revised the corresponding part as:

Dipterocarpoideae species benefit greatly from the height of their adult trees, by which they preempt light resources and facilitate pollen and samara dispersal^{16, 46}, but these advantages come with the costs, such as direct exposure of adult trees to intense UV and high vulnerability to droughts. Given the positively selected genes supported by both comparisons are functionally associated with these stresses²⁷⁻³⁰, our results offer the molecular basis for further exploring the adaptation of Dipterocarpoideae to environmental stresses in Asian rainforests. (Lines 329 – 334)

6. data availability

Data availability section should state more specifically the availability of the assemblies and annotation files.

>>R: We have revised the data availability statement as follows:

All data used in the analysis of this study have been deposited in the CNSA (<https://db.cngb.org/cnsa/>) in the BIG Data Center, Beijing Institute of Genomics (BIG), Chinese Academy of Sciences, BioProject: ID: PRJCA017262.

>This is fine, but as noted at the beginning, code availability is missing although the Editorial Check List said "We have provided a full code availability statement in the manuscript". Reproducibility is important.

>>R: Many thanks for addressing the importance of reproducibility of our results. We have provided the codes of bioinformatic analysis, which are available online (see Methods and the table below).

Software	Code source
RAxML v8.0.19	https://github.com/Moleculology/Dipterocarpoideae-genome/Supplementary_Code1.RAxML.sh
MCMCtree in PAML	https://github.com/Moleculology/Dipterocarpoideae-genome/Supplementary_Code2.mcmctree
KaKs_Calculator v2	https://github.com/Moleculology/Dipterocarpoideae-genome/Supplementary_Code3.KaKs_Calculator
QUOTA-ALIGN script	https://github.com/Moleculology/Dipterocarpoideae-genome/Supplementary_Code4.QUOTA-ALIGN

RGAugury pipeline	https://github.com/Molecology/Dipterocarpoideae-genome/Supplementary_Code5.RGAugury.sh
Sequencing mapping and variant calling	https://github.com/Molecology/Dipterocarpoideae-genome/Supplementary_Code6.mapping.varcall.sh
SMC++ v 1.15.2	https://github.com/Molecology/Dipterocarpoideae-genome/Supplementary_Code7.SMC.sh
PSMC v0.6.4	https://github.com/Molecology/Dipterocarpoideae-genome/Supplementary_Code8.PSMC.sh
PROVEAN v1.1.5	https://github.com/Molecology/Dipterocarpoideae-genome/Supplementary_Code9.proven.sh
SIFT4G v2.0.0	https://github.com/Molecology/Dipterocarpoideae-genome/Supplementary_Code10.SIFT.sh
Derived alleles identification	https://github.com/Molecology/Dipterocarpoideae-genome/Supplementary_Code11.derived
Runs of homozygosity in PLINK	https://github.com/Molecology/Dipterocarpoideae-genome/Supplementary_Code12.ROH.sh

Minor issues

line 184, which is involved

>>R: Revised. Line 242.

>**Fine.**

>>R: Thanks.

Reviewer #2 (Remarks to the Author):

Comments for authors

NCOMMS-22-41572A: Liu et al. Life up high: the adaptations and demography of the Asian rainforest titans, Dipteroocarpoideae.

The authors have carefully taken up all my comments and satisfactorily addressed some of them. Really appreciate the effort for trying to improve the manuscript.

However, I have several comments on the revised manuscript:

>>R: Thank you very much for your positive comments and constructive suggestions.

We have revised the manuscript according to your suggestions and made our point-by-point response. We hope you find that the solidity of this study has been further improved after these revisions. All revised contents are highlighted in green.

Page 6&7, lines 119-124: The histograms obtained from ploidy screening using the DNA flow cytometry were not really convincing particularly showing broad DNA peak rather than single prominent DNA peak. Even the standard reference DNA peak was broad. Perhaps this may be due to the handling error during samples chopping/preparation. I strongly encourage this to be addressed.

>>R: Many thanks for this constructive suggestion, and we have addressed this in the legend of Supplementary Fig.1, as “The histograms show broad DNA peaks for both tetraploid *Hopea* species and the reference species, probably due to some handling errors during sample chopping/preparation.”

Page 15: lines 319-320: “Is there sufficient intraspecific genetic variation within species to keep pace with the ongoing environmental changes?” The authors discussed it in Page 18: lines 385-391, does this really answer the question above?

>>R: Thank you very much for this constructive comment. The evidence of this study could not fully answer this question so far. Therefore, we revised this question listed at the beginning of the discussion as “Exploring the processes underlying adaptive diversification and population demography can provide critical insights into the conservation of endangered species^{3, 38, 39}.” (Lines 300 – 301)

Page 15: lines 324-325: “has resolved the previous debate on the origin of this clade” is a very bold statement, perhaps the jury is still out there. Hence, I would suggest that this statement be revised to, “The comprehensive comparative genomic analysis further support the Gondwanan origin of Dipterocarpoideae species”

>>R: Thanks. This sentence is revised accordingly. Lines 307–308.

Page 16: lines 348-351: This is crucial. The natural distribution of *H. hainanensis* only can be found in Hainan province and northern part of Vietnam. They suggested that previously reported genome of *H. hainanensis* by Wang et al. 2022 was ‘probably’ from diploid population. Where is the evidence of diploid populations?

>>**R**: Apologies for the confusion. The intended meaning previously was: *H. hainanensis* may have both tetraploid and diploid populations. We adopted the “may” to indicate this was highly speculative (Please see our reply below). The sequenced individual by Wang et al. (2022) was collected from Ruili Botanical Garden (Yunnan, China), and the field origin of this sample remains unknown. To avoid misunderstandings, we have removed the sentence indicating the possibility of both tetraploid and diploid populations.

If both tetraploid and diploid populations do exist, how do they know the 30 *H. hainanensis* individuals used for resequencing study were all autotetraploid individuals (minus 4 individuals tested for kmer analysis)?

>>**R**: It is a nice idea to validate that all individuals are tetraploid by *k*-mer analysis. Accordingly, we conducted *k*-mer analysis for the remaining 26 samples, and they are indeed all tetraploid, with one sample (JF4) difficult to judge. Except this sample, the depth of the common peak on the right was four times that of the heterozygous peak (Please refer to the figure below), supporting that they are tetraploid. It appears that the tetraploid pattern is only apparent with extremely deep sequencing (Supplementary Fig.1a), relatively clear with deep sequencing (Supplementary Fig.1o-r, showing individuals with the highest resequencing depth), and almost invisible at low depths (Supplementary Fig. 1 in Wang et al. 2022). Supplementary Fig. 1 in Wang et al. (2022) only had one peak (possibly due to low sequencing depth), providing no clear support of diploid or tetraploid.

Supplementary Fig. 1a, o-r in our study

Supplementary Fig. 1 in Wang et al. 2022

In addition, why the suspected diploid genome of *H. hainanensis* (Wang et al 2022) were not included in the analysis?

>>**R:** Through comparisons of *k*-mer analysis, the ploidy of *H. hainanensis* in Wang et al. (2022) is not clear due to the low depth of survey data. We therefore did not include this suspected diploid genome in the analysis.

REVIEWER COMMENTS

Reviewer #1 (Remarks to the Author):

New comments are shown by >>>.

Reviewer #1 (Remarks to the Author):

General comments

The comments by reviewer 2 were highly similar to mine. The authors provided additional meaningful experimental data using flow cytometry. However, in bioinformatic analysis, many additional analyses were superficial and did not address the fundamental theoretical issues in phylogenetic and population genetic analysis. The methods are so short that I cannot judge if the analyses were appropriate. Alternatively, code can be informative on the methods, but the main text does not have a statement of code availability, although the Editorial Check List said "We have provided a full code availability statement in the manuscript". The authors should consider reproducibility seriously.

>>R: Many thanks for the constructive suggestions in both rounds of review, and for the positive comments on our previous reply. In this revision, we have improved methodological details, and in particular, we have provided the analysis codes and made them publicly available. Concerning the genomic analysis of polyploid species, we fully agree with the reviewer that this field is challenged by the lack of generalized methodologies. Our study, representing the first genome assembly of wild autotetraploid species (previous autotetraploid assemblies were sugarcane (Zhang et al. *Nature Genetics*, 50: 1565–1573, 2018), alfalfa (Chen et al. *Nature Communications*, 11: 2494, 2020), and potato (Sun et al. *Nature Genetics*, 54: 342–348, 2022)), has made various attempts to explore methodology following your constructive suggestions, and we hope to promote the research on polyploid genomes together with the reviewers. Please note that the comments are bold in black, and our responses are highlighted in blue, with the revised contents are in green. The contents from the previous round of revision are in grey.

My comments on each point will be shown by >.

>>R: We highlight these comments in bold to separate them from the previous round of revision, and our responses are in blue.

Response to reviewers' comments

Reviewer #1 (Remarks to the Author):

Dipterocarp trees are dominant in Southeast Asian tropical forests and important for environments, forestry, and medicine. Recently, a few genomic studies reactivated long-standing debates about its evolutionary history centered on its Gondwanan vicariance hypothesis. The authors reported chromosome-level assemblies of seven dipterocarp tree species. Two of them are autotetraploid tree species, which can be a unique resource to study plant evolution in general. However, a major revision would be necessary to place the study in the context of previous research.

>>R: Many thanks for your positive comments and your constructive suggestions. Following your suggestions, we have revised the manuscript thoroughly; please find the point-to-point reply below. We hope you will find that novelty and the significance of this study have been substantially strengthened.

1. The genome assemblies of dipterocarps, i.e., *Shorea leprosula* (Ng et al. *Commun Biol* 7:1166, 2021) with transcriptome and resequencing of other dipterocarps, *Dipterocarpus turbinatus* and *Hopea hainanensis* (Wang et al. *Plant Biotechnol* 20:538, 2022, Epub 2021 Dec), were already reported some time ago. Many results of this manuscript, such as ancient polyploidy and time estimation, were reported. The same species, *Hopea hainanensis*, was sequenced before. The current manuscript would give a false impression of novelty. This is not scientifically proper. Rather, I would suggest that the comparison with previous studies and the introduction explaining unsolved issues would highlight the importance of this new study as explained below. Their report should be integrated fully in the introduction and discussion. Furthermore, Tian et al. (*Plant Communications* 2022, 100464, 2022) reported 13 dipterocarp genomes in October 2022. I am not sure if this is before or after the submission of this manuscript and may depend on the journal policy, but integrating it would further strengthen this manuscript.

>>R: Many thanks for your suggestions and the references. Indeed, we agree that it is important

to acknowledge the previous discoveries. Following your suggestions, we have highlighted the contributions of Ng et al. 2021, Wang et al. 2022, and Tian et al. 2022 while introducing the debates surrounding the origin of the Dipterocarpaceae. We have rephrased the paragraphs in the introduction and discussion sections (The revision is highlighted in blue in the main text, please also see below).

In the introduction, we added:

Earlier studies proposed that Dipterocarpaceae originated in Western Gondwanaland in Early Cretaceous (c.120 million years ago (MYA)), before the separation of Africa and South America¹⁴. Recent evidence from pollen fossils¹⁵ and a phylogenomic study¹³ supported such a Western Gondwanaland origin, albeit with a slightly recent dating (c.102.9 MYA¹⁵ and 105.0 MYA¹³). These studies suggested that Dipterocarpaceae dispersed to India during the mid-Cretaceous and subsequent to Southeast Asia facilitated by India-Asia collision. However, genomic studies involving different sets of dipterocarp species revealed a much more recent divergence between Dipterocarpaceae and its closest family Malvaceae (c. 84 MYA¹⁶ and 86–98 MYA¹⁷). Therefore, there is still a controversy surrounding the evolution of these keystone rainforest species; resolving it will provide valuable insights into the origins of megathermal angiosperms.

In the discussion, we added:

Here, the phylogenetic history constructed using a total of 19 Dipterocarpoideae genomes agrees with the results from the early view¹⁴ and some recent phylogenomic/phylogenetic studies^{13, 15}, thus supporting the Western Gondwanaland origin of Dipterocarpaceae. Different calibrations yielded consistent results, indicating this estimation of divergent time is robust. This finding is largely consistent with the evolutionary history of angiosperms revealed by genomic data³⁵ and new fossil evidence³⁶, showing far earlier splits between major clades of angiosperms (occurring at the Jurassic and Lower Cretaceous) than traditionally expected.

Furthermore, as you suggested, we have integrated all published Dipterocarpaceae genomes, including those in Tian et al. 2022 in the revised phylogenomic analysis, and thus form a dataset of 19 Dipterocarp genomes. These suggestions significantly improve the novelty and solidity of our study. Please refer to the response for this specific issue below.

>The new paragraphs partly solved the lack of introduction. However, other important conclusions of the manuscript include genome duplication. A more thorough description of previous publications is necessary.

>>R: Thanks. According to your suggestions, we have now included a more thorough description of previous findings, including the genome duplication event and others.

In the introduction, we added:

Recent genomic studies^{13, 14, 15} found a whole genome duplication (WGD) event preceding the divergence of Dipterocarpoideae species. This WGD event increased the chromosome number of Dipterocarpoideae ancestors to 12, and the following two steps of karyotype evolution gave rise to 11-chromosome species (Dipterocarpeae tribe) and 9-chromosome species (Shoreeae tribe)¹³. (Lines 64 – 68)

>>>This is fine.

2. Novelty in autopolyploidy

Wang et al. reported the assembly of the same species *Hopea hainanensis* but they did not mention autotetraploidy at all and treated it as a normal diploid species. This manuscript provided chromosome count (Supplementary Figure 1) and careful k-mer analysis to provide strong evidence of polyploidy. Furthermore, they seem to have successfully provided phased four assemblies, although further quality checks in this aspect should be reported. In general, the genome assembly of autopolyploid species remains a major challenge in genomics. For example, the assembly of autotetraploid *Arabidopsis arenosa* is difficult, and instead, self-fertilizing allotetraploid *A. suecica* derived from *A. arenosa* was sequenced instead (Burns et al. *Nature Ecology and Evolution*, 10:1367, 2021; Jiang et al. *Nature Ecology and Evolution*, 10:1382, 2021). Supposing the quality of the phasing is quantified, the new assembly of *H. hainanensis* would provide interesting insights into the autopolyploid genome.

To support the tetraploidy, please add the empirical data of genome size measured by flow cytometry. Ng et al. (*Plant Ecology and Diversity* 9:437, 2016) explained a standard method of dipterocarps.

>>R: Many thanks for acknowledging the significance of our tetraploid assembly and for the references. The autotetraploid assembly presented a major challenge until recently. Zhang et al.

2018 Nat Genet provided the first allele-aware assembly of sugarcane and corresponding methods were published by Zhang et al. 2019 Nat Plants. We followed their approach and maintained close communication with the authors since we started the project in 2019, and these efforts resulted in the assembly of comparable quality to Zhang et al. 2018 Nat Genet. We also provided evidence showing the quality of the two autotetraploid genomes (Lines 133-140, 144-147, and 150-154; Supplementary Tables 3-5). So far, the allele-aware genome assembly of autotetraploid was still limited, in particular for wild species, and we hope that our work will inspire further involvement in studies of Dipterocarpaceae and wild tetraploid species.

Furthermore, we highly appreciate the methods you provided. We conducted flow cytometry analysis and estimated the C-values accordingly. The corresponding results have been added to Figure S1 and Table S1 (please also see below), providing robust evidence in support of tetraploidization. We have also revised the results section accordingly.

In the results, we added:

Moreover, by using *Solanum lycopersicum* ($2C = 2.12$) as a reference, the flow cytometry results revealed the C-values of *H. chinensis*, *H. reticulata* and *H. hainanensis* are $2C$ equal to 0.63, 1.36 and 1.66 (Supplementary Fig. 1j-l and Supplementary Table 1). Compared to the diploid *H. chinensis*, the cytometry analysis identified a clear peak confirming the tetraploidy of these two species (Supplementary Fig. 1m-n).

In the methods, we added:

To further test whether *H. hainanensis* and *H. reticulata* are tetraploid species, we estimated their genome sizes and ploidy using flow cytometry. For these two species and *H. chinensis* (a diploid congener), we collected the fresh young leaves from the sampled trees used for the de novo genome assembly. We also sampled young leaves of *Solanum lycopersicum* (genome size: 2.07 Gb; $2C = 2.12$ pg53) as the reference for genome size estimation. We first estimated the genome sizes and $2C$ values of the three *Hopea* species by measuring the fluorescence of nuclei from the mixture of each *Hopea* species and *S. lycopersicum* and those from each single species. Then, we assessed the ploidy for *H. hainanensis* and *H. reticulata* based on their fluorescence of nuclei setting *H. chinensis* as the diploid reference. The nuclei suspension was prepared for each species following the methods described in Ng et al. 54. We measured the fluorescence from the nuclei suspension using a Sysmex CyFlow Cube6 flow cytometer, and data analysis was performed using FCSExpress 7plus.

>Flow cytometry data are satisfactory. This also solved a similar issue raised by Reviewer 2.

>>R: Thanks.

The methods to analyze the demographic analysis of the autotetraploid need to be thoroughly revised. As far as reading the Methods, software developed for diploid species is applied to the data. However, many assumptions are not filled. Please refer to the population genetic analysis of autotetraploid *A. arenosa* (Monnahan et al. Nature Ecol Evol 3: 457, 2019) and theoretical studies cited there (particularly Otto and Whitton, Ann Rev Genet 34:401, 2000). Here I list a few examples below but the authors should investigate the effect of autopolyploidy thoroughly. As far as I understand, SMC++ and PSMC assume diploid inheritance like a human being. Intuitively, it may be usable if you randomly choose two haplotypes, but a thorough theoretical basis should be explained if it would be included in the revised manuscript.

Furthermore, the generation time of 10 and 15 years (line 741) are very likely minimum estimates, because the trees will produce seeds for a long until a gap may be formed in the forest or other reasons. Relative time estimation of N_e would be interesting but the usage of absolute time to compare the demography with glacial cycles should be omitted.

>>R: Thank you for your suggestions, especially those regarding tetraploid analysis. We may not have fully understood the methods presented in Monnahan et al. (2019) Nature Ecol Evol 3: 457, as they did not seem to conduct a demographic analysis using those tools. Instead, we appreciate your suggestion of extracting and analyzing two haplotypes. This approach treats the tetraploid genome as a diploid genome and aligns with the model assumptions. Accordingly, we extracted the two longest monoploid genomes for analysis (Reviewer 2 recommended this approach as it would include the most genomic information). Furthermore, we also adjusted the generation times to 15, 20, and 30 according to your suggestions.

>The newly added method is just a single sentence without a reference.

Line 779 "To meet the standards of population demography analyses (using SMC++ and PSMC), we mapped the filtered reads to the longest two monoploid genomes of each species to form an approximation of diploidy."

This is far too obscure. Furthermore, no codes were provided. What is the motivation to map reads to two monoploid genomes? Then, most genes have two duplicated copies. Mapping to duplicated genes is a complex issue. As a reviewer, I cannot be sure if the analysis was appropriate.

>>R: This suggestion is gratefully accepted. The codes of PSMC (https://github.com/Molecology/Dipterocarpoideae-genome/Supplementary_Code8.PSMC.sh) and SMC++ (https://github.com/Molecology/Dipterocarpoideae-genome/Supplementary_Code7.SMC.sh) have been provided and the reference was added. Indeed, mapping to duplicated genomes is complex, and to ensure reliable demography analysis, we only included the uniquely mapped reads (BWA mapping quality > 0) in the analysis. In the PSMC analysis, we set -Q 20 in vcf2fq to indicate the minimum mapping quality is 20, and used the default --minimum-mapping-quality 10 in the GATK variant calling (https://github.com/Molecology/Dipterocarpoideae-genome/Supplementary_Code6.mapping.varcall.sh) whose output .vcf was input for SMC++ analysis. We also added details of these parameters in the methods (Please see below).

Concerning the motivation to map reads to two monoploid genomes, we are inspired by your recommended references, i.e. the empirical study (Monnahan et al. *Nature Ecol Evol* 3: 457, 2019) and the theoretical study (Otto and Whitton, *Ann Rev Genet* 34:401, 2000). These studies indicate the general methodology for polyploid genomics is currently lacking, and it is necessary to adopt novel approaches. Therefore, we decided to carry out the attempt using two haplotype genomes other than the traditional approach using only one haplotype genome, despite that the reliability of this method requires further investigation due to the lack of precedents. In the methods, we added a paragraph to interpret this motivation and cite the references.

The challenge of demographic analysis for autopolyploid species is mainly caused by the lack of generalized methodologies^{48, 108}. To date, the algorithms and software (e.g. SMC++³⁶ and PSMC³⁵) developed for effective population size analysis are restricted to diploid analysis. To meet the requirements of these analyses, we made an attempt to treat the autotetraploid species as diploids by using the longest two monoploid genomes as the reference genome. (Lines 776 – 780) We estimated the recent-past effective population sizes (N_e) of *H. hainanensis* and *H. reticulata* separately using the coalescent-based inferences in SMC++ v 1.15.236. The VCF files, including 30 *H. hainanensis* samples and 32 *H. reticulata* samples, were converted to the SMC++ input files using the 'vcf2smc' command. The estimate was run with generation times of 15, 20 and 30 separately and the corresponding estimated mutation rates (Supplementary Table 20). Codes are available at https://github.com/Molecology/Dipterocarpoideae-genome/Supplementary_Code7.SMC.sh. (Lines 781 – 787)

In addition, we inferred historical changes in N_e of *H. hainanensis* and *H. reticulata* using PSMC v0.6.435. For each of 30 *H. hainanensis* samples and 32 *H. reticulata* samples, we performed the analysis with generation times of 15, 20 and 30 and the corresponding estimated mutation rates separately (Supplementary Table 20). The whole-genome consensus sequence was generated by SAMTOOLS v1.3, bcftools, and vcfutils.pl, setting the minimum read depth to 10, maximum read depth to 100, and minimum mapping quality 20. We ran PSMC with parameters (psmc -N30 -t15 -r5 -p 4+25*2+4+6) (https://github.com/Molecology/Dipterocarpoideae-genome/Supplementary_Code8.PSMC.sh). (Lines 788 – 795)

>>>I cannot be sure that the results of these analyses are interpretable. PSMC or SMC++ were developed for diploid species. When a particular algorithm is applied to unintended type of datasets, simulation or theoretical validation (or citation to previous validation) is necessary. The authors said: "despite that the reliability of this method requires further investigation due to the lack of precedents". However, it is not a future issue, but it must be provided here.

I can imagine many pitfalls. As far as I understand, both in PSMC and SMC++, alleles should be randomly chosen. It is unclear the usage of two haplotype genomes can accurately estimate four alleles. Suppose three alleles are similar and one allele is divergent in the individual of reference genome. This would lead to two assembled haplotypes. If the sequence reads of the same species is mapped, one allele would show very low polymorphism, and another with erroneously high level of polymorphisms due to the mixture of three alleles. In other situations, four alleles can be assembled into two and two. Then, either of the two are biased to be similar. This will affect the N_e estimates. These are only a few scenarios. I cannot be sure about the analysis. Potentially theoreticians may be able to evaluate or support that the analyses.

Due to this reason, I cannot agree with a major sentence in the abstract lines 46-48

"Resequencing of two endangered species showed an expansion of effective population size after

the last glacial period and a recent sharp decline coincidental with the history of local human activities."

Surprisingly, different generation times and methods (SMC++ and PSMC) yielded highly consistent results. Please see below for the revised Figure 4a and Supplementary Figure 11. The results section was also revised thoroughly (please also see below). It is indeed a challenge to provide a clear theoretical explanation for the mechanism behind these robust results. Intuitively, your suggestion that treating the tetraploid genome as diploid to fit the model assumption likely contributes significantly to the robustness of the results. Many thanks again for the suggestion. (Discussion about Otto and Whitton, 2000; please see the reply below)

We revised the results as:

The demographic history for both species was performed using SMC++30 (see Methods), based on generation times of 15, 20, and 30 years (Supplementary Table 21). We found that the effective population sizes of both species increased after the Last Glacial Maximum (LGM, c. 19,000 years ago), and reached a maximum around 6,000 – 10,000 years ago for *H. hainanensis* and 3,000 – 9,000 years ago for *H. reticulata* (Fig. 4a). However, both species experienced a sharp decline recently around 1,500 – 2,400 years ago for *H. hainanensis* and 1,200 – 2,400 years ago for *H. reticulata*. During this period, effective population sizes decreased from the maximum (c. 5.0×10^6 for *H. hainanensis* and c. 4.0×10^6 – 6.0×10^6 for *H. reticulata*) to less than 1,000 (Fig. 4a). The demographic analysis with PSMC31 showed consistent results (Supplementary Fig. 11). Since human activities in Hainan Island can be traced back to 3000 years ago³², the above evidence suggests that human intervention may have disrupted the maintenance of their populations after the glacial period, thus serving as a primary major factor contributing to species endangerment. >Seeing the Figures, I do not understand why the authors stated "The demographic analysis with PSMC31 showed consistent results". As far as I saw, they were different in the important aspects. In the SMC analysis of *H. hainanensis* (left, top), the population size increased after the band of LGM, then drop. However, in the PSMC analysis, the vast majority of the lines did not show an increase. A subsequent drop was not observed, either. I wonder if the authors carefully examined the results. By the way, it is well known that the estimation in very recent times using SMC and PSMC may not be reliable, so should not be interpreted much.

>>R: Many thanks for the suggestions. Indeed, PSMC analysis is not reliable within the last ten thousand years (Li H, and Durbin R, Nature, 475: 493–496 2011). Although SMC appears to provide accurate estimates across a time span of three orders of magnitude (103–106 years ago), it is not reliable within the last thousand years (Terhorst et al. Nature Genetics, 49: 303–312, 2017). Thus, for the period between the last ten thousand and one thousand years, corresponding to after the Last Glacial Maximum (LGM), it seems only SMC provided relatively reliable results. In the revision, we removed the comparison between the two approaches, and interpreted the results only on their reliable ranges.

In the results, we added:

The demographic analysis with PSMC35 showed a minor declining trend in effective population sizes (N_e) for both *H. hainanensis* and *H. reticulata* from one hundred thousand years ago and continuing throughout Last Glacial Maximum (LGM, c. 19,000 years ago) (Supplementary Fig. 10). For more recent demography, the SMC++36 analysis based on three different generation times (15, 20, and 30 years; Supplementary Table 20) revealed that the N_e of both species increased after the LGM, reaching a maximum six to ten thousand years ago for *H. hainanensis* and three to nine thousand years ago for *H. reticulata* (Fig. 4a). (Lines 263 – 269)

>>The point itself is fine, but the mapping problem in the last item must be solved before this analysis.

In line 239, a comparison with cherry-picking several tree species does not make sense and so should be omitted. In line 221, Tajima's D, CLR, and iHS are not designed for autopolyploid species, although they can work for allopolyploid species.

>>R: Thanks. Accordingly, we have removed the relevant descriptions of genetic diversity of several tree species from the main text. Indeed, Tajima's D, CLR, and iHS are not designed for autopolyploid species. Monnahan et al. (2019) and Zhang et al. (2018) converted the tetraploid data into diploid data by mapping against one monoploid genome and considering two-allelic sites only. Our previous approach followed their methods. However, their analyses mainly focused on

neutral processes and did not include selection analysis. Therefore, there is currently no reference for population genomic analysis regarding the signature of selection in autopolyploids.

To explore the effect of autopolyploidy and determine appropriate methods, we followed your previous advice and re-ran the entire population genomics analysis, including structure, selection analysis, and ROH, using the two longest monoloids as templates for mapping. This approach treats the tetraploid genome as a diploid genome (referred to as the diploid model analysis). To facilitate comparison, we have compiled all the results together (please see the figure below). In the structure analysis of the neutral process, we found consistent results between the two methods. However, in the selection analysis, the diploid model analysis identified a greater number of positively selected genes than the tetraploid model. Otto and Whitton, 2000 predicted that the efficiency of positive selection is reduced in polyploids, particularly when the mutation is not dominant. Our observation is consistent with their prediction that beneficial mutations are less efficient in reaching high frequencies in polyploid, leading to less prominent positive selection signatures in the tetraploid model analysis. On the other hand, having four alleles per locus (a characteristic of autotetraploidy) increased the genomic heterozygosity. Therefore, although the previous method of mapping against the longest monoloid genome was not perfect, it remains the optimal solution. We retained our previous methods in this manuscript. Thank you for the fruitful discussion, which has greatly enhanced our understanding of the theory and analysis on tetraploid species.

>I do not think the authors addressed the point. A deep understanding of population genetics is lacking.

>>R: We misunderstood your suggestion previously. We have now realized that your suggestion was to omit analysis of Tajima's D, CLR, and iHS, and delete the results. In this revision, we revised it accordingly.

>>>Fine.

3. Ancient polyploidy was already reported

Ng et al. (2021) and Wang et al. (2022) reported a genome duplication event preceding the divergence of Dipterocarpoideae species. The authors did find it out, but the original paper should be cited. Otherwise, it will give a false impression of novelty.

>>R: Many thanks. We agree it is critically important to acknowledge previous findings. We therefore referred to these excellent studies in the results and discussion, indicating our findings were consistent with them.

In the results, we revised and added:

The genetic distance of paralogs within these species was found to be greater than the distance of orthologs between pairs of species, indicating that WGD-1 occurred before the diversification of Dipterocarpoideae species (Fig. 2b and Supplementary Fig. 4). This finding is consistent with previous genomic studies^{13, 16, 17}, suggesting that WGD facilitated the diversification of Dipterocarpoideae species.

In the discussion, we revised and added:

Detection of WGD-1 and WGD-2 probably reflect a genome-scale adaptation to ancient climate/environmental changes. We found that the WGD-1 preceded the diversification of Dipterocarpoideae species, which is consistent with several previously published studies on genomes in this clade^{13, 16, 17}.

>The content is fine but please add this to the introduction as mentioned above.

>>R: Thanks. This is added in the introduction as:

Recent genomic studies^{13, 14, 15} found a whole genome duplication (WGD) event preceding the divergence of Dipterocarpoideae species. This WGD event increased the chromosome number of Dipterocarpoideae ancestors to 12, and the following two steps of karyotype evolution gave rise to 11-chromosome species (Dipterocarpeae tribe) and 9-chromosome species (Shoreeae tribe)¹³. (Lines 64 – 68)

>>>Fine.

4. Time estimation

There has been a long dispute on when Dipterocarpaceae/Dipterocarpoideae originated (see references cited in the literature below). Traditionally, the Gondwanan vicariance hypothesis based on broad geographic distribution proposed an old origin of >120 million years ago. However, the

estimates are variable among phylogenetic studies, and most importantly, the Gondwanan hypothesis contradicts the general time scale of angiosperm evolution. Ng et al. conducted a detailed analysis using the genome data and fossil calibration and reported much younger divergence. In 2022, new microfossils of dipterocarps were reported, which can push the divergence time even younger (Bansal et al. *Science* 375:455, 2022). The issue is currently unsolved and can be potentially a major contribution to this manuscript. The authors cited Bansal et al. 2022 but did not discuss the dispute at all.

I should note that the methodologies the author used are primitive. The authors should examine the complexity and many assumptions in the time estimation. For calibration, authors took values from the TimeTree webpage, but this does not seem sophisticated. Particularly, little data on dipterocarp and its relatives were available. For example, when "Hopea" and *Theobroma cacao* was filled in on its webpage, the outcome was 80 MYA (CI 38.7-85.0 MYA). In line 139, the authors reported c. 68.2 MYA without providing a confidence interval. This illustrates the insufficient quality of the current analysis. At the moment, it would be wise to use two different ways of fossil calibration, the new ones by Bansal et al., and traditional ones used by Ng et al. Otherwise, it may be better to remove time estimation from this manuscript.

>>R: Many thanks for introducing the Gondwana hypothesis into this study, which has substantially enhanced the significance of this study. Regarding the controversies surrounding this topic, we have referred to these previous studies and made a throughout revision in the introduction and discussion sections. Please refer to our responses to the first major comment. Following your suggestions, we identified 12 dipterocarp genomes from Ng et al. 2021, Wang et al. 2022 and Tian et al. 2022, and included them all in our phylogenomic analysis. Furthermore, we calibrated the phylogenetic tree using the fossil points from Bansal et al. and the calibration provided by Ng et al. Both calibrations yielded consistent estimates of divergent times, supporting the Gondwana origin hypothesis. Please refer to the revised Fig. 2a and Supplementary Fig. 4a attached below. We have also updated the methods and results selections (Please also see below). We would like to note that the gene family expansion/contraction analysis re-run using the new dating, and the results were largely consistent with the previous version (Fig. 4e; Supplementary Fig. 6), which revealed one expanded and 36 contracted gene families in the comparison with Malvaceae species, and the comparison with temperate trees identified three expanded and 53 contracted gene families.

We revised the maintext as: To investigate the phylogenetic relationships among Dipterocarpoideae species and their divergence with the closest family Malvaceae, we performed phylogenomic analysis including 19 Dipterocarpoideae species and two Malvaceae species. These are seven Dipterocarpoideae species assembled in this study, 12 Dipterocarpoideae species (*Dipterocarpus alatus*, *Dipterocarpus gracilis*, *Dipterocarpus intricatus*, *Dipterocarpus zeylanicus*, *Hopea mollissima*, *Hopea odorata*, *Shorea leprosula*, *Shorea roxburghii*, *Shorea henryana*, *Vatica xishuangbannaensis*, *Vatica odorata* and *Vatica rassak*) from Tian et al.¹³, *Theobroma cacao* and *Gossypium raimondii*^{25, 26}. The maximum likelihood (ML) tree constructed with 918 single-copy orthologs (from 21,109 – 48,040 clustered gene families (Supplementary Fig. 3 and Supplementary Table 9)). We adopted the set of calibrations in Bansal et al.¹⁵, including the divergence time of *Vatica* species (54 MYA), *Shorea* species (54 MYA), and *Dipterocarpus* species (68.5 MYA) (Fig. 2a). The calibrated tree dated the divergence between Dipterocarpoideae and Malvaceae to 108.0 (95% PHD: 97.4-117.3) MYA, and the earliest divergence within Dipterocarpoideae was estimated to occur at 101.7 (95% PHD: 91.5-115.1) MYA (Fig. 2a). To test the robustness of this estimation, we adopted a different set of calibration in Ng et al.¹⁷ including the divergence time between *T. cacao* and *G. raimondii* (55.8 MYA), and that between *Shorea* and *Dryobalanops* (34 MYA). The calibrated tree revealed the divergent time between Dipterocarpoideae and Malvaceae as 125.7 (95% PHD: 102.1-173.3) MYA and that among Dipterocarpoideae species being 94.6 (95% PHD: 73.3-128.1) MYA (Supplementary Fig. 4b). As these two calibration sets reveal generally consistent results, our study represents a most comprehensive phylogenomic analysis of Dipterocarpoideae, supporting the origin of Dipterocarpaceae in Western Gondwanaland.

>The details of the methods of the new analyses were not described. No codes were provided, either. Seeing the results, it is strange that similar times were estimated by two analyses using very distinct fossil calibrations. It cannot be excluded that the analysis may be fine, but for example, it is possible that priors specifying the lower and upper boundaries in the time in the Bayesian phylogenetic analysis may have biased the results. Without the details of the analysis, I cannot be sure if the analysis was appropriate.

>>R: Thank you very much for this instructive comment. We have now provided the codes for phylogenomic analyses (https://github.com/Molecology/Dipterocarpoideae-genome/Supplementary_Code2.mcmctree), and revised the methods to add the details (please see below). The previous setting of the upper boundary of the root was 113 MYA, but this may not be reasonable as it is smaller than the estimation (147.3 (95% PHD: 125.8–159.4) MYA, Tian et al. 2022). We therefore revised to use a more feasible setting of upper boundary as 160 MYA. Under this setting, both calibrations and their combinations yielded consistent results (Revised Supplementary Fig. 4a and Fig. 2a, please also see below).

Revisions in the results:

The time-calibrated phylogeny dated the divergence of Dipterocarpoideae and Malvaceae at 148.7 (95% PHD: 121.9–168.1) MYA, and the split within Dipterocarpoideae was estimated to have occurred at 122.7 (95% PHD: 98.5–151.6) MYA (Fig. 2a). To test the reliability of this estimation, we used different sets of calibrations in independent analyses. The calibrated trees based on the calibrations from either Bansal et al.17 or Ng et al.15 revealed very similar divergence estimates between Dipterocarpoideae and Malvaceae (150.7 (95% PHD: 127.8–170.8) MYA and 144.8 (95% PHD: 107.6–169.3) MYA) (Supplementary Fig. 4a). (Lines 173 – 180)

Revisions in the methods:

We used MCMC tree program of PAML v4.792 with correlated molecular clock and JC69 model to estimate divergence time, setting 1,000,000 Markov chain Monte Carlo iterations and a burn-in of 100,000 iterations. The MCMC trees were constructed using different sets of calibrations and their combination. The first set included three calibrations listed in Bansal et al.17 (the divergence time among *Vatica* species (54 MYA), that among *Shorea* species (54 MYA), and that among *Dipterocarpus* species (68.5 MYA)), and the second set comprised two calibrations in Ng et al.15 (the divergence time between *Theobroma cacao* and *Gossypium raimondii* (55.8 MYA), and that between *Shorea* and *Dryobalanops* (34 MYA)). Based on the earliest divergence time between Dipterocarpaceae and Malvaceae (147.3 (95% PHD: 125.8–159.4) MYA) reported previously¹³, we set the upper boundary of their divergent time as 160 MYA (RootAge = <1.6 in the `mcmctree.cti` file). For the full list of parameter settings, please refer to the code (https://github.com/Molecology/Dipterocarpoideae-genome/Supplementary_Code2.mcmctree). (Lines 659 – 671)

Revised Supplementary Fig. 4a

Revised Fig. 2a

>>>Despite additional sentences, my original question "Seeing the results, it is strange that similar times were estimated by two analyses using very distinct fossil calibrations." was not solved. Remembering the first submission, the authors stated oppositely recent estimates of 68.2 MYA of the divergence between Dipterocarpoideae and Malvaceae (line 139).

Seeing the Github files (`tmp_2.tree` and `tmp_3.tree` files), it is unclear how the authors included the calibration points. For example, Ng et al. said "A minimum age of 34 Ma was assigned to the *Shorea*–*Dryobalanops* node based on fossils from the late Eocene attributed to *Shorea*", but the number of 34 was not explicitly found.

I myself am not an expert in phylogenetic analysis using MCMC in PAML. Experts in the field may be able to provide a better evaluation.

The authors did not provide enough background on the discussion on the Gondwanan vicariance hypothesis. Bansal et al. 2022 provided older fossils but did not mention the "Gondwanan vicariance hypothesis", and rather suggested Africa-India floristic interchange. The hypothesis contradicts the timescale of angiosperm evolution as pointed out by Ng et al. 2021.

I cannot give support to a major statement in the abstract (lines 41-43) "The divergence time between Dipterocarpoideae and Malvaceae was estimated to be 148.7 (121.9–168.1) MYA, supporting the Gondwanan vicariance hypothesis."

5. Cherry-picking of genes under selection

Although the authors said "GO enrichment analysis" (line 174), I could not find any statistics. Rather, it seems that the authors arbitrarily picked up genes relevant to UV, oxidative stress, and whatever they want to discuss. The authors can find standard GO enrichment analysis in many genome papers. "Temperate tree comparison" or "Malvaceae comparison" does not seem to make

sense because dipterocarp species and Malvaceae species would form a single clade, respectively, and after all, there is only a single comparison. If this should be retained, the validity of the methods should be explained in detail. Moreover, their hypothesis in line 62 "At the emergent layer, these species are challenged with intensive ultraviolet (UV) radiation and associated oxidative stress. Adaptation to such conditions is thus critical for these species to occupy this niche" does not seem to be well defended. For example, tree species living in forest gaps may be subjected to stronger stress, while the seedlings of canopy species may grow understory. For example, Ng et al. 2021 tested the importance of drought stress using the retained duplicated genes, and here the authors could test it using the signature of selection on each gene. Even if it is supported or not, the test can be interesting.

>>R: Manty thanks for the comments. In the previous version, perhaps we did not make it clear why we conducted these two comparisons and how we identified these genes. This may lead to some misunderstandings and give the impression that we were cherry-picking these genes. The two comparisons were intended to find genes responsible for the adaptation to the emergent layer of tropical rainforest. We did not report GO or KEGG enrichment results as none terms were significant due to the limited number of total genes. The genes were located systematically by intersecting the functional annotation of comparisons. We have made a thorough revision to clarify these points. Please see below for the related revision in the results section. We hope these revisions convince you that our approach was systematic and unbiased.

In the results, we revised and added:

Since Dipterocarpoideae and Malvaceae species involved were both tropical tree species but occupying different canopy layers, the comparative analysis between them aimed to identify positively selected genes of dipterocarps specifically adapted to the emergent canopy layers. The comparative analysis against temperate tree species was then to identify the positively selected genes adapted to tropical environments. The intersection of these two sets of genes thus would provide robust signatures of positive selection regarding the adaptation of Dipterocarpoideae to the emergent canopy layer of tropical rainforests.

The comparative analysis against Malvaceae identified 171 positively selected genes, whereas that against temperate tree species identified 191 (Supplementary Fig 6). As the number of genes was limited and the corresponding KEGG and GO enrichment analysis yielded no significant results, we presented the functional annotation of GO and KEGG of these genes instead (Supplementary Tables 11 and 12). We found that both comparative analyses annotated functions relating to DNA repair and antioxidation, including Nucleotide excision repair (KEGG), Riboflavin metabolism (KEGG), DNA repair (GO), Double-strand break repair via homologous recombination (GO), Nucleotide-excision repair (GO) and Response to oxidative stress (GO) (Fig. 3a,b; Supplementary Tables 11 and 12). Such a consistent functional annotation arose from different genes identified in either analysis, such as Bloom syndrome protein (BLM) in DNA repair, UDP-sugar pyrophosphorylase (USP) in antioxidation, DNA mismatch repair protein (MSH6) in DNA repair and Pyridoxine 4-dehydrogenase (PLR1) in antioxidation (Supplementary Tables 13 and 14). More importantly, such a consistent functional annotation also arose from the same genes identified by both analyses, including DNA repair protein RAD51D (in homologous recombination), Structural maintenance of chromosomes protein 5 (SMC5), and riboflavin kinase (RFK) in antioxidation (Fig. 3c, Supplementary Fig. 7, Supplementary Tables 13 and 14).

As you suggested, adaptation to drought may also be an important mechanism for dipterocarps to thrive in tropical environments. This adaptation is not specific to the emergent canopy layer but to the overall tropical environments. In our comparison with temperate tree species, we have identified two key candidate genes associated with this adaptation. Please see below for the related revision in the results section.

In the comparative analysis against temperate tree species, we identified two positively selected genes (O-succinylbenzoate-CoA ligase (AEE14) and L-ascorbate peroxidase relevant to the plants' response to drought stress (Supplementary Table 13), which is involved in KEGG pathways of ubiquinone and other terpenoid-quinone biosynthesis (map00130) and glutathione metabolism (map00480), respectively. The finding of these positively selected genes is consistent with a previous study indicating that dipterocarps have adapted to the rare and irregular drought events in tropical environments¹⁷.

>I do not think the authors answered the point. Diverse genes were identified by the selection analysis. Although the lack of enrichment in Gene Ontology analysis or any other analysis, the authors picked up several genes relevant to UV responses. Seeing the same data, other authors can make their own stories. This is not considered objective.

>>R: We highly appreciate this thoughtful comment. We have now understood that we had arbitrarily selected genes and established their associations with UV adaptation. Therefore, we must increase the objectivity in identifying genes under selection. To achieve this, we intersected the positively selected genes from the two comparisons and found 37 genes (these are listed in Supplementary Table 13), four of which were related to the stress response according to previous studies. Moreover, we also have made efforts to improve the objectivity in interpreting our results. As Dipterocarps experience a range of stresses, including UV and drought stresses, the positively selected genes should be interpreted as adaptation to such a broad range of stressors. To achieve both ends, we made a thorough revision in the abstract, introduction, results and discussion. In the abstract, we revised the corresponding interpretation as:

Several positively selected genes were involved in antioxidation and DNA repair functions, likely facilitating adaptation to environmental stresses in Asian rainforests. (Lines 44 – 46)

In the introduction, we added:

As a consequence of above-canopy life in Asian rainforests, these species are challenged with various abiotic stresses such as irregular droughts¹⁵ and intensive ultraviolet (UV) radiation¹⁸. (Lines 83 – 84)

In the results, we revised the corresponding part as:

There were 37 positively selected genes supported by both comparisons, among which four genes were found to be associated with plants response to environmental stresses (Supplementary Table 13). These four genes included DNA repair protein RAD51D (in homologous recombination RAD51), structural maintenance of chromosomes protein 5 (SMC5), and carotenoid ϵ -hydroxylase (LUT1) involved in DNA repair, and riboflavin kinase (RFK) involved in antioxidation (Fig. 3c, Supplementary Fig. 6 and 7, Supplementary Table 13). RAD5127 and SMC528 are found to be associated with UV adaptation, and LUT129 and RFK30 have been reported to be linked with drought tolerance. Moreover, according to the functional annotation in DNA repair and antioxidation, we identified several additional genes in either analysis, such as Bloom syndrome protein (BLM) and DNA mismatch repair protein (MSH6) in DNA repair, and UDP-sugar pyrophosphorylase (USP) and pyridoxine 4-dehydrogenase (PLR1) in antioxidation (Supplementary Table 14). (Lines 217 – 228)

Given that DNA repair and antioxidation are involved in the responses of plants to most environmental stresses^{18, 31}, these results effectively revealed the molecular footprints likely contributing to the adaptation of dipterocarps to environmental stresses in tropical rainforests. (Lines 235 – 238)

In the discussion, we revised the corresponding part as:

Dipterocarpoideae species benefit greatly from the height of their adult trees, by which they preempt light resources and facilitate pollen and samara dispersal^{16, 46}, but these advantages come with the costs, such as direct exposure of adult trees to intense UV and high vulnerability to droughts. Given the positively selected genes supported by both comparisons are functionally associated with these stresses²⁷⁻³⁰, our results offer the molecular basis for further exploring the adaptation of Dipterocarpoideae to environmental stresses in Asian rainforests. (Lines 329 – 334)

>>>I do not think that the authors after all answered to the point: "Temperate tree comparison" or "Malvaceae comparison" does not seem to make sense because dipterocarp species and Malvaceae species would form a single clade, respectively, and after all, there is only a single comparison." The two groups has many differences other than canopy layers. Two comparisons the authors provided are fundamentally the same.

I found that the statement in the abstract is too strong. At most, "genes that showed a signature of selection" may be fine.

6. data availability

Data availability section should state more specifically the availability of the assemblies and annotation files.

>>R: We have revised the data availability statement as follows:

All data used in the analysis of this study have been deposited in the CNSA (<https://db.cngb.org/cnsa/>) in the BIG Data Center, Beijing Institute of Genomics (BIG), Chinese Academy of Sciences, BioProject: ID: PRJCA017262.

>This is fine, but as noted at the beginning, code availability is missing although the Editorial Check List said "We have provided a full code availability statement in the manuscript".

Reproducibility is important.

>>R: Many thanks for addressing the importance of reproducibility of our results. We have provided the codes of bioinformatic analysis, which are available online (see Methods and the table blow).

Software Code source

RAxML v8.0.19 https://github.com/Molecology/Dipterocarpoideae-genome/Supplementary_Code1.RAxML.sh

Supplementary_Code1.RAxML.sh

MCMCtree in PAML https://github.com/Molecology/Dipterocarpoideae-genome/Supplementary_Code2.mcmctree

Supplementary_Code2.mcmctree

KaKs_Calculator v2 https://github.com/Molecology/Dipterocarpoideae-genome/Supplementary_Code3.KaKs_Calculator

Supplementary_Code3.KaKs_Calculator

QUOTA-ALIGN script https://github.com/Molecology/Dipterocarpoideae-genome/Supplementary_Code4.QUOTA-ALIGN

Supplementary_Code4.QUOTA-ALIGN

RGAugury pipeline https://github.com/Molecology/Dipterocarpoideae-genome/Supplementary_Code5.RGAugury.sh

Supplementary_Code5.RGAugury.sh

Sequncing mapping and variant calling https://github.com/Molecology/Dipterocarpoideae-genome/Supplementary_Code6.mapping.varcall.sh

Supplementary_Code6.mapping.varcall.sh

SMC++ v 1.15.2 https://github.com/Molecology/Dipterocarpoideae-genome/Supplementary_Code7.SMC.sh

Supplementary_Code7.SMC.sh

PSMC v0.6.4 https://github.com/Molecology/Dipterocarpoideae-genome/Supplementary_Code8.PSMC.sh

Supplementary_Code8.PSMC.sh

PROVEAN v1.1.5 https://github.com/Molecology/Dipterocarpoideae-genome/Supplementary_Code9.proven.sh

Supplementary_Code9.proven.sh

SIFT4G v2.0.0 https://github.com/Molecology/Dipterocarpoideae-genome/Supplementary_Code10.SIFT.sh

Supplementary_Code10.SIFT.sh

Derived alleles identification https://github.com/Molecology/Dipterocarpoideae-genome/Supplementary_Code11.derived

Supplementary_Code11.derived

Runs of homozygosity in PLINK https://github.com/Molecology/Dipterocarpoideae-genome/Supplementary_Code12.ROH.sh

Supplementary_Code12.ROH.sh

>>>This is fine.

Minor issues

line 184, which is involved

>>R: Revised. Line 242.

>Fine.

>>R: Thanks.

Reviewer #2 (Remarks to the Author):

Comments for authors

NCOMMS-22-41572B: Liu et al. Life up high: the adaptations and demography of the Asian rainforest titans, Dipterocarpoideae.

Page 6&7, lines 119-124: The histograms obtained from ploidy screening using the DNA flow cytometry were not really convincing particularly showing broad DNA peak rather than single prominent DNA peak. Even the standard reference DNA peak was broad. Perhaps this may be due to the handling error during samples chopping/preparation. I strongly encourage this to be addressed.

>>R: Many thanks for this constructive suggestion, and we have addressed this in the legend of Supplementary Fig.1, as "The histograms show broad DNA peaks for both tetraploid Hopea species and the reference species, probably due to some handling errors during sample chopping/preparation."

Comment:

Unfortunately, this is not a satisfactory response I was expecting. To resolve this, additional flow cytometry experiment on the samples need to be carried out. Broad DNA peak is just not convincing (even for standards reference DNA). Please refer to Dolezel et al 2007 Nature Protocols.

Page 16: lines 348-351: This is crucial. The natural distribution of *H. hainanensis* only can be found in Hainan province and northern part of Vietnam. They suggested that previously reported genome of *H. hainanensis* by Wang et al. 2022 was 'probably' from diploid population. Where is the evidence of diploid populations?

>>R: Apologies for the confusion. The intended meaning previously was: *H. hainanensis* may have both tetraploid and diploid populations. We adopted the "may" to indicate this was highly speculative (Please see our reply below). The sequenced individual by Wang et al. (2022) was collected from Ruili Botanical Garden (Yunnan, China), and the field origin of this sample remains unknown. To avoid misunderstandings, we have removed the sentence indicating the possibility of both tetraploid and diploid populations.

Comment:

The above explanation is not convincing to me. I would suggest that the authors to include some form of empirical evidence or justification in their manuscript as to why *H. hainanensis* is a tetraploid species compared to that of Wang et al 2022 which was reported as diploid individual. Perhaps, the used of a study by Wang et al 2020 (PloS One) using 12 SSR markers on 10 populations of *H. hainanensis* (refer S1 File) as support that the allelic dosage was used to determine the amplified peaks based on ratios between peak intensities from SSR markers suggesting autotetraploidy.

If both tetraploid and diploid populations do exist, how do they know the 30 *H. hainanensis* individuals used for resequencing study were all autotetraploid individuals (minus 4 individuals tested for kmer analysis)?

>>R: It is a nice idea to validate that all individuals are tetraploid by k-mer analysis. Accordingly, we conducted k-mer analysis for the remaining 26 samples, and they are indeed all tetraploid, with one sample (JF4) difficult to judge. Expect this sample, the depth of the common peak on the right was four times that of the heterozygous peak (Please refer to the figure below), supporting that they are tetraploid. It appears that the tetraploid pattern is only apparent with extremely deep sequencing (Supplementary Fig.1a), relatively clear with deep sequencing (Supplementary Fig.10-r, showing individuals with the highest resequencing depth), and almost invisible at low depths (Supplementary Fig. 1 in Wang et al. 2022). Supplementary Fig. 1 in Wang et al. (2022) only had one peak (possibly due to low sequencing depth), providing no clear support of diploid or tetraploid.

Comment:

The kmer analysis is acceptable. It should be included as supplementary figure and revise the main text accordingly (y-axis: 'Frequency'). However, there is one individuals JF4 need to be clarified further as to the ploidy status of the individual. This could be a diploid individual. If there are the actual occurrences of diploid and tetraploid mix within population these need to be clarified.

Additional comments:

Lines 41-43: This is a rather bold and misleading claim. I would suggest it to be removed. The estimated divergence time between Dipterocarpaceae and Malvaceae was not consistent with generally accepted divergence timeline of various angiosperm/eudicots. The statement "Gondwanan vicariance hypothesis" just came out all of the sudden. Completely no discussion on Gondwanan vicariance hypothesis in the main text. No doubt, the timing and the nature of Gondwanan breakup is highly debatable but a meaningful and convincing discussion in relation to the current findings need to be presented.

Line 109: 'PHD' or 'HPD' = highest posterior density. Please check it throughout the MS.

Line 197 & Line 199: Please be specific on what you meant by "...of the three species"

Lines 198-201: Just curious, from Fig.2a, what would be the explanation for *Hopea odorata* as this species has been previously identified as 2x (2n=14) and 3x (2n=20-22) Kaur et al. (1986); Sarkar et al. (1982); Jong & Lethbridge (1967), Roy & Jha (1965); Tixier (1960). This would be interesting discussion as to why *Hopea* spp. exhibit several types of ploidy levels presently.

Fig. 2a: I do not see the purpose of putting the fruit images in the figure unless they mean something.

Reviewer #3 (Remarks to the Author):

In an effort to address concerns about running PSMC and SMC++ demographic analyses on polyploid plants, the authors used non-polyploid reference genomes to map their reads to. If the reference and mapped species are +/- evolutionarily close, there seems to be nothing "wrong" with this approach. We have found, however, that running such analyses with reads mapped to distant references can heavily bias results; as such, in our projects, we have resorted to running PSMC (for example) only in self-self mode, i.e., using the same species' assembly as reference for reads mapping, even if that assembly is nowhere near as contiguous as a chromosome-scale reference. The limitations of using shorter contigs appear to outweigh the limitations of evolutionarily distant (i.e., poor) reads mapping. The results shown in Fig 4a are coincident enough to suggest that any reference bias was not appreciable. However, the *Hopea* SMC++ curves show slight shifts in years before present between the two species, which could reflect heterozygosity differences between them rather than real demographic/geological-event differences. I suggest the authors refer to this possibility, which was recently explored in detail as part of demographic work for the lychee genome paper: <https://www.nature.com/articles/s41588-021-00971-3> - see supplementary material, Supplementary Note II.

RESPONSE TO REVIEWERS' COMMENTS

Many thanks for the constructive suggestions from all reviewers. In this revision, according to the suggestions by Reviewer #1, we have revised our phylogenomic analysis based on the generally accepted evolutionary timeline of angiosperms/eudicots and the analysis of positively selected genes. Furthermore, we have re-performed the experiments and analyses addressed by Reviewer #2. We also would like to thank Reviewer #1 and Reviewer #3, whose suggestions have significantly improved our understanding of demographic analysis. The reviewers' comments in this round of revision are bold in black and the authors' responses are in blue, with revised contents highlighted in yellow. The contents from the previous rounds of revision are in grey.

Reviewer #1 (Remarks to the Author):

New comments are shown by >>>.

Reviewer #1 (Remarks to the Author):

General comments

The comments by reviewer 2 were highly similar to mine. The authors provided additional meaningful experimental data using flow cytometry. However, in bioinformatic analysis, many additional analyses were superficial and did not address the fundamental theoretical issues in phylogenetic and population genetic analysis. The methods are so short that I cannot judge if the analyses were appropriate. Alternatively, code can be informative on the methods, but the main text does not have a statement of code availability, although the Editorial Check List said "We have provided a full code availability statement in the manuscript". The authors should consider reproducibility seriously.

>>R: Many thanks for the constructive suggestions in both rounds of review, and for the positive comments on our previous reply. In this revision, we have improved methodological details, and in particular, we have provided the analysis codes and made them publicly available. Concerning the genomic analysis of polyploid species, we

fully agree with the reviewer that this field is challenged by the lack of generalized methodologies. Our study, representing the first genome assembly of wild autotetraploid species (previous autotetraploid assemblies were sugarcane (Zhang et al. Nature Genetics, 50: 1565–1573, 2018), alfalfa (Chen et al. Nature Communications, 11: 2494, 2020), and potato (Sun et al. Nature Genetics, 54: 342–348, 2022)), has made various attempts to explore methodology following your constructive suggestions, and we hope to promote the research on polyploid genomes together with the reviewers. Please note that the comments are bold in black, and our responses are highlighted in blue, with the revised contents are in green. The contents from the previous round of revision are in grey.

My comments on each point will be shown by >.

>>R: We highlight these comments in bold to separate them from the previous round of revision, and our responses are in blue.

>>R: Many thanks for your constructive suggestions in the revision. We find that we have successfully addressed the following major concerns in the previous revision, including: 1. The integration of the previous genome assembly; 3. Ancient polyploidy was already reported; 6. Data availability; and part of 2. Novelty in autopolyploidy, as you commented on our reply as ">>>This is fine." or ">>>Fine.". Therefore, in this revision, we will focus on the concerns that are not fully addressed: 4. Time estimation; 5. Cherry-picking of genes under selection; and the population demography part of 2. Novelty in autopolyploidy. Please see the point-to-point reply below, where your comments are in bold black, previous comments are in grey, responses are in blue, and revisions in the manuscript are highlighted in yellow. We hope you will find that the solidity and logical progression of this manuscript have been significantly improved.

Response to reviewers' comments

Reviewer #1 (Remarks to the Author):

Dipterocarp trees are dominant in Southeast Asian tropical forests and important for environments, forestry, and medicine. Recently, a few genomic studies reactivated long-standing debates about its evolutionary history centered on its Gondwanan

vicariance hypothesis. The authors reported chromosome-level assemblies of seven dipterocarp tree species. Two of them are autotetraploid tree species, which can be a unique resource to study plant evolution in general. However, a major revision would be necessary to place the study in the context of previous research.

>>R: Many thanks for your positive comments and your constructive suggestions. Following your suggestions, we have revised the manuscript thoroughly; please find the point-to-point reply below. We hope you will find that novelty and the significance of this study have been substantially strengthened.

1. The genome assemblies of dipterocarps, i.e., *Shorea leprosula* (Ng et al. *Commun Biol* 7:1166, 2021) with transcriptome and resequencing of other dipterocarps, *Dipterocarpus turbinatus* and *Hopea hainanensis* (Wang et al. *Plant Biotechnol* 20:538, 2022, Epub 2021 Dec), were already reported some time ago. Many results of this manuscript, such as ancient polyploidy and time estimation, were reported. The same species, *Hopea hainanensis*, was sequenced before. The current manuscript would give a false impression of novelty. This is not scientifically proper. Rather, I would suggest that the comparison with previous studies and the introduction explaining unsolved issues would highlight the importance of this new study as explained below. Their report should be integrated fully in the introduction and discussion. Furthermore, Tian et al. (*Plant Communications* 2022, 100464, 2022) reported 13 dipterocarp genomes in October 2022. I am not sure if this is before or after the submission of this manuscript and may depend on the journal policy, but integrating it would further strengthen this manuscript.

>>R: Many thanks for your suggestions and the references. Indeed, we agree that it is important to acknowledge the previous discoveries. Following your suggestions, we have highlighted the contributions of Ng et al. 2021, Wang et al. 2022, and Tian et al. 2022 while introducing the debates surrounding the origin of the Dipterocarpaceae. We have rephrased the paragraphs in the introduction and discussion sections (The revision is highlighted in blue in the main text, please also see below).

In the introduction, we added:

Earlier studies proposed that Dipterocarpaceae originated in Western Gondwanaland in

Early Cretaceous (c.120 million years ago (MYA)), before the separation of Africa and South America¹⁴. Recent evidence from pollen fossils¹⁵ and a phylogenomic study¹³ supported such a Western Gondwanaland origin, albeit with a slightly recent dating (c.102.9 MYA¹⁵ and 105.0 MYA¹³). These studies suggested that Dipterocarpaceae dispersed to India during the mid-Cretaceous and subsequent to Southeast Asia facilitated by India-Asia collision. However, genomic studies involving different sets of dipterocarp species revealed a much more recent divergence between Dipterocarpaceae and its closest family Malvaceae (c. 84 MYA¹⁶ and 86–98 MYA¹⁷). Therefore, there is still a controversy surrounding the evolution of these keystone rainforest species; resolving it will provide valuable insights into the origins of megathermal angiosperms.

In the discussion, we added:

Here, the phylogenetic history constructed using a total of 19 Dipterocarpoideae genomes agrees with the results from the early view¹⁴ and some recent phylogenomic/phylogenetic studies^{13, 15}, thus supporting the Western Gondwanaland origin of Dipterocarpaceae. Different calibrations yielded consistent results, indicating this estimation of divergent time is robust. This finding is largely consistent with the evolutionary history of angiosperms revealed by genomic data³⁵ and new fossil evidence³⁶, showing far earlier splits between major clades of angiosperms (occurring at the Jurassic and Lower Cretaceous) than traditionally expected.

Furthermore, as you suggested, we have integrated all published Dipterocarpaceae genomes, including those in Tian et al. 2022 in the revised phylogenomic analysis, and thus form a dataset of 19 Dipterocarp genomes. These suggestions significantly improve the novelty and solidity of our study. Please refer to the response for this specific issue below.

>The new paragraphs partly solved the lack of introduction. However, other important conclusions of the manuscript include genome duplication. A more thorough description of previous publications is necessary.

>>R: Thanks. According to your suggestions, we have now included a more thorough description of previous findings, including the genome duplication event and others.

In the introduction, we added:

Recent genomic studies^{13, 14, 15} found a whole genome duplication (WGD) event preceding the divergence of Dipterocarpoideae species. This WGD event increased the chromosome number of Dipterocarpoideae ancestors to 12, and the following two steps of karyotype evolution gave rise to 11-chromosome species (Dipterocarpeae tribe) and 9-chromosome species (Shoreeae tribe)¹³. (Lines 64 – 68)

>>>**This is fine.**

>>**R:** Thank you very much.

2. Novelty in autopolyploidy

Wang et al. reported the assembly of the same species *Hopea hainanensis* but they did not mention autotetraploidy at all and treated it as a normal diploid species. This manuscript provided chromosome count (Supplementary Figure 1) and careful k-mer analysis to provide strong evidence of polyploidy. Furthermore, they seem to have successfully provided phased four assemblies, although further quality checks in this aspect should be reported. In general, the genome assembly of autopolyploid species remains a major challenge in genomics. For example, the assembly of autotetraploid *Arabidopsis arenosa* is difficult, and instead, self-fertilizing allotetraploid *A. suecica* derived from *A. arenosa* was sequenced instead (Burns et al. *Nature Ecology and Evolution*, 10:1367, 2021; Jiang et al. *Nature Ecology and Evolution*, 10:1382, 2021). Supposing the quality of the phasing is quantified, the new assembly of *H. hainanensis* would provide interesting insights into the autopolyploid genome.

To support the tetraploidy, please add the empirical data of genome size measured by flow cytometry. Ng et al. (*Plant Ecology and Diversity* 9:437, 2016) explained a standard method of dipterocarps.

>>R: Many thanks for acknowledging the significance of our tetraploid assembly and for the references. The autotetraploid assembly presented a major challenge until recently. Zhang et al. 2018 *Nat Genet* provided the first allele-aware assembly of sugarcane and corresponding methods were published by Zhang et al. 2019 *Nat Plants*.

We followed their approach and maintained close communication with the authors since we started the project in 2019, and these efforts resulted in the assembly of comparable quality to Zhang et al. 2018 Nat Genet. We also provided evidence showing the quality of the two autotetraploid genomes (Lines 133-140, 144-147, and 150-154; Supplementary Tables 3-5). So far, the allele-aware genome assembly of autotetraploid was still limited, in particular for wild species, and we hope that our work will inspire further involvement in studies of Dipterocarpaceae and wild tetraploid species.

Furthermore, we highly appreciate the methods you provided. We conducted flow cytometry analysis and estimated the C-values accordingly. The corresponding results have been added to Figure S1 and Table S1 (please also see below), providing robust evidence in support of tetraploidization. We have also revised the results section accordingly.

In the results, we added:

Moreover, by using *Solanum lycopersicum* ($2C = 2.12$) as a reference, the flow cytometry results revealed the C-values of *H. chinensis*, *H. reticulata* and *H. hainanensis* are $2C$ equal to 0.63, 1.36 and 1.66 (Supplementary Fig. 1j-l and Supplementary Table 1). Compared to the diploid *H. chinensis*, the cytometry analysis identified a clear peak confirming the tetraploidy of these two species (Supplementary Fig. 1m-n).

In the methods, we added:

To further test whether *H. hainanensis* and *H. reticulata* are tetraploid species, we estimated their genome sizes and ploidy using flow cytometry. For these two species and *H. chinensis* (a diploid congener), we collected the fresh young leaves from the sampled trees used for the de novo genome assembly. We also sampled young leaves of *Solanum lycopersicum* (genome size: 2.07 Gb; $2C = 2.12$ pg⁵³) as the reference for genome size estimation. We first estimated the genome sizes and $2C$ values of the three *Hopea* species by measuring the fluorescence of nuclei from the mixture of each *Hopea* species and *S. lycopersicum* and those from each single species. Then, we assessed the ploidy for *H. hainanensis* and *H. reticulata* based on their fluorescence of nuclei setting *H. chinensis* as the diploid reference. The nuclei suspension was prepared for each

species following the methods described in Ng et al.⁵⁴. We measured the fluorescence from the nuclei suspension using a Sysmex CyFlow Cube6 flow cytometer, and data analysis was performed using FCSExpress 7plus.

>Flow cytometry data are satisfactory. This also solved a similar issue raised by Reviewer 2.

>>R: Thanks.

The methods to analyze the demographic analysis of the autotetraploid need to be thoroughly revised. As far as reading the Methods, software developed for diploid species is applied to the data. However, many assumptions are not filled. Please refer to the population genetic analysis of autotetraploid *A. arenosa* (Monnahan et al. *Nature Ecol Evol* 3: 457, 2019) and theoretical studies cited there (particularly Otto and Whitton, *Ann Rev Genet* 34:401, 2000). Here I list a few examples below but the authors should investigate the effect of autopolyploidy thoroughly. As far as I understand, SMC++ and PSMC assume diploid inheritance like a human being. Intuitively, it may be usable if you randomly choose two haplotypes, but a thorough theoretical basis should be explained if it would be included in the revised manuscript. Furthermore, the generation time of 10 and 15 years (line 741) are very likely minimum estimates, because the trees will produce seeds for a long until a gap may be formed in the forest or other reasons. Relative time estimation of N_e would be interesting but the usage of absolute time to compare the demography with glacial cycles should be omitted.

>>R: Thank you for your suggestions, especially those regarding tetraploid analysis. We may not have fully understood the methods presented in Monnahan et al. (2019) *Nature Ecol Evol* 3: 457, as they did not seem to conduct a demographic analysis using those tools. Instead, we appreciate your suggestion of extracting and analyzing two haplotypes. This approach treats the tetraploid genome as a diploid genome and aligns with the model assumptions. Accordingly, we extracted the two longest monoplloid genomes for analysis (Reviewer 2 recommended this approach as it would include the most genomic information). Furthermore, we also adjusted the generation

times to 15, 20, and 30 according to your suggestions.

>The newly added method is just a single sentence without a reference.

Line 779 "To meet the standards of population demography analyses (using SMC++ and PSMC), we mapped the filtered reads to the longest two monoploid genomes of each species to form an approximation of diploidy."

This is far too obscure. Furthermore, no codes were provided. What is the motivation to map reads to two monoploid genomes? Then, most genes have two duplicated copies. Mapping to duplicated genes is a complex issue. As a reviewer, I cannot be sure if the analysis was appropriate.

>>R: This suggestion is gratefully accepted. The codes of PSMC (https://github.com/Molecology/Dipterocarpoideae-genome/Supplementary_Code8.PSMC.sh) and SMC++ (https://github.com/Molecology/Dipterocarpoideae-genome/Supplementary_Code7.SMC.sh) have been provided and the reference was added. Indeed, mapping to duplicated genomes is complex, and to ensure reliable demography analysis, we only included the uniquely mapped reads (BWA mapping quality > 0) in the analysis. In the PSMC analysis, we set -Q 20 in vcf2fq to indicate the minimum mapping quality is 20, and used the default --minimum-mapping-quality10 in the GATK variant calling (https://github.com/Molecology/Dipterocarpoideae-genome/Supplementary_Code6.mapping.varcall.sh) whose output .vcf was input for SMC++ analysis. We also added details of these parameters in the methods (Please see below).

Concerning the motivation to map reads to two monoploid genomes, we are inspired by your recommended references, i.e. the empirical study (Monnahan et al. Nature Ecol Evol 3: 457, 2019) and the theoretical study (Otto and Whitton, Ann Rev Genet 34:401, 2000). These studies indicate the general methodology for polyploid genomics is currently lacking, and it is necessary to adopt novel approaches. Therefore, we decided to carry out the attempt using two haplotype genomes other than the traditional approach using only one haplotype genome, despite that the reliability of this method requires further investigation due to the lack of precedents. In the methods, we added a paragraph to interpret this motivation and cite the references.

The challenge of demographic analysis for autopolyploid species is mainly caused by the lack of generalized methodologies^{48, 108}. To date, the algorithms and software (e.g. SMC++³⁶ and PSMC³⁵) developed for effective population size analysis are restricted to diploid analysis. To meet the requirements of these analyses, we made an attempt to treat the autotetraploid species as diploids by using the longest two monoploid genomes as the reference genome. (Lines 776 – 780)

We estimated the recent-past effective population sizes (N_e) of *H. hainanensis* and *H. reticulata* separately using the coalescent-based inferences in SMC++ v 1.15.236. The VCF files, including 30 *H. hainanensis* samples and 32 *H. reticulata* samples, were converted to the SMC++ input files using the ‘vcf2smc’ command. The estimate was run with generation times of 15, 20 and 30 separately and the corresponding estimated mutation rates (Supplementary Table 20). Codes are available at https://github.com/Molecology/Dipterocarpoideae-genome/Supplementary_Code7. SMC.sh. (Lines 781 – 787)

In addition, we inferred historical changes in N_e of *H. hainanensis* and *H. reticulata* using PSMC v0.6.435. For each of 30 *H. hainanensis* samples and 32 *H. reticulata* samples, we performed the analysis with generation times of 15, 20 and 30 and the corresponding estimated mutation rates separately (Supplementary Table 20). The whole-genome consensus sequence was generated by SAMTOOLS v1.3, bcftools, and vcfutils.pl, setting the minimum read depth to 10, maximum read depth to 100, and minimum mapping quality 20. We ran PSMC with parameters (psmc -N30 -t15 -r5 -p 4+25*2+4+6) (https://github.com/Molecology/Dipterocarpoideae-genome/Supplementary_Code8. PSMC.sh). (Lines 788 – 795)

>>>I cannot be sure that the results of these analyses are interpretable. PSMC or SMC++ were developed for diploid species. When a particular argorism is applied to unintended type of datasets, simulation or theoretical validation (or citation to previous validation) is necessary. The authors said: "despite that the reliability of this method requires further investigation due to the lack of precedents". However, it is not a future issue, but it must be provided here.

I can imagine many pitfalls. As far as I understand, both in PSMC and SMC++,

alleles should be randomly chosen. It is unclear the usage of two haplotype genomes can accurately estimate four alleles. Suppose three alleles are similar and one allele is divergent in the individual of reference genome. This would lead to two assembled haplotypes. If the sequence reads of the same species is mapped, one allele would show very low polymorphism, and another with erroneously high level of polymorphisms due to the mixture of three alleles. In other situations, four alleles can be assembled into two and two. Then, either of the two are biased to be similar. This will affect the N_e estimates. These are only a few scenarios. I cannot be sure about the analysis. Potentially theoreticians may be able to evaluate or support that the analyses.

Due to this reason, I cannot agree with a major sentence in the abstract lines 46-48 "Resequencing of two endangered species showed an expansion of effective population size after the last glacial period and a recent sharp decline coincidental with the history of local human activities."

>>R: Thank you for your patience and your rigorous approach to scientific inquiries. We fully agree with you that the legitimation of the demographic methodology and the robustness of corresponding results are critical issues. Moreover, the suggestions raised by the editor and Reviewer 3 also provide further insights into this analysis. The possible explanation concerning the demographic analysis is illustrated as below.

The reviews by Parisod et al. (2010, *New Phytologist*, 186: 5–17) and Li et al. (2021, *Annu. Rev. Plant Biol.* 72:387–410) have shown that autopolyploid genomes are expected to experience rapid cytological diploidization, i.e., the establishment of bivalent pairing (e.g., evolving from 4 homologous chromosomes in an autotetraploid genome to two pairs of chromosomes). This is because natural selection tends to eliminate multivalent pairings, which often leads to chromosomal abnormalities and infertility. While bivalent pairing keeps the chromosomes in the same pair genetically similar, chromosomes from different bivalent pairs diverge with time. Such divergence is likely to cause divergence among alleles in the scenarios mentioned in your comment. Signals of ongoing divergence were also indicated by the K_s analysis in the two autopolyploid species, as the results exhibit a clear peak of pairs of paralogs within each

genome though at a very small genetic distance (Fig. 2b).

We also agree that the results of PSMC or SMC++ should be interpreted with caution. Ideally, data mapping should use divergent chromosomes from two bivalent pairs as the reference (If AAAA was divergent into A_1A_1 and A_2A_2 , where A_1A_1 and A_2A_2 were two bivalent pairs, the ideal reference is A_1A_2). However, due to the challenge of distinguishing these divergent chromosomes, two chromosomes from the same bivalent pair (A_1A_1 or A_2A_2) may be selected for population genomics. This will result in the SNPs including both polymorphism and divergence, i.e., overestimating the polymorphism.

Regarding the caveat in interpreting the results, Reviewer #3 gave us a valuable simulation (Nature Genetics 2022, 54:73-83, Supplementary Note II. Please also see the figure below). This simulation suggests that changes in polymorphism do not affect the shape of the curve but cause a shift along the timeline. Specifically, a decrease in polymorphism shifts the curve to the left. Consistent with this expectation, the demography of *H. hainanensis* and *H. reticulata* in the current analysis (using two monoploid genomes as reference) has shifted the curve to the left compared to the previous analysis (using the longest monoploid genome as reference), because the current analysis captures a part of divergence whereas previous analysis captures almost all divergence. We anticipate that if the ideal references were adopted (polymorphism may decrease), the curve would shift further to the left compared to the current analysis. Therefore, the conclusion that the effective population size rose after the last glacial period and experienced a recent sharp decline remains robust.

Nature Genetics 2022, 54:73-83, Supplementary Note II. The model simulated 50% or 75% loss of heterozygosity loci due to selfing.

The SMC++ results of the previous analysis.

The SMC++ results of the current analysis.

Surprisingly, different generation times and methods (SMC++ and PSMC) yielded highly consistent results. Please see below for the revised Figure 4a and Supplementary Figure 11. The results section was also revised thoroughly (please also see below). It is indeed a challenge to provide a clear theoretical explanation for the mechanism behind these robust results. Intuitively, your suggestion that treating the tetraploid genome as diploid to fit the model assumption likely contributes significantly to the robustness of the results. Many thanks again for the suggestion. (Discussion about Otto and Whitton, 2000; please see the reply below)

We revised the results as:

The demographic history for both species was performed using SMC++30 (see Methods), based on generation times of 15, 20, and 30 years (Supplementary Table 21). We found that the effective population sizes of both species increased after the Last Glacial Maximum (LGM, c. 19,000 years ago), and reached a maximum around 6,000 – 10,000 years ago for *H. hainanensis* and 3,000 – 9,000 years ago for *H. reticulata* (Fig. 4a). However, both species experienced a sharp decline recently around 1,500 – 2,400 years ago for *H. hainanensis* and 1,200 – 2,400 years ago for *H. reticulata*. During this

period, effective population sizes decreased from the maximum (c. 5.0×10^6 for *H. hainanensis* and c. $4.0 \times 10^6 - 6.0 \times 10^6$ for *H. reticulata*) to less than 1,000 (Fig. 4a). The demographic analysis with PSMC31 showed consistent results (Supplementary Fig. 11). Since human activities in Hainan Island can be traced back to 3000 years ago³², the above evidence suggests that human intervention may have disrupted the maintenance of their populations after the glacial period, thus serving as a primary major factor contributing to species endangerment.

>Seeing the Figures, I do not understand why the authors stated "The demographic analysis with PSMC31 showed consistent results". As far as I saw, they were different in the important aspects. In the SMC analysis of *H. hainanensis* (left, top), the population size increased after the band of LGM, then drop. However, in the PSMC analysis, the vast majority of the lines did not show an increase. A subsequent drop was not observed, either. I wonder if the authors carefully examined the results. By the way, it is well known that the estimation in very recent times using SMC and PSMC may not be reliable, so should not be interpreted much.

>>R: Many thanks for the suggestions. Indeed, PSMC analysis is not reliable within the last ten thousand years (Li H, and Durbin R, *Nature*, 475: 493–496 2011). Although SMC appears to provide accurate estimates across a time span of three orders of magnitude (10³–10⁶ years ago), it is not reliable within the last thousand years (Terhorst et al. *Nature Genetics*, 49: 303–312, 2017). Thus, for the period between the last ten thousand and one thousand years, corresponding to after the Last Glacial Maximum (LGM), it seems only SMC provided relatively reliable results. In the revision, we removed the comparison between the two approaches, and interpreted the results only on their reliable ranges.

In the results, we added:

The demographic analysis with PSMC35 showed a minor declining trend in effective population sizes (N_e) for both *H. hainanensis* and *H. reticulata* from one hundred thousand years ago and continuing throughout Last Glacial Maximum (LGM, c. 19,000 years ago) (Supplementary Fig. 10). For more recent demography, the SMC++³⁶ analysis based on three different generation times (15, 20, and 30 years; Supplementary

Table 20) revealed that the N_e of both species increased after the LGM, reaching a maximum six to ten thousand years ago for *H. hainanensis* and three to nine thousand years ago for *H. reticulata* (Fig. 4a). (Lines 263 – 269)

>>>The point itself is fine, but the mapping problem in the last item must be solved before this analysis.

>>R: Thank you very much!

In line 239, a comparison with cherry-picking several tree species does not make sense and so should be omitted. In line 221, Tajima's D, CLR, and iHS are not designed for autopolyploid species, although they can work for allopolyploid species.

>>R: Thanks. Accordingly, we have removed the relevant descriptions of genetic diversity of several tree species from the main text. Indeed, Tajima's D, CLR, and iHS are not designed for autopolyploid species. Monnahan et al. (2019) and Zhang et al. (2018) converted the tetraploid data into diploid data by mapping against one monoploid genome and considering two-allelic sites only. Our previous approach followed their methods. However, their analyses mainly focused on neutral processes and did not include selection analysis. Therefore, there is currently no reference for population genomic analysis regarding the signature of selection in autopolyploids.

To explore the effect of autopolyploidy and determine appropriate methods, we followed your previous advice and re-ran the entire population genomics analysis, including structure, selection analysis, and ROH, using the two longest monoploids as templates for mapping. This approach treats the tetraploid genome as a diploid genome (referred to as the diploid model analysis). To facilitate comparison, we have compiled all the results together (please see the figure below).

In the structure analysis of the neutral process, we found consistent results between the two methods. However, in the selection analysis, the diploid model analysis identified a greater number of positively selected genes than the tetraploid model. Otto and Whitton, 2000 predicted that the efficiency of positive selection is reduced in polyploids, particularly when the mutation is not dominant. Our observation is consistent with their prediction that beneficial mutations are less efficient in reaching high frequencies in

polyploid, leading to less prominent positive selection signatures in the tetraploid model analysis. On the other hand, having four alleles per locus (a characteristic of autotetraploidy) increased the genomic heterozygosity. Therefore, although the previous method of mapping against the longest monoploid genome was not perfect, it remains the optimal solution. We retained our previous methods in this manuscript. Thank you for the fruitful discussion, which has greatly enhanced our understanding of the theory and analysis on tetraploid species.

>I do not think the authors addressed the point. A deep understanding of population genetics is lacking.

>>R: We misunderstood your suggestion previously. We have now realized that your suggestion was to omit analysis of Tajima's D, CLR, and iHS, and delete the results. In this revision, we revised it accordingly.

>>>**Fine.**

>>R: Thank you very much!

3. Ancient polyploidy was already reported

Ng et al. (2021) and Wang et al. (2022) reported a genome duplication event preceding the divergence of Dipterocarpoideae species. The authors did find it out, but the original paper should be cited. Otherwise, it will give a false impression of novelty.

>>R: Many thanks. We agree it is critically important to acknowledge previous findings. We therefore referred to these excellent studies in the results and discussion, indicating our findings were consistent with them.

In the results, we revised and added:

The genetic distance of paralogs within these species was found to be greater than the distance of orthologs between pairs of species, indicating that WGD-1 occurred before the diversification of Dipterocarpoideae species (Fig. 2b and Supplementary Fig. 4). This finding is consistent with previous genomic studies^{13, 16, 17}, suggesting that WGD facilitated the diversification of Dipterocarpoideae species.

In the discussion, we revised and added:

Detection of WGD-1 and WGD-2 probably reflect a genome-scale adaptation to ancient

climate/environmental changes. We found that the WGD-1 preceded the diversification of Dipterocarpoideae species, which is consistent with several previously published studies on genomes in this clade^{13, 16, 17}.

>The content is fine but please add this to the introduction as mentioned above.

>>R: Thanks. This is added in the introduction as:

Recent genomic studies^{13, 14, 15} found a whole genome duplication (WGD) event preceding the divergence of Dipterocarpoideae species. This WGD event increased the chromosome number of Dipterocarpoideae ancestors to 12, and the following two steps of karyotype evolution gave rise to 11-chromosome species (Dipterocarpeae tribe) and 9-chromosome species (Shoreeae tribe)¹³. (Lines 64 – 68)

>>>Fine.

>>R: Many thanks.

4. Time estimation

There has been a long dispute on when Dipterocarpaceae/Dipterocarpoideae originated (see references cited in the literature below). Traditionally, the Gondwanan vicariance hypothesis based on broad geographic distribution proposed an old origin of >120 million years ago. However, the estimates are variable among phylogenetic studies, and most importantly, the Gondwanan hypothesis contradicts the general time scale of angiosperm evolution. Ng et al. conducted a detailed analysis using the genome data and fossil calibration and reported much younger divergence. In 2022, new microfossils of dipterocarps were reported, which can push the divergence time even younger (Bansal et al. *Science* 375:455, 2022). The issue is currently unsolved and can be potentially a major contribution to this manuscript. The authors cited Bansal et al. 2022 but did not discuss the dispute at all.

I should note that the methodologies the author used are primitive. The authors should examine the complexity and many assumptions in the time estimation. For calibration, authors took values from the TimeTree webpage, but this does not seem sophisticated. Particularly, little data on dipterocarp and its relatives were available. For

example, when "Hopea" and *Theobroma cacao* was filled in on its webpage, the outcome was 80 MYA (CI 38.7-85.0 MYA). In line 139, the authors reported c. 68.2 MYA without providing a confidence interval. This illustrates the insufficient quality of the current analysis. At the moment, it would be wise to use two different ways of fossil calibration, the new ones by Bansal et al., and traditional ones used by Ng et al. Otherwise, it may be better to remove time estimation from this manuscript.

>>R: Many thanks for introducing the Gondwana hypothesis into this study, which has substantially enhanced the significance of this study. Regarding the controversies surrounding this topic, we have referred to these previous studies and made a throughout revision in the introduction and discussion sections. Please refer to our responses to the first major comment.

Following your suggestions, we identified 12 dipterocarp genomes from Ng et al. 2021, Wang et al. 2022 and Tian et al. 2022, and included them all in our phylogenomic analysis. Furthermore, we calibrated the phylogenetic tree using the fossil points from Bansal et al. and the calibration provided by Ng et al. Both calibrations yielded consistent estimates of divergent times, supporting the Gondwana origin hypothesis. Please refer to the revised Fig. 2a and Supplementary Fig. 4a attached below. We have also updated the methods and results selections (Please also see below).

We would like to note that the gene family expansion/contraction analysis re-run using the new dating, and the results were largely consistent with the previous version (Fig. 4e; Supplementary Fig. 6), which revealed one expanded and 36 contracted gene families in the comparison with Malvaceae species, and the comparison with temperate trees identified three expanded and 53 contracted gene families.

We revised the maintext as: To investigate the phylogenetic relationships among Dipterocarpoideae species and their divergence with the closest family Malvaceae, we performed phylogenomic analysis including 19 Dipterocarpoideae species and two Malvaceae species. These are seven Dipterocarpoideae species assembled in this study, 12 Dipterocarpoideae species (*Dipterocarpus alatus*, *Dipterocarpus gracilis*, *Dipterocarpus intricatus*, *Dipterocarpus zeylanicus*, *Hopea mollissima*, *Hopea odorata*, *Shorea leprosula*, *Shorea roxburghii*, *Shorea henryana*, *Vatica xishuangbannaensis*,

Vatica odorata and *Vatica rassak*) from Tian et al.¹³, *Theobroma cacao* and *Gossypium raimondii*^{25, 26}. The maximum likelihood (ML) tree constructed with 918 single-copy orthologs (from 21,109 – 48,040 clustered gene families (Supplementary Fig. 3 and Supplementary Table 9)). We adopted the set of calibrations in Bansal et al.¹⁵, including the divergence time of *Vatica* species (54 MYA), *Shorea* species (54 MYA), and *Dipterocarpus* species (68.5 MYA)) (Fig. 2a). The calibrated tree dated the divergence between Dipterocarpoideae and Malvaceae to 108.0 (95% PHD: 97.4-117.3) MYA, and the earliest divergence within Dipterocarpoideae was estimated to occur at 101.7 (95% PHD: 91.5-115.1) MYA (Fig. 2a). To test the robustness of this estimation, we adopted a different set of calibration in Ng et al.¹⁷ including the divergence time between *T. cacao* and *G. raimondii* (55.8 MYA), and that between *Shorea* and *Dryobalanops* (34 MYA). The calibrated tree revealed the divergent time between Dipterocarpoideae and Malvaceae as 125.7 (95% PHD: 102.1-173.3) MYA) and that among Dipterocarpoideae species being 94.6 (95% PHD: 73.3-128.1) MYA) (Supplementary Fig. 4b). As these two calibration sets reveal generally consistent results, our study represents a most comprehensive phylogenomic analysis of Dipterocarpoideae, supporting the origin of Dipterocarpaceae in Western Gondwanaland.

>The details of the methods of the new analyses were not described. No codes were provided, either. Seeing the results, it is strange that similar times were estimated by two analyses using very distinct fossil calibrations. It cannot be excluded that the analysis may be fine, but for example, it is possible that priors specifying the lower and upper boundaries in the time in the Bayesian phylogenetic analysis may have biased the results. Without the details of the analysis, I cannot be sure if the analysis was appropriate.

>>R: Thank you very much for this instructive comment. We have now provided the codes for phylogenomic analyses (https://github.com/Molecology/Dipterocarpoideae-genome/Supplementary_Code2.mcmcree), and revised the methods to add the details (please see below). The previous setting of the upper boundary of the root was 113 MYA, but this may not be reasonable as it is smaller than the estimation (147.3 (95%

PHD: 125.8–159.4) MYA, Tian et al. 2022). We therefore revised to use a more feasible setting of upper boundary as 160 MYA. Under this setting, both calibrations and their combinations yielded consistent results (Revised Supplementary Fig. 4a and Fig. 2a, please also see below).

Revisions in the results:

The time-calibrated phylogeny dated the divergence of Dipterocarpoideae and Malvaceae at 148.7 (95% PHD: 121.9–168.1) MYA, and the split within Dipterocarpoideae was estimated to have occurred at 122.7 (95% PHD: 98.5–151.6) MYA (Fig. 2a). To test the reliability of this estimation, we used different sets of calibrations in independent analyses. The calibrated trees based on the calibrations from either Bansal et al.17 or Ng et al.15 revealed very similar divergence estimates between Dipterocarpoideae and Malvaceae (150.7 (95% PHD: 127.8–170.8) MYA and 144.8 (95% PHD: 107.6–169.3) MYA) (Supplementary Fig. 4a). (Lines 173 – 180)

Revisions in the methods:

We used MCMC tree program of PAML v4.792 with correlated molecular clock and JC69 model to estimate divergence time, setting 1,000,000 Markov chain Monte Carlo iterations and a burn-in of 100,000 iterations. The MCMC trees were constructed using different sets of calibrations and their combination. The first set included three calibrations listed in Bansal et al.17 (the divergence time among *Vatica* species (54 MYA), that among *Shorea* species (54 MYA), and that among *Dipterocarpus* species (68.5 MYA)), and the second set comprised two calibrations in Ng et al.15 (the divergence time between *Theobroma cacao* and *Gossypium raimondii* (55.8 MYA), and that between *Shorea* and *Dryobalanops* (34 MYA)). Based on the earliest divergence time between Dipterocarpaceae and Malvaceae (147.3 (95% PHD: 125.8–159.4) MYA) reported previously¹³, we set the upper boundary of their divergent time as 160 MYA (RootAge = <1.6 in the mcmctree.ctl file). For the full list of parameter settings, please refer to the code (https://github.com/Molecology/Dipterocarpoideae-genome/Supplementary_Code2.mcmctree). (Lines 659 – 671)

Revised Supplementary Fig. 4a

Revised Fig. 2a

>>>Despite additional sentences, my original question "Seeing the results, it is strange that similar times were estimated by two analyses using very distinct fossil calibrations." was not solved. Remembering the first submission, the authors stated oppositely recent estimates of 68.2 MYA of the divergence between Dipterocarpoidea and Malvaceae (line 139).

Seeing the Github files (tmp_2.tree and tmp_3.tree files), it is unclear how the authors included the calibration points. For example, Ng et al. said "A minimum age of 34 Ma was assigned to the Shorea–Dryobalanops node based on fossils from the late Eocene attributed to Shorea", but the number of 34 was not explicitly found. I myself am not an expert in phylogenetic analysis using MCMC in PAML. Experts in the field may be able to provide a better evaluation.

The authors did not provide enough background on the discussion on the Gondwanan vicariance hypothesis. Bansal et al. 2022 provided older fossils but did not mention the "Gondwanan vicariance hypothesis", and rather suggested Africa-India floristic interchange. The hypothesis contradicts the timescale of angiosperm evolution as pointed out by Ng et al. 2021.

I cannot give support to a major statement in the abstract (lines 41-43) "The divergence time between Dipterocarpoidea and Malvaceae was estimated to be 148.7 (121.9–168.1) MYA, supporting the Gondwanan vicariance hypothesis."

>>R: Thank you for your patience and constructive suggestions. In the previous version, the similar divergence time between Dipterocarpaceae and Malvaceae with two sets of fossil calibrations is possibly because this time estimate is at the root of the tree, which is mainly constrained by the prior we set for the root (<160 MYA). Therefore, regardless of the fossil calibration used, we obtained relatively consistent estimates (please note that the divergence time estimate for the internal clade did vary between different calibrations). As the estimation of root time highly depends on the priors, adding outgroups to place the node under consideration as an internal node is necessary to obtain reliable estimates.

Regarding the calibration points, these are conventionally given as a range or with upper and lower limits. For the calibration of 34 MYA, this is defined as ">0.33<0.35"

in the code.

To obtain reliable estimates based on the generally accepted timescale of angiosperm evolution, we have re-performed the phylogenomic analysis with additional four outgroup species (*Aquilaria sinensis*, *Arabidopsis thaliana*, *Oryza sativa* and *Amborella trichopoda*, see revision in Supplementary Table 9), added additional calibration point of the crown age of Malvales as 113 MYA (Vega et al., 2006) and constrained the root age using the age of Angiospermae crown-group (167–199 MYA) (Bell et al. 2010) (The method is also revised accordingly Lines 693-700). Please refer to our revision below, including the revised results, Fig. 2a, and Fig. S4. Note that though the additional two calibrations in this revision were referred to Tian et al. (2022), an obvious difference is that in Tian et al. (2022) fossil evidence was only used for limiting the minimum age of the crown of Malvales, while our study constrained the time range (both the maximum and the minimum ages) using this fossil calibration. By doing this, we think that the generally accepted timescale of angiosperm evolution can be fully considered in our phylogenomic analysis.

In the results, we revised:

To investigate the phylogenetic relationships among Dipterocarpoideae species and their divergence with the closest family Malvaceae, we performed phylogenomic analysis including 19 Dipterocarpoideae species (seven Dipterocarpoideae species assembled in this study plus 12 Dipterocarpoideae species from Tian et al.¹³) and other 6 plant species (see Supplementary Table 9). The maximum likelihood (ML) tree was constructed with 304 single-copy orthologs (from 21,109 – 48,040 clustered gene families (Supplementary Fig. 3 and Supplementary Table 9)). Using MCMCTree based on fossils and secondary calibrations^{15, 17, 25, 26}, the time-calibrated phylogeny dated the divergence of Dipterocarpoideae and Malvaceae at 108.2 (95% HPD: 97.8–118.2) MYA, and the split within Dipterocarpoideae was estimated to have occurred at 88.4 (95% HPD: 77.7–102.9) MYA (Fig. 2a).

To test the reliability of this estimation, we used different subsets of calibrations in independent analyses. The calibrated trees based on the calibrations from either Bansal et al.¹⁷ or Ng et al.¹⁵ revealed similar divergence estimates between

Dipterocarpoideae and Malvaceae (106.8 (95% HPD: 93.8–116.8) MYA and 102.5 (95% HPD: 83.6–122.2) MYA). Although the median crown age of Dipterocarpoideae from the calibrations of Bansal et al.¹⁷ (89.3 MYA) was earlier than that from Ng et al.¹⁵ (65.8 MYA), the 95% HPD estimated from the two analyses overlapped (76.4–99.7 and 48.1–109.8 MYA) (Supplementary Fig. 4a).

a

Figure 2a (revised)

a

Supplementary Figure 4a (revised)

To provide enough background on the Gondwanan origin of Dipteroocarpoideae and Africa-India floristic interchange, we have revised the introduction, and revised the discussion to put our results into such contexts.

In the introduction, we revised:

Earlier studies proposed that Dipteroocarpaceae originated in Western Gondwanaland in

Early Cretaceous (*c.*120 million years ago (MYA))¹⁶. A recent phylogenomic study¹³ supports this Western Gondwanaland origin, showing a very ancient divergence between Dipterocarpaceae and its closest family Malvaceae (147.3 MYA¹³). New evidence from pollen fossils suggested the diversification of Dipterocarpaceae since *c.*102.9 MYA¹⁷, and indicated that the dispersal of dipterocarps from Africa to India occurred during the late Cretaceous through Kohistan-Ladakh Island Arc and subsequently to Southeast Asia during the middle to late Eocene after India-Asia collision. In contrast to these findings, genomic studies involving different sets of dipterocarp species revealed a more recent divergence between Dipterocarpaceae and Malvaceae (*c.* 84 MYA¹⁴ and 86–98 MYA¹⁵).

In the discussion, we revised:

The estimated divergent time between Dipterocarpoideae and Malvaceae in this study appears to be earlier than the results from Ng et al.¹⁵ and Wang et al.¹⁴, and later than Tian et al.¹³. The estimated crown age of Dipterocarpoideae overlaps with the previous estimation by Bansal et al.¹⁷ (~94.6 MYA). Although the estimations in this study did not contradict the crown age of eudicots (*c.* 132 MYA)⁴⁰, it indicates an earlier origin of Malvales than previously⁴⁰, and appears to support the ancient origin of Dipterocarpaceae (*c.*120 MYA) in Western Gondwana¹⁶. This origin is more recent than the separation of Eastern and Western Gondwana, and the dispersal of dipterocarps to India likely occurred through the Kohistan-Ladakh Island Arc, which collided with the Indian Plate in the late Jurassic¹⁷. The dispersal to Southeast Asia was likely associated with the collision between the Indian and the Asian Plates in the late Eocene¹⁷. In particular, the genus *Hopea* was primarily distributed across Southeast Asia, and the estimated crown age of this genus (35.7 MYA) in our study suggests its diversification after the India-Asia collision. Thus, by integrating more genomes, our results support the tropical-African origin of Dipterocarpoideae and the role of Africa-India floristic interchange in forming the current distribution of Dipterocarpoideae.

The divergence time between Dipterocarpaceae and Malvaceae seems to be not strongly impacted by the calibrations within Dipterocarpoideae, while the median

crown age of Dipterocarpoideae appears to be sensitive to the within-subfamily calibration(s). The broader range of 95% HPD using the calibrations of Ng et al.¹⁵ is likely due to the inclusion of only one calibration within this subfamily. Intriguingly, the emergence of new fossil records of Dipterocarpaceae from the late Cretaceous, Paleocene, and Miocene¹⁷ supports an earlier timeframe for phylogenomic dating, resulting in an earlier crown age estimation. This phenomenon mirrors the extension of the history of angiosperms⁴¹, where the discovery of more ancient fossils provides novel opportunities to test the biogeographic hypotheses. In addition, it is worth noting that our results are only based on Dipterocarpoideae genomes, and novel insights associated with the origin of Dipterocarpaceae are likely to be provided when high-quality genomes from the other two subfamilies (Monotoideae and Pakaraimaeoideae) are available.

Finally, for the suggestion concerning the abstract, this is now revised according to the new results as:

The divergence time between Dipterocarpoideae and Malvaceae and within Dipterocarpoideae was estimated to be 108.2 (97.8–118.2) and 88.4 (77.7–102.9) million years ago

5. Cherry-picking of genes under selection

Although the authors said "GO enrichment analysis" (line 174), I could not find any statistics. Rather, it seems that the authors arbitrarily picked up genes relevant to UV, oxidative stress, and whatever they want to discuss. The authors can find standard GO enrichment analysis in many genome papers. "Temperate tree comparison" or "Malvaceae comparison" does not seem to make sense because dipterocarp species and Malvaceae species would form a single clade, respectively, and after all, there is only a single comparison. If this should be retained, the validity of the methods should be explained in detail. Moreover, their hypothesis in line 62 "At the emergent layer, these species are challenged with intensive ultraviolet (UV) radiation and associated

oxidative stress. Adaptation to such conditions is thus critical for these species to occupy this niche" does not seem to be well defended. For example, tree species living in forest gaps may be subjected to stronger stress, while the seedlings of canopy species may grow understory. For example, Ng et al. 2021 tested the importance of drought stress using the retained duplicated genes, and here the authors could test it using the signature of selection on each gene. Even if it is supported or not, the test can be interesting.

>>R: Manty thanks for the comments. In the previous version, perhaps we did not make it clear why we conducted these two comparisons and how we identified these genes. This may lead to some misunderstandings and give the impression that we were cherry-picking these genes. The two comparisons were intended to find genes responsible for the adaptation to the emergent layer of tropical rainforest. We did not report GO or KEGG enrichment results as none terms were significant due to the limited number of total genes. The genes were located systematically by intersecting the functional annotation of comparisons. We have made a thorough revision to clarify these points. Please see below for the related revision in the results section. We hope these revisions convince you that our approach was systematic and unbiased.

In the results, we revised and added:

Since Dipterocarpoideae and Malvaceae species involved were both tropical tree species but occupying different canopy layers, the comparative analysis between them aimed to identify positively selected genes of dipterocarps specifically adapted to the emergent canopy layers. The comparative analysis against temperate tree species was then to identify the positively selected genes adapted to tropical environments. The intersection of these two sets of genes thus would provide robust signatures of positive selection regarding the adaptation of Dipterocarpoideae to the emergent canopy layer of tropical rainforests.

The comparative analysis against Malvaceae identified 171 positively selected genes, whereas that against temperate tree species identified 191 (Supplementary Fig 6). As the number of genes was limited and the corresponding KEGG and GO enrichment analysis yielded no significant results, we presented the functional

annotation of GO and KEGG of these genes instead (Supplementary Tables 11 and 12). We found that both comparative analyses annotated functions relating to DNA repair and antioxidation, including Nucleotide excision repair (KEGG), Riboflavin metabolism (KEGG), DNA repair (GO), Double-strand break repair via homologous recombination (GO), Nucleotide-excision repair (GO) and Response to oxidative stress (GO) (Fig. 3a,b; Supplementary Tables 11 and 12). Such a consistent functional annotation arose from different genes identified in either analysis, such as Bloom syndrome protein (BLM) in DNA repair, UDP-sugar pyrophosphorylase (USP) in antioxidation, DNA mismatch repair protein (MSH6) in DNA repair and Pyridoxine 4-dehydrogenase (PLR1) in antioxidation (Supplementary Tables 13 and 14). More importantly, such a consistent functional annotation also arose from the same genes identified by both analyses, including DNA repair protein RAD51D (in homologous recombination), Structural maintenance of chromosomes protein 5 (SMC5), and riboflavin kinase (RFK) in antioxidation (Fig. 3c, Supplementary Fig. 7, Supplementary Tables 13 and 14).

As you suggested, adaptation to drought may also be an important mechanism for dipterocarps to thrive in tropical environments. This adaptation is not specific to the emergent canopy layer but to the overall tropical environments. In our comparison with temperate tree species, we have identified two key candidate genes associated with this adaptation. Please see below for the related revision in the results section.

In the comparative analysis against temperate tree species, we identified two positively selected genes (O-succinylbenzoate-CoA ligase (AEE14) and L-ascorbate peroxidase relevant to the plants' response to drought stress (Supplementary Table 13), which is involved in KEGG pathways of ubiquinone and other terpenoid-quinone biosynthesis (map00130) and glutathione metabolism (map00480), respectively. The finding of these positively selected genes is consistent with a previous study indicating that dipterocarps have adapted to the rare and irregular drought events in tropical environments¹⁷.

>I do not think the authors answered the point. Diverse genes were identified by the selection analysis. Although the lack of enrichment in Gene Ontology analysis or

any other analysis, the authors picked up several genes relevant to UV responses. Seeing the same data, other authors can make their own stories. This is not considered objective.

>>R: We highly appreciate this thoughtful comment. We have now understood that we had arbitrarily selected genes and established their associations with UV adaptation. Therefore, we must increase the objectivity in identifying genes under selection. To achieve this, we intersected the positively selected genes from the two comparisons and found 37 genes (these are listed in Supplementary Table 13), four of which were related to the stress response according to previous studies. Moreover, we also have made efforts to improve the objectivity in interpreting our results. As *Dipterocarps* experience a range of stresses, including UV and drought stresses, the positively selected genes should be interpreted as adaptation to such a broad range of stressors. To achieve both ends, we made a thorough revision in the abstract, introduction, results and discussion.

In the abstract, we revised the corresponding interpretation as:

Several positively selected genes were involved in antioxidation and DNA repair functions, likely facilitating adaptation to environmental stresses in Asian rainforests. (Lines 44 – 46)

In the introduction, we added:

As a consequence of above-canopy life in Asian rainforests, these species are challenged with various abiotic stresses such as irregular droughts¹⁵ and intensive ultraviolet (UV) radiation¹⁸. (Lines 83 – 84)

In the results, we revised the corresponding part as:

There were 37 positively selected genes supported by both comparisons, among which four genes were found to be associated with plants response to environmental stresses (Supplementary Table 13). These four genes included DNA repair protein RAD51D (in homologous recombination RAD51), structural maintenance of chromosomes protein 5 (SMC5), and carotenoid ϵ -hydroxylase (LUT1) involved in DNA repair, and riboflavin kinase (RFK) involved in antioxidation (Fig. 3c, Supplementary Fig. 6 and 7, Supplementary Table 13). RAD51²⁷ and SMC5²⁸ are found to be associated with UV adaptation, and LUT1²⁹ and RFK³⁰ have been reported to be linked with drought

tolerance. Moreover, according to the functional annotation in DNA repair and antioxidation, we identified several additional genes in either analysis, such as Bloom syndrome protein (BLM) and DNA mismatch repair protein (MSH6) in DNA repair, and UDP-sugar pyrophosphorylase (USP) and pyridoxine 4-dehydrogenase (PLR1) in antioxidation (Supplementary Table 14). (Lines 217 – 228)

Given that DNA repair and antioxidation are involved in the responses of plants to most environmental stresses^{18, 31}, these results effectively revealed the molecular footprints likely contributing to the adaptation of dipterocarps to environmental stresses in tropical rainforests. (Lines 235 – 238)

In the discussion, we revised the corresponding part as:

Dipterocarpoideae species benefit greatly from the height of their adult trees, by which they preempt light resources and facilitate pollen and samara dispersal^{16, 46}, but these advantages come with the costs, such as direct exposure of adult trees to intense UV and high vulnerability to droughts. Given the positively selected genes supported by both comparisons are functionally associated with these stresses²⁷⁻³⁰, our results offer the molecular basis for further exploring the adaptation of Dipterocarpoideae to environmental stresses in Asian rainforests. (Lines 329 – 334)

>>>I do not think that the authors after all answered to the point: "Temperate tree comparison" or "Malvaceae comparison" does not seem to make sense because dipterocarp species and Malvaceae species would form a single clade, respectively, and after all, there is only a single comparison." The two groups has many differences other than canopy layers. Two comparisons the authors provided are fundamentally the same.

I found that the statement in the abstract is too strong. At most, "genes that showed a signature of selection" may be fine.

>>R: This comment is gratefully accepted. We therefore have removed the relevant results and screened the positively selected genes associated with the plants' responses to environmental stresses according to the published literature. In addition, we have revised the abstract using "genes that showed a signature of selection".

In the abstract, we revised:

We found several genes that showed a signature of selection, likely associated with the adaptation of Dipterocarpoideae to Asian rainforests.

In the results, we revised:

To explore the molecular footprints associated with the adaptation of Dipterocarpoideae, we conducted comparative genomic analyses between five diploid Dipterocarpoideae species and five species of temperate trees (i.e., temperate tree comparison) (Supplementary Table 9). This comparative analysis was expected to identify the positively selected genes adapted to the stresses of tropical environments.

The comparative analysis identified 192 positively selected genes (Fig. 3a). As the number of genes was limited and the corresponding KEGG and GO enrichment analysis yielded no significant results, we presented the functional annotation of GO and KEGG of these genes instead (Supplementary Tables 11 and 12). After searching published literature, 20 positively selected genes were found to be associated with plants response to environmental stresses (Supplementary Fig. 6 and 7; Supplementary Table 13). These genes were involved in DNA repair, abnormal mRNA degradation, antioxidation, anaerobic respiration, endoplasmic reticulum associated degradation, pathogen resistance and cellular energy homeostasis (Supplementary Table 13). Among these genes, seven (e.g. DNA repair protein RAD51D²⁷ and structural maintenance of chromosomes protein 5 (*SMC5*)²⁸) were detected to be associated with UV adaptation; nine (e.g., carotenoid ϵ -hydroxylase (*LUTI*)²⁹, and riboflavin kinase (*RFK*)³⁰) and five genes (e.g., molecular chaperone GrpE (*GRPE*)³¹) were related with drought and heat tolerance; we also found three and two genes participating in the plants' responses to salt stress and flooding. These results thus revealed the molecular footprints likely contributing to the adaptation of dipterocarps to environmental stresses in tropical rainforests.

All these genes were expressed in all the five diploid Dipterocarpoideae species (Supplementary Fig. 8). Furthermore, to validate the function of the positively selected genes, we selected UDP-sugar pyrophosphorylase (*USP*), and to further validate the

results of functional annotation, the gene pyridoxine 4-dehydrogenase (*PLRI*) was also chosen. We used the sequences of these two genes from the genome of *H. chinensis* and synthesized their recombinants (Supplementary Table 14), which were used to conduct *in vitro* enzyme activity assay (see Methods). The final products of the *in vitro* reactions identified by LC-MS are consistent with the standards (Fig. 3b), validating the enzyme activity of the two genes in synthesizing UDP-glucose and pyridoxine, respectively.

The comparison revealed three expanded and 53 contracted gene families (Fig. 3a; Supplementary Table 15). Among the expanded gene families, we identified one family of disease-resistance proteins (Supplementary Tables 16 and 17), which is absent in the temperate tree species and is specific to Dipterocarpoideae species (Fig. 3c). Ten contracted families were annotated in the plant-pathogen interaction pathway, including seven families of leucine-rich repeat-receptor-like kinases (LRR-RLKs), two families of coiled coil-nucleotide binding site-leucine rich repeat (CC-NBS-LRR) and one toll interleukin receptor-nucleotide binding site-leucine rich repeat family (TIR-NBS-LRR) (Fig. 3c, Supplementary Tables 16 and 17). These gene families are the major protein receptors through which plants recognize pathogens^{32, 33}.

6. data availability

Data availability section should state more specifically the availability of the assemblies and annotation files.

>>R: We have revised the data availability statement as follows:

All data used in the analysis of this study have been deposited in the CNSA (<https://db.cngb.org/cnsa/>) in the BIG Data Center, Beijing Institute of Genomics (BIG), Chinese Academy of Sciences, BioProject: ID: PRJCA017262.

>This is fine, but as noted at the beginning, code availability is missing although the Editorial Check List said "We have provided a full code availability statement in the manuscript". Reproducibility is important.

>>R: Many thanks for addressing the importance of reproducibility of our results. We have provided the codes of bioinformatic analysis, which are available online (see Methods and the table blow).

Software Code source

RAxML v8.0.19 [https://github.com/ Molecology/Dipterocarpoideae-genome/
Supplementary_Code1.RAxML.sh](https://github.com/Molecology/Dipterocarpoideae-genome/Supplementary_Code1.RAxML.sh)

MCMCtree in PAML [https://github.com/Molecology/Dipterocarpoideae-genome/
Supplementary_Code2.mcmctree](https://github.com/Molecology/Dipterocarpoideae-genome/Supplementary_Code2.mcmctree)

KaKs_Calculator v2 [https://github.com/Molecology/Dipterocarpoideae-genome/
Supplementary_Code3.KaKs_Calculator](https://github.com/Molecology/Dipterocarpoideae-genome/Supplementary_Code3.KaKs_Calculator)

QUOTA-ALIGN script [https://github.com/Molecology/Dipterocarpoideae-genome/
Supplementary_Code4.QUOTA-ALIGN](https://github.com/Molecology/Dipterocarpoideae-genome/Supplementary_Code4.QUOTA-ALIGN)

RGAugury pipeline [https://github.com/Molecology/Dipterocarpoideae-genome/
Supplementary_Code5.RGAugury.sh](https://github.com/Molecology/Dipterocarpoideae-genome/Supplementary_Code5.RGAugury.sh)

Seqencing mapping and variant calling
[https://github.com/Molecology/Dipterocarpoideae-genome/
Supplementary_Code6.mapping.varcall.sh](https://github.com/Molecology/Dipterocarpoideae-genome/Supplementary_Code6.mapping.varcall.sh)

SMC++ v 1.15.2 [https://github.com/Molecology/Dipterocarpoideae-genome/
Supplementary_Code7.SMC.sh](https://github.com/Molecology/Dipterocarpoideae-genome/Supplementary_Code7.SMC.sh)

PSMC v0.6.4 [https://github.com/Molecology/Dipterocarpoideae-genome/
Supplementary_Code8.PSMC.sh](https://github.com/Molecology/Dipterocarpoideae-genome/Supplementary_Code8.PSMC.sh)

PROVEAN v1.1.5 [https://github.com/Molecology/Dipterocarpoideae-genome/
Supplementary_Code9.proven.sh](https://github.com/Molecology/Dipterocarpoideae-genome/Supplementary_Code9.proven.sh)

SIFT4G v2.0.0 [https://github.com/ Molecology/Dipterocarpoideae-genome/
Supplementary_Code10.SIFT.sh](https://github.com/Molecology/Dipterocarpoideae-genome/Supplementary_Code10.SIFT.sh)

Derived alleles identification [https://github.com/Molecology/Dipterocarpoideae-
genome/ Supplementary_Code11.derived](https://github.com/Molecology/Dipterocarpoideae-genome/Supplementary_Code11.derived)

Runs of homozygosity in PLINK [https://github.com/Molecology/Dipterocarpoideae-
genome/ Supplementary_Code12.ROH.sh](https://github.com/Molecology/Dipterocarpoideae-genome/Supplementary_Code12.ROH.sh)

>>>**This is fine.**

>>**R: Thank you very much.**

Minor issues

line 184, which is involved

>>R: Revised. Line 242.

>Fine.

>>R: Thanks.

Reviewer #2 (Remarks to the Author):

Comments for authors

NCOMMS-22-41572B: Liu et al. Life up high: the adaptations and demography of the Asian rainforest titans, Dipterocarpoideae.

Page 6&7, lines 119-124: The histograms obtained from ploidy screening using the DNA flow cytometry were not really convincing particularly showing broad DNA peak rather than single prominent DNA peak. Even the standard reference DNA peak was broad. Perhaps this may be due to the handling error during samples chopping/preparation. I strongly encourage this to be addressed.

>>R: Many thanks for this constructive suggestion, and we have addressed this in the legend of Supplementary Fig.1, as “The histograms show broad DNA peaks for both tetraploid *Hopea* species and the reference species, probably due to some handling errors during sample chopping/preparation.”

Comment:

Unfortunately, this is not a satisfactory response I was expecting. To resolved this, additional flow cytometry experiment on the samples need to be carried out. Broad DNA peak is just not convincing (even for standards reference DNA). Please refer to Dolezel et al 2007 Nature Protocols.

>>R: To enhance the quality of flow cytometry results, we have resampled the wild trees of *Hopea* species and re-performed flow cytometry experiments. After referring Dolezel et al. (2007), we have adjusted the parameters in the experiments (e.g., voltage). Using the modified protocol, we ran individual samples for each of the three *Hopea* species (*H. Chinesis*, *H. hainanensis* and *H. reticulata*) and *S. lycopersicum* separately,

and ran samples combining each *Hopea* species with *S. lycopersicum* to estimate the genome size (Supplementary Fig 1c (the upper panel) and Supplementary Table 1). Then, we ran diploid *Hopea* species and tetraploid *Hopea* species together to validate ploidy (Supplementary Fig. 1, the lower panel). We hope you find the results with sharper peaks satisfactory to demonstrate the ploidy of these species.

Supplementary Table 1. The results of genome size estimation using flow cytometry setting the genome of *Solanum lycopersicum* (genome size: 2.07 Gb; 2C = 2.12) as the reference.

Sample	Fuorescence intensity	CV error (%)	2C (pg)	Genome size (Gb)	Ploidy
Hopea chinensis	18.20	4.30	0.81	0.79	2X
Solanum lycopersicum	47.40	1.76	2.12	2.07	
Hopea hainanensis	29.20	4.71	1.60	1.56	4X
Solanum lycopersicum	38.80	3.61	2.12	2.07	
Hopea reticulata	26.90	4.88	1.64	1.60	4X
Solanum lycopersicum	34.70	3.29	2.12	2.07	

Page 16: lines 348-351: This is crucial. The natural distribution of *H. hainanensis* only can be found in Hainan province and northern part of Vietnam. They suggested that previously reported genome of *H. hainanensis* by Wang et al. 2022 was ‘probably’ from diploid population. Where is the evidence of diploid populations?

>>R: Apologies for the confusion. The intended meaning previously was: *H. hainanensis* may have both tetraploid and diploid populations. We adopted the “may” to indicate this was highly speculative (Please see our reply below). The sequenced individual by Wang et al. (2022) was collected from Ruili Botanical Garden (Yunnan, China), and the field origin of this sample remains unknown. To avoid misunderstandings, we have removed the sentence indicating the possibility of both tetraploid and diploid populations.

Comment:

The above explanation is not convincing to me. I would suggest that the authors to include some form of empirical evidence or justification in their manuscript as to why *H. hainanensis* is a tetraploid species compared to that of Wang et al 2022 which was reported as diploid individual. Perhaps, the used of a study by Wang et al 2020 (PloS One) using 12 SSR markers on 10 populations of *H. hainanensis* (refer S1 File) as support that the allelic dosage was used to determine the amplified peaks based on ratios between peak intensities from SSR markers suggesting autotetraploidy.

>>R: Thanks. We understand and fully agree that it is important to justify multi-ploidy for *H. hainanensis*, in particular, as the genus *Hopea* species often shows multi-ploidy (Please see our reply below). The challenge is that we have little knowledge of individuals/populations beyond our study, and all samples surveyed in this study are validated as tetraploid by *k*-mer analysis (Please see our reply below). As to the population genetic study by Wang et al. (2020) using SSR markers, this study has taken tetraploidy as default since the authors had contacted with us and known the results of genome assembly of *H. hainanensis*. To ensure accuracy, we have added the following statement to the discussion: “While the previous study suggested the existence of diploid populations of *H. hainanensis*¹⁴, *k*-mer analysis revealed all individuals of *H. hainanensis* sampled in this study from Hainan Island as tetraploids (Supplementary Fig. 1d).”. Further studies are necessary to address the multi-ploidy issue in *Hopea*.

If both tetraploid and diploid populations do exist, how do they know the 30 H.

hainanensis individuals used for resequencing study were all autotetraploid individuals (minus 4 individuals tested for kmer analysis)?

>>R: It is a nice idea to validate that all individuals are tetraploid by k-mer analysis. Accordingly, we conducted k-mer analysis for the remaining 26 samples, and they are indeed all tetraploid, with one sample (JF4) difficult to judge. Except this sample, the depth of the common peak on the right was four times that of the heterozygous peak (Please refer to the figure below), supporting that they are tetraploid. It appears that the tetraploid pattern is only apparent with extremely deep sequencing (Supplementary Fig.1a), relatively clear with deep sequencing (Supplementary Fig.1o-r, showing individuals with the highest resequencing depth), and almost invisible at low depths (Supplementary Fig. 1 in Wang et al. 2022). Supplementary Fig. 1 in Wang et al. (2022) only had one peak (possibly due to low sequencing depth), providing no clear support of diploid or tetraploid.

Comment:

The kmer analysis is acceptable. It should be included as supplementary figure and revise the main text accordingly (y-axis: ‘Frequency’). However, there is one individuals JF4 need to be clarified further as to the ploidy status of the individual. This could be a diploid individual. If there are the actual occurrences of diploid and tetraploid mix within population these need to be clarified.

>>R: This comment is really helpful when trees with different ploidies may be mixed within a single species. According to your suggestions, we have now included *k*-mer analysis for each re-sequenced sample in Supplementary Fig. 1 and revised the title of y-axis as “Frequency”.

For the individual JF4, we have re-extracted DNA and re-sequenced an addition of 25Gb data, and conducted *k*-mer analysis by merging with the previous Data. Compared with the previous analysis, the new result yields a clear pattern of tetraploidy for JF4. Consequently, this analysis reveals all samples collected in this study for *Hopea hainanensis* as tetraploids.

The figure on the left used the previous re-sequencing data, and the figure on the right (the new result) used the merged data.

Additional comments:

Lines 41-43: This is a rather bold and misleading claim. I would suggest it to be removed. The estimated divergence time between Dipterocarpoideae and Malvaceae was not consistent with generally accepted divergence timeline of various angiosperm/eudicots. The statement “Gondwanan vicariance hypothesis” just came out all of the sudden. Completely no discussion on Gondwanan vicariance hypothesis in the main text. No doubt, the timing and the nature of Gondwanan breakup is highly debatable but a meaningful and convincing discussion in relation to the current findings need to be presented.

>>**R:** This suggestion is gratefully accepted. We agree that this claim is misleading and the divergent time between Dipterocarpoideae and Malvaceae requires further justifications. In the analysis, we incorporated additional taxa and calibration points for phylogenomic analysis following the suggestions from Reviewer 1 (The new analysis results are presented below, for details, please refer to our response to Reviewer 1). The revised results now do not contradict the estimations of divergent time for eudicots (~132 (161-125) MYA and others, reviewed by Li et al. 2019).

Following your suggestion, we revised the statement in the abstract as “**The divergence time between Dipterocarpoideae and Malvaceae and within**

Dipterocarpoideae was estimated to be 108.2 (97.8–118.2) and 88.4 (77.7–102.9) million years ago”, and removed the mention of the “Gondwanan vicariance hypothesis”. To provide background on the Gondwanan origin of Dipterocarpoideae and put our findings into the contexts of generally accepted evolutionary history of angiosperms, we revised the introduction and discussion (Please see below).

In the introduction, we revised:

Earlier studies proposed that Dipterocarpaceae originated in Western Gondwanaland in Early Cretaceous (*c.*120 million years ago (MYA))¹⁶. A recent phylogenomic study¹³ supports this Western Gondwanaland origin, showing a very ancient divergence between Dipterocarpaceae and its closest family Malvaceae (147.3 MYA¹³). New evidence from pollen fossils suggested the diversification of Dipterocarpaceae since *c.*102.9 MYA¹⁷, and indicated that the dispersal of dipterocarps from Africa to India occurred during the late Cretaceous through Kohistan-Ladakh Island Arc and subsequently to Southeast Asia during the middle to late Eocene after India-Asia collision. In contrast to these findings, genomic studies involving different sets of dipterocarp species revealed a more recent divergence between Dipterocarpaceae and Malvaceae (*c.* 84 MYA¹⁴ and 86–98 MYA¹⁵).

In the discussion, we revised:

The estimated divergent time between Dipterocarpoideae and Malvaceae in this study appears to be earlier than the results from Ng et al.¹⁵ and Wang et al.¹⁴, and later than Tian et al.¹³. The estimated crown age of Dipterocarpoideae overlaps with the previous estimation by Bansal et al.¹⁷ (~94.6 MYA). Although the estimations in this study did not contradict the crown age of eudicots (c. 132 MYA)⁴⁰, it indicates an earlier origin of Malvales than previously⁴⁰, and appears to support the ancient origin of Dipterocarpaceae (c.120 MYA) in Western Gondwana¹⁶. This origin is more recent than the separation of Eastern and Western Gondwana, and the dispersal of dipterocarps to India likely occurred through the Kohistan-Ladakh Island Arc, which collided with the Indian Plate in the late Jurassic¹⁷. The dispersal to Southeast Asia was likely associated with the collision between the Indian and the Asian Plates in the late Eocene¹⁷. In particular, the genus *Hopea* was primarily distributed across Southeast Asia, and the estimated crown age of this genus (35.7 MYA) in our study suggests its diversification after the India-Asia collision. Thus, by integrating more genomes, our results support the tropical-African origin of Dipterocarpoideae and the role of Africa-India floristic interchange in forming the current distribution of Dipterocarpoideae.

Line 109: ‘PHD’ or ‘HPD’= highest posterior density. Please check it throughout the MS.

>>R: Thanks. We have checked it throughout the manuscript and corrected this mistake.

Line 197 & Line 199: Please be specific on what you meant by “...of the three species”

>>R: Thanks. We meant the three *Hopea* species (*H. chinensis*, *H. reticulata* and *H. hainanensis*), and we have revised the relevant contents. Lines 193-194 and 196.

Lines 198-201: Just curious, from Fig.2a, what would be the explanation for *Hopea odorata* as this species has been previously identified as 2x (2n=14) and 3x (2n=20-

22) Kaur et al. (1986); Sarkar et al. (1982); Jong & Lethbridge (1967), Roy & Jha (1965); Tixier (1960). This would be interesting discussion as to why *Hopea* spp. exhibit several types of ploidy levels presently.

>>R: Many thanks for the references; we have carefully checked them, except Tixier (1960), which is in French, and Roy & Jha (1965, *The Journal of the Indian Botanical Society*, 44:387-397), which we couldn't obtain the full text. An interesting discovery is that 3X (2n=20-22) samples of *Hopea odorata* were collected from Malaysia by Kaur et al. (1986) or by Oginuma et al. (2020) from Singapore, whereas diploid samples (2n=14) were collected from east India (close to the border between India and Bangladesh) by Sarkar et al. (1982). As *Hopea odorata* is distributed among Bangladesh, Cambodia, India, Laos, Malaysia, Myanmar, Thailand, and Vietnam, these pieces of information indicate diploids may exist around the distribution center, while triploids may be present in the distribution boundary. We are not sure if it is the case *H. hainanensis* and *H. reticulata*. In our study, all individuals we re-sequenced were tetraploid (please see our reply above), and the populations were from the distribution boundary of *Hopea* genus, but we have little knowledge of the origin and population of the diploid *H. hainanensis* sequenced by Wang et al. (2022) (It was indicated as collected from Ruili Botanical Garden, Yunnan, China, and its field origin remains unknown.).

For the question why *Hopea* spp. exhibit several types of ploidy levels, we speculate that it may be related to climate changes during the diversification of this lineage. Please see our revision below.

In the discussion, we revised:

Karyotype analysis reveals that multiple species within the genus *Hopea* have polyploid populations^{44, 45, 46}, including *Hopea odorata* (3X), *Hopea subalata* (3X) and *Hopea nutans* (4X). Polyploidy appeared to be commonly observed in habitats characterized by climatic and edaphic fluctuations, and most documented polyploidization events to date are closely linked to environmental changes⁴⁷. The diversification of this genus is quite recent in Dipterocarpoideae, with a crown age of 35.7 (95% HPD: 18.4–48.2)

MYA (Fig. 2a), corresponding to the late Eocene¹⁷. The temperature decline and fluctuations in the Pliocene, along with irregular drought events in this area^{15, 44} may have contributed to genome duplications or polyploidizations of this lineage. While the previous study suggested the existence of diploid populations of *H. hainanensis*¹⁴, *k*-mer analysis revealed all individuals of *H. hainanensis* sampled in this study from Hainan Island as tetraploids (Supplementary Fig. 1d). Such distribution ranges of *H. hainanensis* and *H. reticulata* are located at the northern boundary of Asian tropical zone, where significant fluctuations in climates (e.g., cooler winters and highly variable precipitation) are anticipated compared with central regions of Asian rainforests. These environmental stresses may be relevant to the polyploidization of these two *Hopea* species.

Fig. 2a: I do not see the purpose of putting the fruit images in the figure unless they mean something.

>>**R:** The fruit images were intended to show the diversity of fruit shapes/sizes of the species we sequenced. Indeed, such information is not relevant to the main theme of our manuscript. We thus removed them in the revision. Please see the revised Fig. 2a.

Reviewer #3 (Remarks to the Author):

In an effort to address concerns about running PSMC and SMC++ demographic analyses on polyploid plants, the authors used non-polyploid reference genomes to map their reads to. If the reference and mapped species are +/- evolutionarily close, there seems to be nothing "wrong" with this approach. We have found, however, that running such analyses with reads mapped to distant references can heavily bias results; as such, in our projects, we have resorted to running PSMC (for example) only in self-self mode, i.e., using the same species' assembly as reference for reads mapping, even if that assembly is nowhere near as contiguous as a chromosome-scale reference. The limitations of using shorter contigs appear to outweigh the limitations of evolutionarily distant (i.e., poor) reads mapping. The

results shown in Fig 4a are coincident enough to suggest that any reference bias was not appreciable. However, the *Hopea* SMC++ curves show slight shifts in years before present between the two species, which could reflect heterozygosity differences between them rather than real demographic/geological-event differences. I suggest the authors refer to this possibility, which was recently explored in detail as part of demographic work for the lychee genome paper: <https://www.nature.com/articles/s41588-021-00971-3> - see supplementary material, Supplementary Note II.

>>R: Thank you very much for this constructive comment, which improves our understanding of demographic analysis and consolidates our results. Here, we assembled the chromosome-level genomes of *Hopea hainanensis* and *Hopea reticulata*, and mapped resequencing data to their respective genomes for the demographic analysis. Our approach is in line with your suggestion: running a self-self model and using references of long contigs, which is likely to yield unbiased results. The supplementary note of Hu et al. (2022) shows a crucial simulation result with SMC++: altering the number of heterozygous sites in the genome biases the estimation of effective population sizes and causes a shift of the demographic curve along the timeline. Many thanks for your suggestion, and accordingly, we referred to Hu et al. (2022) and added this caveat to the discussion as:

Note that the SMC++ curves of these two *Hopea* species showed slightly shifts in time, which is likely to reflect heterozygosity differences between them rather than real demographic/geological-event differences⁵¹.

REVIEWERS' COMMENTS

Reviewer #2 (Remarks to the Author):

Comments to authors

NCOMMS-22-41572C: Liu et al. Life up high: the adaptations and demography of the Asian rainforest titans, Dipterocarpoideae.

My concerns have been addressed. I have no further comments and would like to congratulate the authors on a nice piece of work!